# evortran: a modern Fortran package for genetic algorithms with applications from LHC data fitting to LISA signal reconstruction

**Thomas Biekötter[1]**[⋆]

**1** Instituto de Física Teórica UAM/CSIC, Calle Nicolás Cabrera 13-15,
Cantoblanco, 28049, Madrid, Spain

⋆ thomas.biekoetter@desy.de

## Abstract

`evortran` **is a modern Fortran library designed for high-performance genetic algorithms and evolutionary optimization.** `evortran` **can be used to tackle a wide range of problems in high-energy physics and beyond, such as derivative-free parameter optimization, complex search taks, parameter scans and fitting experimental data under the presence of instrumental noise. The library is built as an** `fpm` **package with flexibility and efficiency in mind, while also offering a simple installation process, user interface and integration into existing Fortran (or Python) programs.** `evortran` **offers a variety of selection, crossover, mutation and elitism strategies, with which users can tailor an evolutionary algorithm to their specific needs.** `evortran` **supports different abstraction levels: from operating directly on individuals and populations, to running full evolutionary cycles, and even enabling migration between independently evolving populations to enhance convergence and maintain diversity. In this paper, we present the functionality of the** `evortran` **library, demonstrate its capabilities with example benchmark applications, and compare its performance with existing genetic algorithm frameworks. As physics-motivated applications, we use** `evortran` **to confront extended Higgs sectors with LHC data and to reconstruct gravitational wave spectra and the underlying physical parameters from LISA mock data, demonstrating its effectiveness in realistic, data-driven scenarios.**

# 1   Introduction

Genetic algorithms (GAs) are a class of optimization techniques inspired by natural selection and evolution. Unlike gradient-based optimization methods, which rely on derivative information to navigate the solution space, GAs explore the solution space through stochastic processes that mimic the principles of biological evolution. A so-called *population* of candidate solutions, each represented as an *individual*, is iteratively evolved over successive generations. Each individual is represented by a set of parameters referred to as *genes*, and the *fitness* of an individual is a measure of how well it solves the optimization problem at hand. The fitness is obtained

by computing the so-called fitness function that depends on the genes of the individual.[1] In order to evolve one or a set of populations, a GA applies genetic operations that act on the individuals:

1. **Selection:** The selection step involves choosing individuals from the population for reproduction based on their fitness, with fitter individuals having a higher probability of being selected to reproduce and pass on their genes.

2. **Crossover:** The crossover (also called mating) step involves combining the genes of two or more parent individuals to create one or more offspring individuals, with the goal of producing new solutions that inherit desirable traits from the parent individuals.

3. **Mutation:** The mutation step introduces random changes to the genes of a subset of the offspring individuals, ensuring diversity within the population and helping to prevent premature convergence.

4. **Elitism:** The elitism step involves preserving a certain number of the fittest individuals from one generation to the next, ensuring that the best solutions are kept and not lost due to the stochastic nature of the previous selection, crossover and mutation steps.

In addition to the total number of individuals contained in the populations, a specific GA is defined by the precise operations performed at each of the four steps and the probabilities assigned to each of these operations. Different optimization problems often require vastly different choices for these operations and probabilities. It is therefore essential to have a flexible framework that allows for easy customization and adaptation to different optimization tasks.

The stochastic-driven approach allows GAs to explore complex, high-dimensional, nonconvex, discontinuous and noisy solution spaces where traditional methods may struggle due to premature convergence to local minima or undefined gradients. GAs have shown great potential in several areas of high-energy physics and cosmology. For instance, in string theory GAs have been used to explore the vast landscape of viable string vacua with phenomenologically consistent properties [1–11]. In particle physics phenomenology, GAs have been applied to scan the parameter spaces of beyond-the-Standard-Model (BSM) theories in order to identify regions of parameter space that are compatible with theoretical and experimental constraints [12–19]. GAs have also played an important role in numerical fitting tasks, such as in the determination of parton distribution functions [20–23]; in the analyse, reconstruction and classification of (detector-level) events at particle accelerators [24–27]; and in the extraction of neutrino oscillation [28–32], astrophysical [33,34], and cosmological parameters [35–41]. These examples are not intended to be exhaustive, but rather to illustrate the broad applicability of GAs across both experimental and theoretical fields of physics.

GAs often require significant computational costs in terms of evaluations of the fitness function due to their stochastic nature, and they may converge slower to the optimal solution compared to derivative-based techniques, such as gradient descent, especially when a high degree of precision is required. This highlights the need for fast and scalable implementations of GAs to make them suitable for large-scale high-dimensional optimization problems. Several well-established libraries exist already that provide implementations of GAs across different programming languages. In Python, popular open-source packages include DEAP [42], a flexible evolutionary computation framework, and PyGAD [43] that supports training deep

---

[1]The fitness function in GAs does not necessarily depend solely on the genes of a single individual. It can also incorporate external parameters that change over time to simulate environmental changes, or depend on the genes of a subset or the entire population to define a relative fitness value. This allows for more complex selection mechanisms, such as competitive or cooperative fitness evaluation, but also increases the number of calls of the fitness function. evortran supports such generalized fitness functions, but for brevity, all examples in this paper will assume the most common case, where the fitness function maps the genes of each individual directly to a fitness value without further dependencies.

learning models created with `Keras`. GA framworks in C and C++ which provide efficient and customizable implementations are, for instance, `CMAES` [44] `GAUL` [45], `EO` [46] and `OpenGA` [47]. Furthermore, `PGAPack` is a general-purpose GA framework written in C and using the Message Passing Interface (MPI) for parallelization [48]. In the Fortran ecosystem, a well-known optimization library based on GAs is `Pikaia` [49], which was originally developed for astrophysical applications and recently has become available as a modern Fortran package converted to free-form source and with a new object-oriented user interface [50].

While several GA libraries with varying degree of flexibility exist, `evortran` was developed to provide a modern Fortran-based alternative that balances performance and ease of use. Like `Pikaia`, it is available as a Fortran package manager (`fpm`) [51,52] package, making the compilation, installation and integration into other Fortran programs seamless. Compared to C/C++ libraries, `evortran` offers a more user-friendly interface, while still making use of the increased computational performance of a low-level programming language. In addition, its main optimization routines can be accessed from Python via lightweight wrappers, allowing seamless integration into Python-based workflows and outperforming native Python implementations. One of the key strengths of GAs is their inherently modular structure, which allows for straightforward and efficient parallelization, operating on individuals in parallel across one or more populations. To take advantage of this property, `evortran` includes support for parallel execution using `OpenMP`, which significantly reduces the runtime on multi-core systems. This makes `evortran` and attractive choice for computationally intensive optimization problems in scientific computing. Moreover, `evortran` is designed with a modular structure that allows new GA operations for selection, crossover, mutation and selection to be easily added, such that the library can naturally grow and evolve over time. Finally, `evortran` stands out due to its flexible design which allows users to work at different abstraction levels: (1) the user can operate on individuals directly, (2) the user can evolve whole populations, (3) and the user can evolve a set of populations in parallel with the possibility of periodically migrating individuals in between the populations.

The migration approach is particularly useful for optimization problems where the goals is not only to find the globally optimal solution but also to identify multiple diverse solutions with sufficiently good fitness (see also Ref. [19]). This feature is relevant for a variety of optimization tasks encountered in high-energy physics. For example, in studies of the phenomenology of BSM theories, one often performs extensive parameter space scans involving models with a large number of free parameters, while at the same time different sets of theoretical and experimental requirements constrain the allowed parameter space (see, e.g. Refs. [12,13,15–17]). These scans are constrained by experimental measurements, which come with their own uncertainties. In such contexts, the goal of the parameter scan is not merely to find a single best-fit point, but rather to identify all distinct regions of parameter space that satisfy the imposed constraints within the uncertainties. The migration-based approach of `evortran` is especially well-suited to this task, as independently evolving populations can converge to different viable regions of the parameter space. This enables a more comprehensive exploration and characterization of phenomenologically acceptable solutions which might be missed in simple random scans or when using gradient-based optimization methods.

The outline of the paper is as follows: in Section 2, we describe the design and core functionality of the `evortran` library in detail. Section 3 provides user instructions, including installation steps and usage guidelines. In Section 4, we present example applications that illustrate the capabilities of the library, focusing on well-known benchmark functions for optimization problems in Section 4.1 and on realistic physics applications in Section 4.2. Finally, Section 5 contains our conclusions. Readers who are already familiar with GAs and are primarily interested in how to install and use `evortran` may wish to skip directly to Section 3.

## 2 Library design and functionality

The design of `evortran` is centered around flexibility, modularity, and performance, enabling users to employ GAs using varying degrees of computational resources depending on the complexity and requirements of their optimization tasks. This section outlines the core components and features of the library, introducing the derived types and utilities that structure the implementation. As already mentioned in Section 1, a key concept underlying `evortran` is its multi-layered interface, which allows users to work at different abstraction levels: (1) work directly on arrays of individuals, (2) handle entire populations of individuals using population-level methods, and (3) apply evolution strategies acting on one or more populations in parallel through a migration framework, with optional inter-population exchange of individuals. This design makes `evortran` suitable for both fine-grained control and high-level optimization applications. The subsections that follow below are organized accordingly. We begin by introducing the derived types for the individual and population data structures in Section 2.1 and Section 2.2, respectively, which serve as the most fundamental building blocks of the library. We then present the available GA operations in Section 2.3, followed by the two key utilities that implement full population evolution in Section 2.4 and migration-based evolution in Section 2.5. Finally, we summarize the core numerical tools and auxiliary utilities that are included in `evortran` in Section 2.7.

### 2.1 Individuals

The most fundamental building block of `evortran` is the `individual` derived type, which represents a single candidate solution within a GA. `evortran` contains two types of individuals, one for integer valued genes and one for genes that can take continuous values as floating-point numbers.

#### 2.1.1 Integer individuals

We start by describing the version of individuals with integer-valued genes, hereafter referred to as *integer individuals*. The declaration of the `individual` type for integer individuals is as follows:

```fortran
type, public :: individual
  integer :: length
  integer :: base_pairs
  integer, dimension(:), allocatable :: genes
  procedure(func_abstract), pointer, private :: func => null()
  real(wp), private :: fitness = 0.0e0_wp
  logical, private :: fitness_calculated = .false.
contains
  procedure, public :: calc_fitness
  procedure, public :: get_fitness
  procedure, public :: reset_fitness
end type individual

interface individual
  procedure create_individual
end interface individual
```

The components of the derived type are as follows:
  - `length`: The number of genes in the individual. It is set during initialization and must be greater than 1.

- `base_pairs`: The number of distinct values that each gene can take. The default is 2, corresponding to binary genomes where each gene is either 0 or 1.
- `genes`: An allocatable integer array of size `length` that stores the genome of the individual.
- `func`: A procedure pointer that points to the fitness function associated with this individual. The fitness function is set during initialization.
- `fitness`: A real number storing the most recently computed fitness value.
- `fitness_calculated`: A logical flag indicating whether the current fitness value corresponds to the current genome. This avoids redundant calls of the fitness function.

To create a new integer individual, users can use the overloaded interface `individual`, which wraps the internal `create_individual` function. This provides a clean and intuitive way to construct an individual object by specifying the number of genes, a fitness function, the number of base pairs, and optionally a seed value for the genes at initialization. For example, a typical initialization might look like:[2]

```
1 use evortran__individuals_integer , only : individual
2
3 type ( individual ) :: ind
4
5 ind = individual ( length =10 , fit_func = fit_func , base_pairs =2)
```

This constructs an individual with 10 binary genes and associates it with the fitness function `fit_func`. Since the optional argument `seed` is not given, the gene values are assigned randomly within the allowed range. The fitness function must be defined by the user (either in the main program or in an external module) following the abstract interface defined as:

```
1 subroutine func_abstract ( ind , f)
2   class ( individual ) , intent ( in ) :: ind
3   real ( wp ) , intent ( out ) :: f
4 end subroutine func_abstract
```

Thus, the fitness function is a subroutine that takes an individual object as input and computes the fitness value that is returned as the second argument. In contrast to many existing GA frameworks that maximize the fitness function, evortran is designed to minimize it, and the fitness function should be defined accordingly. The type `real(wp)` uses the working precision `wp` defined in the module `evortran__util_kinds`. By default, `wp` corresponds to double precision, but it can be changed to quad precision at compile time by providing the preprocessor flag –DQUAD. Further details on compilation and installation are provided in Section 3. The declarations of the procedure associated to the derived type of integer individuals are as follows:

```
1 subroutine calc_fitness ( this )
2   class ( individual ) , intent ( inout ) :: this
3   real ( wp ) :: f
4 end subroutine calc_fitness
5
6 function get_fitness ( this ) result (f)
7   class ( individual ) , intent ( inout ) :: this
8   real ( wp ) :: f
```

---

[2]For brevity, we do not show the statements required to initialize the PRNG here and in the following code snippets contained in this section. These should be called at the beginning of any program that uses the evortran library. For details, see the discussion in Section 2.7.1.

```
 9  end function get_fitness
10
11  subroutine reset_fitness(this)
12    class(individual), intent(inout) :: this
13  end subroutine reset_fitness
```

These procedures carry out the following tasks:

- `calc_fitness`: Calculates the fitness with the current gene values by calling the fitness function if the fitness has not yet been computed.
- `get_fitness`: Same as `calc_fitness`, but also returns the computed fitness value.
- `reset_fitness`: Resets the fitness to zero and marks it as not yet computed, which is necessary after any genetic operation that modifies the genes of the individual.

### 2.1.2 Float individuals

In addition to supporting individuals with discrete integer-valued genes, `evortran` also provides a derived type for individuals with genes represented by continuous floating-point numbers. These *float individuals* are useful for optimization problems in continuous parameter spaces. Below is the definition of the derived type used to represent float individuals:

```
 1  type, public :: individual
 2    integer :: length
 3    real(wp) :: lower_lim = 0.0e0_wp
 4    real(wp) :: upper_lim = 1.0e0_wp
 5    real(wp), dimension(:), allocatable :: genes
 6    procedure(func_abstract), pointer, private :: func => null()
 7    real(wp), private :: fitness = 0.0e0_wp
 8    logical, private :: fitness_calculated = .false.
 9  contains
10    procedure, public :: calc_fitness
11    procedure, public :: get_fitness
12    procedure, public :: reset_fitness
13  end type individual
14
15  interface individual
16    procedure create_individual
17  end interface individual
```

Here, `wp` is the real kind working precision with which `evortran` operates, defined in the module `evortran_util_kinds`. By default, `evortran` uses double precision with 15 significant digits, but the precision can be changed to quadruple-precision with 30 significant digits at compile time, see Section 3.1. The components of the float individual type are:

- `length`: The number of genes in the individual. It is set during initialization and must be greater than 1.
- `lower_lim`: the lower bound of the valid range for gene values (default is 0.0).
- `upper_lim`: the upper bound of the valid range for gene values (default is 1.0).
- `genes`: a real-valued array of size `length` that stores the gene values.
- `func`: a procedure pointer to the fitness function.
- `fitness`: the cached fitness value.
- `fitness_calculated`: logical flag indicating whether the fitness has already been computed.

Furthermore, the derived type for the float individuals are associated to the same procedures as present in the type for integer individuals, which are described above. New float individual objects can be created using the generic interface `individual`, which internally calls the `create_individual` function, in the same way as for integer individuals. A float individual is initialized by providing the gene length, the fitness function, and optionally the lower and upper bounds for the gene values, as well as a random seed:

```
1 use evortran__individuals_float, only : individual
2
3 type(individual) :: ind
4
5 ind = individual(  &
6   length=10, fit_func=fit_func,  &
7   lower_lim=-1.0e0_wp, upper_lim=1.0e0_wp, seed=0.0e0_wp)
```

This creates a float individual with 10 floating-point number genes and associates it with the fitness function `fit_func`. The fitness function must be defined by the user using the same abstract interface as for the integer individuals discussed above. The optional arguments `lower_lim` and `upper_lim` enforce that allowed gene values should lie between -1 and 1, and the optional `seed` argument is provided to initialize all 10 gene values with the value 0.

### 2.1.3 Operate on individuals

Although `evortran` is primarily designed to carry out optimization tasks at higher abstraction levels, through evolving entire populations or even sets of populations, it is also possible to operate directly on individual objects. This can be useful for testing, debugging, or for users who require fine-grained control over their GA.

The following example program demonstrates how to manually create and operate on float individuals. It shows the initialization of two individuals, application of simulated binary crossover (see Section 2.3.2) to generate two offspring individuals, and it uses the shuffle mutation routine (see Section 2.3.3) to modify the genes of one of the two offspring individuals.

```
1 program operate_on_individuals
2
3   use evortran__util_kinds, only : wp
4   use evortran__individuals_float, only : individual
5   use evortran__crossovers_sbx, only :
        simulated_binary_crossing
6   use evortran__mutations_shuffle, only : shuffle_mutate
7   use evortran__prng_rand, only : initialize_rands
8
9   implicit none
10
11   type(individual) :: ind1
12   type(individual) :: ind2
13   type(individual) :: ind3
14   type(individual) :: ind4
15
16   call initialize_rands(mode='twister')
17
18   ind1 = individual(6, func)
19   ind2 = individual(6, func)
20
```

```
330  21     call simulated_binary_crossing(  &
331  22       ind1, ind2, ind3, ind4, eta_c=30e0_wp, p_c=0.5e0_wp)
332  23
333  24     call shuffle_mutate(ind3)
334  25
335  26   contains
336  27
337  28     subroutine func(ind, f)
338  29
339  30       class(individual), intent(in) :: ind
340  31       real(wp), intent(out) :: f
341  32
342  33       f = sum(ind%genes**2)
343  34
344  35     end subroutine func
345  36
346  37 end program operate_on_individuals
```

Specifically, this example illustrates the following sequence of operations:

- The pseudo-random number generator is initialized using the Mersenne Twister algorithm (see also Section 2.7.1) by calling `initialize_rands`.
- Two float individuals `ind1` and `ind2` are created with 6 genes each. They are associated with the fitness function `func`, which in this example computes the sum of squares of the gene values.
- The procedure `simulated_binary_crossing` performs crossover between the parent individuals `ind1` and `ind2`, producing two offspring individuals `ind3` and `ind4`. The parameters `eta_c` and `p_c` are optional arguments that control the shape and probability of the crossover, see Section 2.3.2 for details.
- A shuffle mutation is applied to one of the offspring (`ind3`) using `shuffle_mutate`, see Section 2.3.3 for details.

While this example provides insight into the inner workings of the library, it is generally not the preferred way to use `evortran` for actual optimization tasks. These are better handled at the level of populations, as discussed in the following.

## 2.2 Populations

Instead of working with individuals directly, it is usually more convenient to work with a set of individuals and perform different steps of a GA on this set as a whole. `evortran` defines for this purpose the derived type `population` both for integer and float individuals. This enables users to apply selection, crossover, mutation, and elitism operations in a modular and efficient way.

### 2.2.1 Integer populations

The declaration of the `population` derived type for integer individuals is given below:

```
370  1 type, public :: population
371  2   integer :: popsize
372  3   integer :: gene_length
373  4   integer :: base_pairs
374  5   type(individual), public, dimension(:), allocatable :: inds
375  6   type(individual), public, dimension(:), allocatable ::
376        selection
```

```
7     type(individual), public, dimension(:), allocatable :: elite
8     type(individual), public, dimension(:), allocatable ::
         offspring
9     integer, public, dimension(:), allocatable ::
         indices_sorted_fitness
10  contains
11    procedure, public :: calc_fitnesses
12    procedure, public :: select_individuals
13    procedure, public :: select_elite
14    procedure, public :: produce_offspring
15    procedure, public :: make_population_from_offspring
16    procedure, public :: get_fittest_individual
17  end type population
18
19  interface population
20    procedure create_population
21  end interface population
```

The components of the integer popolation type are:

- `popsize`: Number of individuals in the population.
- `gene_length`: Length of the gene array for each individual.
- `base_pairs`: Number of possible discrete values each gene can take (default is 2).
- `inds`: The array of actual individual objects in the population.
- `selection`: Array storing selected individuals.
- `elite`: Array for elite individuals that are preserved between generations.
- `offspring`: Array holding new individuals generated from crossover and mutation.
- `indices_sorted_fitness`: Indices of individuals sorted by fitness.

A new integer population object can be initialized using the constructor interface:

```
1  use evortran__populations_integer, only: population
2
3  type(population) :: pop
4
5  pop = population(  &
6    popsize, gene_length, fit_func,  &
7    base_pairs, seed, inds)
```

Here, `popsize` is the number of individuals, `gene_length` the length of the gene array of each individual, `fit_func` the fitness procedure, `base_pairs` (optional) the number of possible gene values, and `seed` (optional) sets the initial values of the genes of all individuals contained in the population. If `seed` is not given, the gene values of the initial populations are randomly generated following a uniorm distribution. Finally, the optional argument `inds` is an array of integer individuals that shall be contained in the initial population. The derived type for integer populations contains the following type-bound procedures:

```
1  subroutine calc_fitnesses(this)
2    class(population), intent(inout) :: this
3  end subroutine
4
5  subroutine select_individuals(  &
6      this, num, mode, tourn_size, wheele_size)
7    class(population), intent(inout) :: this
8    integer, intent(in) :: num
```

```
 9    character(len=*), intent(in) :: mode
10    integer, intent(in), optional :: tourn_size
11    integer, intent(in), optional :: wheele_size
12  end subroutine
13
14  subroutine select_elite(this, num, mode)
15    class(population), intent(inout) :: this
16    integer, intent(in) :: num
17    character(len=*), intent(in) :: mode
18  end subroutine
19
20  subroutine produce_offspring(  &
21      this, num, mating, mating_prob, mutate, mutate_prob,  &
22      mutate_gene_prob, include_elite, uniform_mating_ratio)
23    class(population), intent(inout) :: this
24    integer, intent(in) :: num
25    character(len=*), intent(in) :: mating
26    real(wp), intent(in), optional :: mating_prob
27    character(len=*), intent(in), optional :: mutate
28    real(wp), intent(in), optional :: mutate_prob
29    real(wp), intent(in), optional :: mutate_gene_prob
30    logical, intent(in), optional :: include_elite
31    real(wp), intent(in), optional :: uniform_mating_ratio
32  end subroutine
33
34  function make_population_from_offspring(  &
35      this, fit_func, add_elite) result(pop)
36    class(population), intent(inout) :: this
37    procedure(func\_abstract) :: fit\_func
38    logical, intent(in), optional :: add\_elite
39    type(population) :: pop
40  end function
41
42  function get_fittest_individual(this) result(ind)
43    class(population), intent(in) :: this
44    type(individual) :: ind
45  end function
```

These procedures carry out the following tasks:

- calc_fitnesses: Computes the fitness values of all individuals in the population by calling each individual's calc_fitness routine.

- select_individuals: Selects num individuals from the population using the method specified by mode. Supported selection modes include tournament, rank, and roulette wheel selection, see Table 1. The optional arguments tourn_size and wheele_size control the subset size in tournament and roulette wheel selection, respectively.

- select_elite: Selects an elite of num high-performing individuals that can be preserved and reintroduced after crossover and mutation to ensure that they are maintained in the next generation of the population. So far the only mode supported is called best_fitness which includes the individuals with the best fitness values in the elite. These individuals are stored in the type-bound elite array.

- produce_offspring: Generates an array of offspring individuals with size num by applying the specified crossover, mutation and elitisim routines. These individuals are stored in the type-bound offspring array. The argument mating selects the crossover method, see Table 2, while mating_prob defines the crossover probability. The argument mutate selects the mutation methiod, see Table 3, while muta-

480 tion is applied according to a probability of `mutate_prob`. If the mutation method
481 accepts a gene-wise mutation probability, this probability can be given with the argu-
482 ment `mutate_gene_prob`. If `include_elite` is set to true, the elite individuals are
483 also added to the offspring. The `uniform_mating_ratio` is used if uniform crossover
484 is selected, see the discussion in Section 2.3.3.

485 – `make_population_from_offspring`: Returns an instance of type `population` with
486 the individuals given by the offspring individuals. The fitness function `fit_func` is
487 associated with the new population. If `add_elite` is set to true, the previously selected
488 elite individuals are also added to the new population.

489 – `get_fittest_individual`: Returns the single individual with the highest fitness
490 value in the population.

### 2.2.2 Float populations

492 Similar to integer individuals, `evortran` provides a `population` derived type to manage and
493 evolve collections of float individuals. These float individuals have real-valued genes and are
494 suited for optimization problems defined over continuous search spaces. The population type
495 offers the same high-level routines for selection, crossover, mutation, and elitism as for integer
496 populations. The definition of the *float population* type is shown below:

```fortran
type, public :: population
  integer :: popsize
  integer :: gene_length
  real(wp) :: lower_lim = 0.0e0_wp
  real(wp) :: upper_lim = 1.0e0_wp
  type(individual), public, dimension(:), allocatable :: inds
  type(individual), public, dimension(:), allocatable :: &
      selection
  type(individual), public, dimension(:), allocatable :: elite
  type(individual), public, dimension(:), allocatable :: &
      offspring
  integer, public, dimension(:), allocatable :: &
      indices_sorted_fitness
contains
  procedure, public :: calc_fitnesses
  procedure, public :: select_individuals
  procedure, public :: select_elite
  procedure, public :: produce_offspring
  procedure, public :: make_population_from_offspring
  procedure, public :: get_fittest_individual
  procedure, public :: get_fittest_individuals
end type population

interface population
  procedure create_population
end interface population
```

523 In addition to the components contained also in the integer population type, see the discussion
524 above, the float population type contains the following additional components:

525 – `lower_lim`: a real number defining the lower bound of possible gene values (default
526 0.0).

527 – `upper_lim`: a real number defining the upper bound of possible gene values (default
528 1.0).

These values are respected when generating initial genes, and when applying crossover and mutation operators, where the stochastic nature of these operations might otherwise lead to gene values outside of the interval defined by `lower_lim` and `upper_lim`.

A new float population object can be created using the same constructor interface as for integer populations:

```
use evortran__populations_float, only: population

type(population) :: pop

pop = population(  &
  popsize, gene_length, fit_func,  &
  lower_lim, upper_lim, seed)
```

Here, `lower_lim` and `upper_lim` (optional) define the possible range the genes. All other arguments behave as in the case of integer populations.

Besides the procedures available for integer populations (see previous subsection), the float population type defines one additional procedure called `get_fittest_individuals`. This routine returns the `n` fittest individuals in the population, sorted by fitness in ascending order.

### 2.2.3 Operate on populations

Similar to operating on individuals, as discussed in Section 2.1.3, evortran allows to create and manipulate directly instances of the `population` types. While operating directly on populations is a step up in abstraction compared to manipulating individual objects, it is still not the most convenient to use evortran for real-world optimization tasks. The library provides higher-level functions which allow users to evolve populations over multiple generations and epochs with minimal code, see the discussions in Section 2.4 and Section 2.5. However, working at the population level offers insights into the inner workings of these functions and can be helpful for advanced users who need more fine-grained control.

The following example demonstrates how to manually evolve a population of float individuals over one generation, using built-in selection, crossover, and mutation operations:

```
program operate_on_populations

  use evortran__util_kinds, only : wp
  use evortran__populations_float, only: population
  use evortran__individuals_float, only: individual
  use evortran__prng_rand, only : initialize_rands

  implicit none

  type(population) :: pop
  type(population) :: new_pop
  type(individual) :: best

  call initialize_rands(mode='twister')

  pop = population(100, 10, func)

  call pop%select_individuals(50, 'roulette', wheele_size=7)

  call pop%produce_offspring(  &
```

```
578  21        100, &
579  22        mating='blend',   &
580  23        include_elite=.true.,   &
581  24        mutate='uniform')
582  25
583  26    new_pop = pop%make_population_from_offspring(func)
584  27    best = new_pop%get_fittest_individual()
585  28
586  29    write(*,*) "Genes of best-fit ind.:  ", best%genes
587  30    write(*,*) "Fitness of best-fit ind.:", best%get_fitness()
588  31
589  32 contains
590  33
591  34    pure subroutine func(ind, f)
592  35
593  36        class(individual), intent(in) :: ind
594  37        real(wp), intent(out) :: f
595  38
596  39        f = sum(ind%genes) / real(ind%length, wp)
597  40
598  41    end subroutine func
599  42
600  43 end program operate_on_populations
```

This example demonstrates a typical evolutionary step operating on a float population:

– The random number generator is initialized with the Mersenne Twister method.

– A population pop of 100 float individuals is created, each with 10 genes, and associated with the custom fitness function func.

– A subset of individuals is selected for reproduction using the roulette selection method, which are stored in pop%selection. The optional argument wheele_size is propagated to the roulette wheel selection routine (see Section 2.3.1).

– The procedure produce_offspring is called to generate 100 new offspring individuals, which are stored in pop%offspring. The arguments set blend crossover and uniform mutation as operations to produce the offspring, and the optional argument include_elite is set to true to enforce that the fittest individual in the population is taken over into the set of offspring individuals.

– By calling make_population_from_offspring, a new population new_pop is constructed from the offspring, using the same fitness function.

– The fittest individual of the new population is extracted by calling get_fittest_individual, Finally, the gene values and the fitness value of the best-fit individual is printed.

This example provides a closer look at the operations evortran offers for manipulating populations manually. It gives users full control over each step of the evolutionary process and illustrates how populations are typically evolved internally when higher-level functions are used.

## 2.3   Genetic algorithm operations

A GA progresses through repeated application of four key operations that mimic the principles of natural selection and evolution: selection, crossover, mutation, and elitism. The specific strategies and implementations of these operations greatly influence the effectiveness and efficiency of the GA. The evortran library provides flexible and modular routines for each of these operations, allowing users to tailor the GA to their demands. In the following

| | Kinds | Optional arguments [default values] |
|---|---|---|
| **tournament** | I/F | `tourn_size` [2] |
| **rank** | I/F | - |
| **roulette** | I/F | `wheele_size` [3] |

Table 1: Selection methods implemented in `evortran`. The first column shows the names of the three selection methods in `evortran`. The respective functions are labeled in the second column as I, F, or I/F, indicating the type of genes they operate on: I for individuals with integer-valued genes, F for those with floating-point genes, and I/F for functions that can handle both types. The last column shows optional arguments of each routine and their default values.

subsections, we describe in detail the selection, crossover, mutation, and elitism mechanisms currently implemented in `evortran`, highlighting their underlying principles and typical use cases. We also comment on the advantages and disadvantages of the different methods.

### 2.3.1 Selection routines

The first step in the evolution of a population from one generation to the other is the selection step. During the selections step, samples of the initial population are selected that are allowed to take part in the subsequent crossover step. The selection of the individuals is guided by their fitness values, with the aim of increasing the likelihood of producing fitter offspring. To this end, the selection procedure favors individuals with good fitness values. In Table 1 we show the selection routines that are currently implemented in `evortran`. These routines are implemented generically and can act on both integer and float individuals.

One of the most commonly used selection strategies is *tournament* selection. In this method, a subset of individuals is randomly sampled from the population, and the individual with the highest fitness within this subset is selected to proceed to the crossover step. This process is repeated as many times as needed to construct the mating pool. The tournament size determines the selection pressure. Larger tournament sizes increase the chance of selecting individuals with good fitness, while smaller ones help preserve diversity. In `evortran`, the default tournament size is set to 2, which provides a good balance between selection pressure and population diversity for many applications.

Another selection strategy implemented in `evortran` is *rank* selection. In this approach, individuals in the population are first sorted according to their fitness values, and the selection is based on their rank rather than their absolute fitness. This method helps prevent the premature domination of highly fit individuals and maintains a more balanced selection pressure, particularly in cases where fitness values vary widely. In `evortran`, rank selection is implemented in a straightforward way: the size of the mating pool is specified, and the top-ranking individuals, i.e. those with the best fitness values, are selected until the pool is full. While this ensures that only the most promising individuals participate in the crossover step, it also reduces diversity more aggressively than other selection methods. Compared to tournament selection, rank selection offers more deterministic control over selection pressure but comes with a computational cost. First, it requires the evaluation of the fitness of all individuals contained in the population. Second, a sorting step is required to rank the individuals, which can become expensive for large population sizes. In contrast, tournament selection is more scalable, as it only operates on small subsets of the population and avoids global sorting.

The third selection method currently implemented in `evortran` is a modified version of *roulette wheel* selection, designed to balance randomness and selection pressure in a com-

putationally more efficient way than the usual wheel selection. Unlike standard fitness-proportionate selection over the entire population, the `evortran` implementation first randomly selects a small subset of individuals called the wheel size $k$. The default value of the wheel size in `evortran` is $k = 3$. Let the subset consist of individuals $i = \{1, 2, \ldots, k\}$. Among these, the best fitness value is determined as

$$f_{\text{best}} = \min\{f_1, f_2, \ldots, f_k\}, \tag{1}$$

where $f_i$ is the fitness values of the individual $i$.[3] Then, for each individual in the subset, a weight parameter

$$w_i = \exp\left[\frac{-(f_i - f_{\text{best}})^2}{f_{\text{best}}^2}\right], \quad \Rightarrow \quad 0 < w_i \leq 1, \quad w_{\text{best}} = 1, \tag{2}$$

is computed. The selection probabilities for the individuals are computed using these weights, instead of using the fitness values $f_i$ themselves as in usual roulette wheel selection, via

$$p_i = \frac{w_i}{\sum_{j=1}^{k} w_j}. \tag{3}$$

Then a single individual from the subset is selected based on these probabilities. The whole operation is repeated until the mating pool is full. Compared to tournament selection and rank selection, this approach can better preserve diversity, with the diversity generally increasing with increasing wheel size $k$, while still biasing toward candidates with better fitness in the population. Unlike tournament selection, where only the best-fitted individual in the subset is selected, here all individuals in the subset have a non-zero probability $p_i$ of being chosen. This selection strategy can therefore be viewed as a more relaxed version of tournament selection (with the tournament size equal to the wheel size). Compared to rank selection, it avoids the need to sort the entire population, making it computationally less expensive for large populations. However, the influence of the wheel size and fitness distribution can significantly affect its behavior, requiring carefully choosing the wheel size $k$ for good performance in specific applications.

### 2.3.2 Crossover routines

The crossover step is the process during which new offspring individuals are created whose genes are determined by exchanging and potentially modifying the genes of two or more parent individuals. `evortran` currently offers several crossover routines that are summarized in Table 2.

The *one-point* crossover routine takes two parent individuals as input and returns two offspring individuals. The genes of the offspring individuals are created by selecting a random point along the genes of the parent individuals and exchanging their genes beyond this point. One-point crossing should be used if one wants to maintain some structure in the gene pool since it preserves large segments of the genes of the parent individuals. However, if the optimization problems is high-dimensional and complex, one-point crossing may not introduce sufficient diversity, leading to premature convergence. Moreover, it disrupts the positional dependencies of neighbouring genes, which may be disadvantageous for optimization problems in which certain genes must remain next to each other for meaningful solutions.

The *two-point* crossover routine works in a very similar way as the one-point crossover routine. The only difference is that two-point crossing uses two (instead of one) randomly chosen points along the genes, and only the genes between these two points are swapped in

---

[3]We remind the reader that `evortran` minimizes the fitness function.

| | Kinds | $N_{\text{par}}$ | $N_{\text{off}}$ | Optional arguments [default values] |
|---|---|---|---|---|
| **one-point** | I/F | 2 | 2 | - |
| **two-point** | I/F | 2 | 2 | - |
| **uniform** | I/F | 2 | 2 | `ratio` [0.5] |
| **blend** | F | 2 | 2 | `alpha` [0.5] |
| **sbx** | F | 2 | 2 | `eta_c` [1.0], `p_c` [0.5] |

Table 2: Crossover methods implemented in `evortran`. The first column shows the names of the four crossover methods in `evortran`. The respective functions are labeled in the second column as I, F, or I/F, indicating the type of genes they operate on: I for individuals with integer-valued genes, F for those with floating-point genes, and I/F for functions that can handle both types. The third and fourth columns show $N_{\text{par}}$ and $N_{\text{off}}$ which are the number of parent individuals and offspring individuals, respectively. The last column shows optional arguments of each routine and their default values.

order to create the genes of two offspring individuals. Compared to one-point crossing, the two-point crossing method provides more genetic mixing, while still preserving large segments of genetic material. However, it also disrupts genetic structure by separating neighbouring genes.

The *uniform* crossover routine takes two parent individuals as input, and their genes are inherited randomly by two offspring individuals. Each gene of the offspring individuals is randomly assigned to be taken over from either the first or the second parent individual. Typically, the probability to inherit a gene from parent A or parent B is set to be equal, such that both offspring individuals on average acquire 50% of their genes from one parent and 50% from the other. This is also the default setting in `evortran`. By disrupting the genes of the parent individuals at various positions along the genes, uniform crossing leads to excellent diversity in the gene pool. This makes it well suited for highly complex optimization tasks because it is less affected by premature convergence compared to one- and two-point crossing. However, the highly disruptive behaviour of uniform crossing with 50% exchange probability does not maintain larger blocks of genes, which may cause slow convergence. In such cases, it can be useful to lower the degree of diversity. This can be achieved in `evortran` by changing the ratio of the genes assigned from either parent A or parent B via the optional argument `ratio`. The value given for this argument is the probability for each gene of parent A to be inherited by the first offspring individual, and the second offspring individual inherits the corresponding gene from parent B. Accordingly, the probability for the second offspring individual to inherit a gene from parent A is one minus the value given for `ratio`.

The *blend* crossover routine (also called BLX-$\alpha$) is a crossing procedure that can only be applied to individuals with genes consisting of continuous numbers and not integers. This method generates two offspring individuals by selecting new gene values within an extended range between the gene values of two parent individuals. With $a_i$ being the genes from parent A and $b_i$ the genes from parent B, the genes $c_i$ and $d_i$ of the offspring individuals C and D are given, respectively, by randomly and uniformly selecting a number in the ranges

$$c_i \in [a_i - \alpha(b_i - a_i), b_i + \alpha(b_i - a_i)] \quad \text{and} \quad d_i \in [b_i - \alpha(a_i - b_i), a_i + \alpha(a_i - b_i)], \quad (4)$$

for all $i = 1, \ldots, N_g$. If the above operation leads to gene values $c_i$ and/or $d_i$ that fall outside of the allowed range of the genes of the individuals, their values are clipped to the nearest valid value, i.e. either the lower or the upper limit. The parameter $\alpha$ controls the range beyond

which the genes of the offspring individuals can extend beyond the ones of the parent individuals. In many applications it is set to $\alpha = 0.5$, and this is also the default setting in `evortran`. The presence of this parameter allows tunable exploration of the solution space, in contrast to the other implemented crossover strategies. However, in many cases a good choice for $\alpha$ can only be obtained on heuristic grounds by "trial and error". Blend crossing is fundamentally different from the other crossover methods due to its continuous nature, creating gene values for the offspring individuals that are similar but not identical to the genes contained in the parent individuals. This feature makes blend crossing often more suitable for continuous optimization problems in which small variations can give rise to significant improvements in the fitness of an individual.

Finally, *simulated binary crossover* (usually abbreviated as SBX) is a widely used operator in GAs for individuals with floating-point genes. Inspired by one-point crossover in binary-coded GAs, SBX simulates a similar effect in continuous search spaces. For each corresponding gene pair $(a_i, b_i)$ from two parent individuals, a random number $0 \leq u_i \leq 1$ is drawn from a uniform distribution. From the random variables $u_i$, spread factor $\beta_{q,i}$ are computed using the distribution index $\eta_c$ via

$$\beta_{q,i}(u) = \begin{cases} (2u_i)^{1/(\eta_c+1)}, & \text{if } u_i \leq 0.5 \\ \left(\frac{1}{2(1-u_i)}\right)^{1/(\eta_c+1)}, & \text{if } u_i > 0.5 \end{cases} . \tag{5}$$

Then the corresponding pair of offspring genes $c_i$ and $d_i$ are computed as

$$\begin{aligned} c_i &= 0.5\left[(1+\beta_{q,i})a_i + (1-\beta_{q,i})b_i\right], \\ d_i &= 0.5\left[(1-\beta_{q,i})a_i + (1+\beta_{q,i})b_i\right]. \end{aligned} \tag{6}$$

This operation is applied independently to each gene pair $(i = 1, \ldots, N_g)$ with a certain gene-wise crossover probability $p_c$. For each gene pair, the SBX operation described above is applied to generate new gene values $c_i$ and $d_i$, and otherwise (with probability $1 - p_c$) the two offspring individuals simply inherit the genes $a_i$ or $b_i$, respectively. As for blend crossover, if the above operation leads to gene values $c_i$ and/or $d_i$ that fall outside of the allowed range of the genes of the individuals, their values are clipped to the lower or upper limit of the allowed range. In `evortran` the probability $p_c$ has the default value $p_c = 0.5$, but its value can be changed by the user. The parameter $\eta_c$ controls the distribution of offspring genes. It has the default value $\eta_c = 1.0$ in `evortran`, but can also be modified by the user. Smaller values of $\eta_c$ promote broader exploration, allowing the genes of the offspring to deviate more significantly from the genes of the parent individuals. On the contrary, larger values of $\eta_c$ lead to offspring genes that are centered more closely to the genes of the parent individuals, which typically yields faster convergence but less exploration of the solution space. Compared to the blend crossover method discussed above, SBX offers a more adjustable and often more efficient trade-off between exploration (searching broadly across the solution space) and exploitation (refining and improving solutions near individuals with high fitness). However, blend crossover requires fewer computation steps and only has a single meta parameter $\alpha$. Therefore, while SBX is often more effective for detailed local optimization in promising regions of the solution space, blend crossover is simpler and better suited for broad, uniform searches across the entire solution space where less precision is initially required.

### 2.3.3  Mutation routines

We discuss here the mutation routines that `evortran` provides. Mutation introduces diversity by randomly modifying gene values, helping the GA escape local optima and explore the search space more comprehensively. The available mutation routines are summarized in Table 3. Each

| | Kinds | Optional arguments [default values] |
|---|---|---|
| **uniform** | I/F | `prob` [1/ind%length] |
| **shuffle** | I/F | `prob` [1/ind%length] |
| **gaussian** | F | `prob` [1/ind%length], `sigma` [1.0] |

Table 3: Mutation methods implemented in `evortran`. The first column shows the names of the three mutation methods in `evortran`. The respective functions are labeled in the second column as I, F, or I/F, indicating the type of genes they operate on: I for individuals with integer-valued genes, F for those with floating-point genes, and I/F for functions that can handle both types. The third column shows optional arguments of each routine and their default values.

mutation routine operates in-place on a given individual, i.e. it directly modifies the genes of the individual given as input. After mutation, the fitness of the individual is invalidated by a call to its internal `reset_fitness()` procedure. This ensures that the next time the fitness is accessed, it is correctly recalculated using the mutated genes.

The *uniform* mutation routine can be applied to both integer and float individuals. Each gene has an independent probability `prob` (defaulting to 1/ind%length, where ind%length is the length of the genes of the individual) of being replaced by a new value sampled uniformly from the full range of allowable gene values. A main advantage of uniform mutation is that it introduces new gene values into the population over the whole possible range. It is therefore particularly useful in early stages of the GA, where the solution space should be covered broadly without converging prematurely into a local minimum. However, uniform mutation might be too disruptive at final stages of the GA, when there are solutions that are already close to the desired (global) optimum, since its coarse nature may modify gene values far away from suitable values instead of refining them.

Also compatible with both integer and float genes is the *shuffle* mutation routine. With the specified probability `prob` per gene (default 1/ind%length), each gene switches the place in the array of genes with another randomly selected gene. In contrast to the uniform mutation discussed above, shuffle mutation only modifies the order of the gene values, but it does not introduce new gene values which were not present in the population before. In a GA with a contineous solution space, it should therefore be combined with a crossover method that produces new gene values (e.g. blend or sbx crossover) instead of only transferring gene values from parent to offspring individuals. Since shuffle mutation keeps the gene values, only altering their positions, the mutation is often less disruptive than uniform mutation and can maintain some beneficial building blocks. This makes it more suitable for permutation-based optimization problems, such as scheduling, path finding or ordering tasks. However, it might lack sufficient exploration power, and it is usually not suitable for problems in which the positions of the gene values within the sequence are highly structured for good solutions to the problem at hand.

Available only for float individuals, `evortran` offers a third mutation routine called *gaussian* mutation. Here, each gene has a chance `prob` to be replaced by a normally distributed random number with mean given by the gene value to be replaced and standard deviation `sigma`. This introduces smooth, local variations suitable for continuous optimization problems. The default value of `sigma` is set to 1.0 because the default range of allowed gene values is from 0.0 to 1.0, see the discussion in Section 2.2.2. If the actual lower and upper limits of possible gene values differ from this default, it is advisable to adjust `sigma` accordingly to ensure that the spread of mutation remains appropriate relative to the full range of

possible gene values. Since the mean of the normal distribution is the original gene value itself, Gaussian mutation tends to produce values close to the original, making this method less disruptive than uniform mutation, where the gene value is replaced with a completely random value from the entire allowed range. The argument `sigma` controls the spread of the distribution. A larger value increases variability and allows for more exploratory mutations, whereas a smaller value makes mutations more conservative, favoring fine-tuning. Gaussian mutation is especially useful if the GA has already determined solutions near a good solution in order to explore the local neighborhood efficiently and fine-tune the final solution.

### 2.3.4 Elitism

Elitism is a common strategy in GAs that ensures the preservation of the best-performing individuals across generations. Its main purpose is to prevent the loss of the best solutions at intermediate stages of the GA due to the stochastic nature of selection, crossover, and mutation. By retaining a subset of the best individuals, elitism promotes convergence and often improves the stability and performance of the GA. However, if the number of elite individuals is too large, it might lead to premature convergence and poor coverage of the solution space.

In `evortran`, elitism is currently implemented in the most simple form. Users can specify a number of elite individuals to be preserved during reproduction. The individuals in the population that have the lowest fitness values are directly appended to the offspring. It is worth noting that this elitism procedure can become computationally significant for large population sizes or when fitness evaluations are costly, as it requires computing the fitness values of all individuals in the population and sorting them based on those values.

## 2.4 Evolution of a population

We have discussed above how to operate on individuals and populations directly. In principle, a user can defined their own GA using these functionalities to their specific needs. However, with a certain optimization task at hand, it is usually more practical to call a function which takes the fitness function as input and performs a whole evolution of a population in order to optimize the fitness function. To this end, `evortran` provides the function `evolve_population`, which corresponds to the next level of abstraction and encapsulates the entire process of evolving a population through the different stages of a GA. This function iteratively applies the four fundamental operations (selection, crossover, mutation, and elitism) until either a maximum number of generations has been reached or a predefined fitness target has been achieved.

This is one of the two interfaces that are most likely to be used by users of `evortran` (with the other one being the `evolve_migration` function discussed in Section 2.5). It allows users to easily apply GAs to their problems with minimal boilerplate code. The only requirement is to implement a user-defined fitness function conforming to the abstract interface described in Section 2.1.1. Once the fitness function is defined, optimization is as simple as calling `evolve_population` with the desired parameters.

The function `evolve_population` is highly flexible, offering a large number of optional arguments that enable customization of the GA, such as gene initialization, selection and mating strategies, mutation behavior, elitism, and output tracking. The whole function declaration is as follow:

```
1 function evolve_population(  &
2   pop_size,  &
3   gene_length,  &
4   fit_func,  &
5   lower_lim,  &
6   upper_lim,  &
```

```
855     7     max_generations,  &
856     8     fitness_target,  &
857     9     verbose,  &
858    10     gene_seed,  &
859    11     add_ind,  &
860    12     selection,  &
861    13     selection_size,  &
862    14     tourn_size,  &
863    15     wheele_size,  &
864    16     elitism,  &
865    17     elite_size,  &
866    18     mating,  &
867    19     offspring_size,  &
868    20     offspring_include_elite,  &
869    21     mating_prob,  &
870    22     blend_alpha,  &
871    23     sbx_eta_c,  &
872    24     sbx_p_c,  &
873    25     uniform_mating_ratio,  &
874    26     mutate,  &
875    27     mutate_prob,  &
876    28     mutate_gene_prob,  &
877    29     mutate_gaussian_sigma,  &
878    30     fittest_inds_from_gen,  &
879    31     pops_from_gen,  &
880    32     init_pop,  &
881    33     final_pop) result(best_ind)
882    34
883    35     integer, intent(in) :: pop_size
884    36     integer, intent(in) :: gene_length
885    37     procedure(func_abstract) :: fit_func
886    38     real(wp), intent(in), optional :: lower_lim
887    39     real(wp), intent(in), optional :: upper_lim
888    40     integer, intent(in), optional :: max_generations
889    41     real(wp), intent(in), optional :: fitness_target
890    42     logical, intent(in), optional :: verbose
891    43     real(wp), intent(in), optional :: gene_seed
892    44     type(individual), intent(in), optional :: add_ind
893    45     character(len=*), intent(in), optional :: selection
894    46     integer, intent(in), optional :: selection_size
895    47     integer, intent(in), optional :: tourn_size
896    48     integer, intent(in), optional :: wheele_size
897    49     character(len=*), intent(in), optional :: elitism
898    50     integer, intent(in), optional :: elite_size
899    51     character(len=*), intent(in), optional :: mating
900    52     integer, intent(in), optional :: offspring_size
901    53     logical, intent(in), optional :: offspring_include_elite
902    54     real(wp), intent(in), optional :: mating_prob
903    55     real(wp), intent(in), optional :: blend_alpha
904    56     real(wp), intent(in), optional :: sbx_eta_c
905    57     real(wp), intent(in), optional :: sbx_p_c
906    58     real(wp), intent(in), optional :: uniform_mating_ratio
907    59     character(len=*), intent(in), optional :: mutate
908    60     real(wp), intent(in), optional :: mutate_prob
909    61     real(wp), intent(in), optional :: mutate_gene_prob
910    62     real(wp), intent(in), optional :: mutate_gaussian_sigma
911    63     type(individual), intent(out), dimension(:),  &
```

```
912   64        allocatable, optional :: fittest_inds_from_gen
913   65     type(population), intent(out), dimension(:),  &
914   66        allocatable, optional :: pops_from_gen
915   67     type(population), intent(in), optional :: init_pop
916   68     type(population), intent(out), optional :: final_pop
917   69     type(individual) :: best_ind
```

The arguments to this function are:

- `pop_size`: Number of individuals in the population.
- `gene_length`: Number of gene values in each individual. Should be equal to the dimension of the fitness function.
- `fit_func`: User-defined fitness function following the abstract interface `func_abstract` given in Section 2.1.1.
- `lower_lim`, `upper_lim`: Limits of the range of possible gene values (defaults are 0.0 and 1.0). These optional arguments can only be set together.
- `max_generations`: Maximum number of generations to run (default is `pop_size`).
- `fitness_target`: Optimization stops early if this fitness value is reached.
- `verbose`: If true, prints evolving summary output during execution.
- `gene_seed`: Seed used to initialize gene values of all individuals in the initial population.
- `add_ind`: An individual to insert into the initial population.
- `selection`: The selection method, currently possible values are `tournament`, `rank` or `roulette`, see Section 2.3.1 (default is `tournament`).
- `selection_size`: Number of individuals to select (default is `pop_size`).
- `tourn_size`: Given as optional argument `tourn_size` when tournament selection is used, see Section 2.3.1.
- `wheele_size`: Given as optional argument `wheele_size` if roulette wheele selection is used, see Section 2.3.1.
- `elitism`: Elitism mode, where currently only the default option `best_fitness` is supported, see Section 2.3.4.
- `elite_size`: Number of elite individuals to keep (default is 1).
- `mating`: The crossover method, currently possible values are `one-point`, `two-point`, `uniform` for both integer and float populations, and additionally `blend`, and `sbx` only for float populations (default is `one-point`).
- `offspring_size`: Number of offspring individuals to generate (default is `pop_size`).
- `offspring_include_elite`: Whether to apply elitism, see Section 2.3.4 (default is `.true.`).
- `mating_prob`: The mating probability, see Section 2.3.2 (default is 0.95).
- `blend_alpha`: Parameter `alpha` for blend crossover, see Section 2.3.2 (default is 0.5).
- `sbx_eta_c`: Parameter `eta_c` for simulated binary crossover, see Section 2.3.2 (default is 1.0).
- `sbx_p_c`: Parameter `p_c` for simulated binary crossover, see Section 2.3.2 (default is 0.9).
- `uniform_mating_ratio`: Parameter `ratio` for uniform crossover, see Section 2.3.2 (default is 0.5).
- `mutate`: The mutation method, currently possible values are `uniform` and `shuffle` for both integer and float populations, and additionally `gaussian` only for float populations, see Section 2.3.3. (default is `uniform`).
- `mutate_prob`: Mutation probability (default is 0.1).

- mutate_gene_prob: Probability of mutating each gene if individual is mutated (default is 0.1).
- mutate_gaussian_sigma: Parameter sigma for gaussian mutation, see Section 2.3.3 (default is 1.0).
- fittest_inds_from_gen: Stores the fittest individual from each generation in an array of length max_generations if the final achieved fitness value is larger than fitness_target. If the GA terminates because a fitness below fitness_target has been achieved, the length of the array will be equal to the number of generations that were created up to this point.
- pops_from_gen: If present, stores the entire population at each generation. The user should ensure that sufficient memory is available.
- init_pop: Initial population to start from instead of ranodmly generating one at the start of the GA.
- final_pop: If present, stores the entire population after the GA has terminated.

The return object best_ind of the function is the individual with the best fitness that was found during the process. A minimal call of evolve_population for a continuous optimization problem (without specifying the optional arguments) and printing out the minimal value of the fitness function that was found, looks like this:

```fortran
1 use evortran__individuals_float, only : individual
2 use evortran__evolutions_float, only : evolve_population
3
4 type(individual) :: best_ind
5
6 best_ind = evolve_population(1000, 20, func)
7 write(*,*) best_ind%get_fitness()
```

Here the function func with 20 arguments is minimized using a GA with a population size of 1000.

In some applications, it may be beneficial to adapt the behavior of the GA over time. For example, starting with more exploratory operations such as broad or disruptive crossover and mutation strategies, and gradually transitioning to more fine-grained, exploitative methods as the search progresses. evortran supports this type of staged evolution by allowing multiple chained calls to evolve_population, each with different parameters. Between calls, one can transfer either only the best individual using the add_ind argument or reuse the entire final population from one stage as the initial population for the next via the final_pop and init_pop arguments. This modular design enables the construction of highly flexible and dynamic GAs that evolve their strategies over the course of the optimization process.

## 2.5  Migration of populations

In addition to the function evolve_population, evortran provides the function evolve_migration, which operates on multiple populations simultaneously. This represents the highest abstraction level in the user-interface of evortran. Specifically, the function evolve_migration evolves multiple populations independently over a series of *epochs*, where each epoch consists of a number of generations. After each epoch, individuals may migrate between populations. This form of co-evolution can help to maintain genetic diversity by reducing premature convergence, explore multiple areas of the search space concurrently, and to increase robustness by yielding several good-fit individuals (each corresponding to local minima of the fitness function and potentially sufficiently good solutions to the problem

at hand). Moreover, this approach naturally allows for performant and straightforward paral-
lelization, as each population can be evolved independently before migration steps are applied,
as is discussed in more detail in Section 2.6.

As in evolve_population, users can configure the GA, including selection, crossover,
mutation, elitism, and stopping criteria. Additional parameters control the migration behavior.
The interface of the function is:

```fortran
function evolve_migration(  &
  pop_number,  &
  epoches,  &
  pop_size,  &
  gene_length,  &
  fit_func,  &
  migration,  &
  migration_size,  &
  migration_order,  &
  lower_lim,  &
  upper_lim,  &
  max_generations,  &
  fitness_target,  &
  verbose,  &
  gene_seed,  &
  selection,  &
  selection_size,  &
  tourn_size,  &
  wheele_size,  &
  elitism,  &
  elite_size,  &
  mating,  &
  mating_prob,  &
  blend_alpha,  &
  sbx_eta_c,  &
  sbx_p_c,  &
  uniform_mating_ratio,  &
  offspring_size,  &
  offspring_include_elite,  &
  mutate,  &
  mutate_prob,  &
  mutate_gene_prob,  &
  mutate_gaussian_sigma,  &
  add_ind,  &
  fittest_inds_final_pops  &
  ) result(best_ind)

  integer, intent(in) :: pop_number
  integer, intent(in) :: epoches
  integer, intent(in) :: pop_size
  integer, intent(in) :: gene_length
  procedure(func_abstract) :: fit_func
  character(len=*), intent(in), optional :: migration
  integer, intent(in), optional :: migration_size
  character(len=*), intent(in), optional :: migration_order
  real(wp), intent(in), optional :: lower_lim
  real(wp), intent(in), optional :: upper_lim
  integer, intent(in), optional :: max_generations
  real(wp), intent(in), optional :: fitness_target
```

```
1061   50      logical, intent(in), optional :: verbose
1062   51      real(wp), intent(in), optional :: gene_seed
1063   52      character(len=*), intent(in), optional :: selection
1064   53      integer, intent(in), optional :: selection_size
1065   54      integer, intent(in), optional :: tourn_size
1066   55      integer, intent(in), optional :: wheele_size
1067   56      character(len=*), intent(in), optional :: elitism
1068   57      integer, intent(in), optional :: elite_size
1069   58      character(len=*), intent(in), optional :: mating
1070   59      real(wp), intent(in), optional :: mating_prob
1071   60      real(wp), intent(in), optional :: blend_alpha
1072   61      real(wp), intent(in), optional :: sbx_eta_c
1073   62      real(wp), intent(in), optional :: sbx_p_c
1074   63      real(wp), intent(in), optional :: uniform_mating_ratio
1075   64      integer, intent(in), optional :: offspring_size
1076   65      logical, intent(in), optional :: offspring_include_elite
1077   66      character(len=*), intent(in), optional :: mutate
1078   67      real(wp), intent(in), optional :: mutate_prob
1079   68      real(wp), intent(in), optional :: mutate_gene_prob
1080   69      real(wp), intent(in), optional :: mutate_gaussian_sigma
1081   70      type(individual), intent(in), optional :: add_ind
1082   71      type(individual), intent(out), dimension(:), allocatable,  &
1083   72          optional :: fittest_inds_final_pops
1084   73      type(individual) :: best_ind
1085   74
1086   75  end function evolve_migration
```

This function contains the following arguments in addition to the ones of the function `evolve_population` that were already discussed in Section 2.4:

– `pop_number`: The number of populations evolved in parallel.

– `epoches`: The number of epochs.

– `migration`: The type of migration that is carried out after each epoche. The currently only implemented option is `rank`, where the best-fit individuals are selected to migrate to other populations.

– `migration_size`: The number of individuals to migrate from each population per epoch.

– `migration_order`: The migration order that determines the population to which individuals migrate from one to the other. Assuming that the populations are labeled by $i = 1,\ldots,N$, with $N$ being equal to `migration_size`, possible options are (default is `random`):

LR: The individuals from population $i$ migrate to population $i + 1$, except the individuals from population $i = N$ migrate to population $i = 0$.

RL: The individuals from population $i$ migrate to population $i - 1$, except the individuals from population $i = 1$ migrate to population $i = N$.

`random`: For each population, after each epoch, the target population is randomly selected. In this case it is possible that individuals from different populations migrate to the same target population.

– `fittest_inds_final_pops`: Stores the best-fit individuals from each population upon completion.

The return value `best_ind` is the best-fit individual found across all populations after the GA completes, either by reaching the fitness target or by reaching the maximum number of epoches.

## 2.6  Parallelization using OpenMP

GAs are particularly well-suited for parallelization due to their inherently population-based structure. Since individuals in a population evolve mostly independently during fitness evaluation, selection, and crossover, many of the operations in a GA can be efficiently distributed across multiple CPU cores.[4] This allows significant speed-up when tackling computationally intensive optimization problems if a sufficient number of CPU cores are available.

evortran supports parallel execution on multi-core CPUs through the OpenMP application programming interface. OpenMP is a widely adopted standard for shared-memory parallel programming in Fortran, C, and C++, which enables easy parallelization of loops and code sections through compiler directives that are supported by many Fortran compilers. evortran uses fpm as build system, where OpenMP support is activated by adding it as a meta-package in the fpm.toml configuration file, as discussed in more detail in Section 3.1.2.

The way parallelization is applied in evortran depends on the abstraction level used by the user. When operating at the level of individual populations (e.g. via evolve_population), evortran parallelizes internal loops over individuals. For instance, fitness evaluation, selection, and offspring generation can each be executed in parallel across individuals within a population. The user does not need to modify the runtime call to benefit from this. At the higher level of abstraction using evolve_migration, evortran evolves multiple populations in parallel. In this case, the outer loop over populations is parallelized, which is often more efficient than parallelizing individual operations, because fewer threads need to be launched and managed, thus reducing overhead. When selecting how many CPU cores to use, it can be beneficial to consider these differences: With evolve_population, performance may improve if the number of individuals in the population is a multiple of the available CPU cores. With evolve_migration, best efficiency is typically achieved when the number of populations matches or is a multiple of the number of available cores.

To ensure correctness during parallel execution, the user must make sure that the custom fitness function is thread-safe, i.e. it should not depend on or modify shared state in a non-synchronized manner. All internal components of evortran that are involved in parallel execution, such as random number generation and sorting routines, are implemented to be thread-safe, as will be discussed in more detail in Section 2.7. The number of threads used during execution can be controlled by the user via the OMP_NUM_THREADS environment variable at compile time (see also the discussion in Section 3.1). While this must be set before compilation, this is not a major limitation in practice, since evortran can be compiled in just a few seconds.[5]

The current implementation of evortran focuses on shared-memory parallelization via OpenMP directives. This prioritises portability and simplicity for users working on typical desktop and single-node workstations, where many scientific users conduct their computations and optimization tasks. Here, OpenMP provides efficient scaling across multi-core CPUs, while being widely supported by modern Fortran compilers. It should be noted, however, that this approach is inherently limited to single-node systems. For large-scale applications (for instance, the reconstruction of gravitational wave spectra from LISA data, see Section 4.2.2), future extensions of evortran could benefit from distributed-memory parallelization using MPI, or from offloading computationally intensive task to GPUs through OpenMP target dire-

---

[4]The suitability of GAs for parallelization is reduced in special cases where operations (such as the fitness function or the selection procedure) depend not only on an individual's genes, but also collectively on the genes of multiple or all individuals in the population. Such global dependencies introduce synchronization constraints that make parallel execution less efficient or even impossible. These cases are not considered in this paper. In evortran, users can disable parallelization if such dependencies are required for a specific application.

[5]The number of threads can be modified at runtime using the omp_lib module, e.g. by calling the subroutine omp_set_num_threads. However, it cannot be set to a value that exceeds the maximum number of threads specified at compile time by the OMP_NUM_THREADS environment variable.

cives or the `CUDA` framework developed by NVIDIA, which provides the `nvfortran` compiler. The efficiency of GPU acceleration depends strongly on the form of the fitness function, where only computationally intensive and side-effect-free fitness functions are likely to benefit significantly from GPU offloading. In some applications, the evaluation of the fitnesses of all individuals contained in a population may be suitable for offloading as a whole to the GPU, for instance, if the computation across the individuals can be expressed in terms of matrix operations. Support for GPU-accelerated GAs in `evortran` is planned for future releases.

## 2.7   Core utilities and numerical tools

This section discusses essential building blocks that underpin the operation of GAs, and their implementation in `evortran`. These core utilities include the pseudo-random number generators (PRNGs), which are fundamental to introduce randomness into the algorithm, and sorting routines, which are crucial for implementing selection mechanisms based on fitness values of the individuals. In addition to these core components, `evortran` also includes a numerical interpolation module that can be valuable for regression or surrogate modeling tasks.

### 2.7.1   Pseudo-random number generation

Pseudorandom number generation is a central component in GAs, as it governs stochastic processes such as initialization, mutation, selection, and crossover. In parallelized GAs, the PRNG desirably is thread-safe to avoid race conditions and ensure reproducibility. `evortran` provides two options for generating random numbers.

The default and recommended PRNG in `evortran` is a thread-safe implementation of the *Mersenne Twister algorithm* [53, 54] (specifically, the 64-bit variant MT19937-64). The implementation contained in `evortran` is adapted from Ref. [55]. The adaptation integrates it with `evortran`'s internal real kind working precision (`wp`) and modifies it for parallel execution using `OpenMP`. Each `OpenMP` thread is assigned its own instance of the PRNG and initialized with a unique seed at the start of the program. This design ensures that random number generation is thread-safe, and that the results of the GA are deterministic and reproducible in parallel execution. Moreover, the performance is competitive compared to Fortran intrinsic PRNGs, making it suitable for large-scale stochastic sampling.[6]

Alternatively, `evortran` offers the option to generate random integer and float numbers based on the Fortran intrinsic function `random_number()`. While slightly faster in sequential execution, this option is not thread-safe, since concurrent calls by multiple threads result in race conditions. As a consequence, results are no longer deterministic when executed in parallel. This option is useful when strict reproducibility is not required and performance is paramount. However, typically the thread-save implementation of the Mersenne Twister is strongly recommended.

Before using any functionality of the `evortran` library, the PRNG must be explicitly initialized. This is done by calling the subroutine `initialize_rands`. For instance, to use the Mersenne Twister PRNG, one has to call:

```
1 use evortran__prng_rand , only : initialize_rands
2
3 call initialize_rands(mode='twister', seed=0)
```

The second argument `seed` is an optional integer argument that sets an initial seed value for the Mersenne Twister PRNG (the default value is 0). Alternatively, to use the Fortran intrinsic PRNG function, one has to call:

---

[6]The Merseene Twister algorithm is also used in the modernizded `fpm` version of `Pikaia` [50], whereas the original version uses the "Minimal standard" PRNG [56].

```
1199   1 call initialize_rands ( mode = 'intrinsic ')
```

Using this mode, the optional `seed` argument is ignored. If this routine is not called, the program will result in a runtime error if `evortran` was compiled in DEBUG mode, or in undefined behavior if compiled without it. The compilation flags, including enabling or disabling DEBUG mode, are discussed in detail in Section 3.1.

### 2.7.2  Sorting methods

Sorting individuals based on their fitness values is a central operation in GAs, especially for ranking-based selection methods and elitism. In `evortran`, sorting is used internally to determine the ordering of individuals according to their fitness, and the library provides two efficient and thread-safe sorting algorithms that are designed to be called in parallel using `OpenMP`.

The first sorting option is a parallel *merge sort* algorithm, which is the default sorting method in `evortran`. It is based on the merge sort implementation from the `orderpack` library [57]. The version used in `evortran` has been adapted from a parallel implementation provided in a public github repository [58], and modified for compatibility with the real working precision kind and data structures used in the library. The second available option is *quick sort*, implemented using a recursive algorithm adapted from a publicly available Fortran quicksort implementation [59]. Like the merge sort routine, it is thread-safe, allowing concurrent execution in `OpenMP`-parallelized regions of user code.

Users can explicitly select the sorting algorithm used by `evortran` by calling the subroutine `set_rank_method` with the desired mode:

```
1220   1 use evortran__sorting_ranking , only : set_rank_method
1221   2
1222   3 ! Select merge sort
1223   4 call set_rank_method ( mode = 'merge ')
1224   5
1225   6 ! Or select quicksort
1226   7 call set_rank_method ( mode = 'quick ')
```

In typical use cases within `evortran`, both sorting methods were found to exhibit similar runtime performance. However, for specific use cases of the GA the user may wish to specifically chose one of the two options. Merge sort guarantees $\mathcal{O}(n \log n)$ time complexity in all cases and is stable (preserves relative ordering of equal elements), which can be advantageous in some applications. Quick sort may be faster in practice due to lower constant factors, but its performance can degrade to $\mathcal{O}(n^2)$ in the worst case, although this is rare with randomized pivoting. Because of its more predictable performance and stability, merge sort is the default sorting method in `evortran`.

### 2.7.3  Function interpolation

While not a core component of GAs, `evortran` includes a utility for performing cubic spline interpolation, which can be valuable in a range of optimization contexts where functions are approximated via discretization and a finite number of points. This functionality is implemented through pure Fortran procedures, making the routines thread-safe and thus suited for parallel execution. In GA applications, spline interpolation can be useful when the fitness function is based on experimental or computational data sampled at discrete values. Instead of restricting the optimization to those discrete points, a spline-based interpolant can provide a smooth and continuous approximation of the function, allowing for evaluations at arbitrary points in

the parameter space. Additionally, spline interpolation may be helpful in hybrid optimization strategies where gradient-free methods like GA are combined with smooth approximations to facilitate local refinement or sensitivity analysis, by enabling smooth representations of otherwise discrete data.

The following code snippet demonstrates how to use the cubic spline interpolation feature to approximate the value of a function between a set of known data points:

```fortran
1  use evortran__util_kinds, only : wp
2  use evortran__util_interp_spline, only : spline_construct
3  use evortran__util_interp_spline, only : spline_getval
4
5  integer, paramter :: n = 4 ! Number of known points
6  real(wp) :: x(n)           ! Array of x-values
7  real(wp) :: y(n)           ! Corresponding y-values = f(x)
8  real(wp) :: b(n)           ! Spline coefficient array
9  real(wp) :: c(n)           ! Spline coefficient array
10 real(wp) :: d(n)           ! Spline coefficient array
11 real(wp) :: xi             ! Point at which to evaluate the
      spline
12 real(wp) :: yi             ! Interpolated value at xi
13
14 ! Define interpolation points: f(x) = x^2
15 x = [1.0e0_wp, 2.0e0_wp, 3.0e0_wp, 4.0e0_wp]
16 y = [1.0e0_wp, 4.0e0_wp, 9.0e0_wp, 16.0e0_wp]
17
18 xi = 2.5e0_wp ! Target x-value for interpolation
19
20 ! Compute spline coefficents based on input data
21 call spline_construct(x, y, b, c, d, n)
22
23 ! Evaluate the spline at xi = 2.5
24 yi = spline_getval(xi, x, y, b, c, d, n)
```

Specifically, this example interpolates the function $f(x) = x^2$ using four data points. After constructing the spline coefficients using `spline_construct`, it evaluates the interpolated value at a midpoint $x_i = 2.5$ using `spline_getval`.

# 3 User instructions

In this section we give practical guidance on how to get started with `evortran`. In Section 3.1, we explain how to install the library and its dependencies. In Section 3.2 we introduce the key functionalities of `evortran` and demonstrates how to develop a custom GA.

## 3.1 Installation

To use `evortran`, a few prerequisites must be installed on the system. These are standard tools in modern Fortran development and are typically easy to set up on most platforms.

### 3.1.1 Prerequisities

`evortran` is written in modern Fortran using an object-oriented user interface. The recommended compiler to build `evortran` is the GNU Fortran compiler `gfortran`.[7] On Ubuntu or

---

[7]While other modern Fortran compilers exist, they are either not yet compatible with key features used in `evortran` or lack integration with the Fortran Package Manager. `lfortran`, currently in alpha stage and under

Debian-based systems `gfortran` can be installed with:

```
1 sudo apt update
2 sudo apt install gfortran
```

`evortran` was developed and tested using the versions 11, 12 and 13 of `gfortran`.

For the build process and for managing dependencies, `evortran` uses the Fortran Package Manager `fpm`[51]. There are various ways to install `fpm`, for instance, using package managers like `pip`,

```
1 pip install fpm
```

or conda,

```
1 conda config --add channels conda-forge
2 conda create -n fpm fpm
3 conda activate fpm
```

One can also install `fpm` by downloading a binary for the latest stable release which are available for Windows, MacOS, and Linux, or build fpm from source, see Ref. [60].

### 3.1.2  Building evortran

With `gfortran` and `fpm` installed, one can build `evortran`. The first step is to clone the repository and to navitage to the `evortran` directory:

```
1 git clone https://gitlab.com/thomas.biekoetter/evortran
2 cd evortran
```

Then one has to source `fpm` environment variables which define compiler flags and the number of `OpenMP` threads that should be used. This can be done by sourcing one of two provided scripts. Running

```
1 source exports_debug.sh
```

sets compiler flags appropriate for debugging, including options that enable stricter compile-time checks. More importantly, it activates a wide range of runtime argument checks specific to `evortran` that help ensure correct usage of the library. These checks are only available in debug mode and are strongly recommended while setting up or developing the GA. For performance runs, one should instead run

```
1 source exports_run.sh
```

which disables these runtime checks through preprocessor directives, activates compiler optimization (`-O3 -march=native`) and omits array out-of-bounds checking. This results in significantly faster execution but should only be used once the implementation is known to be correct. In both cases, the number of threads used for parallel regions is controlled by the `OMP_NUM_THREADS` environment variable, which is set to 8 by default in both files but can be modified as needed.

---

active development, still lacks full support for `OpenMP`. Moreover, LLVM `flang` is not yet widely supported by `fpm` (as of version 0.12.0).

As was already mentioned in Section 2.1.2, by default the real kind working precision $wp$ corresponds to double precision, as defined in the module `evortran_util_kinds` using `selected_real_kind(15, 307)`. This corresponds to a precision of at least 15 significant digits. The precision can be upgraded to quadruple precision, using `selected_real_kind(30, 4931)`, by setting the QUAD preprocessor flag. In this case, `evortran` operates with at least 30 significant digits. The real kind precision can be changed at compile time by appending -DQUAD to the FPM_FFLAGS environment variable, for example:

```
1 export FPM_FFLAGS="$FPM_FFLAGS -DQUAD"
```

Alternatively, the user can add the -DQUAD flag directly to the `exports_debug.sh` or `exports_run.sh` files, where FPM_FFLAGS is defined. These files should be sourced before compilation to ensure the correct build configuration is used, as discussed above.

Once the environment variables are sourced, one can compile the project with:

```
1 fpm build
```

The build process is handled by `fpm`, which automatically downloads and compiles all required dependencies. These include the meta-dependency `openmp`, which enables multithreaded parallelization, and a Fortran error-handling module:

```
1 [dependencies]
2 openmp = "\*"
3 error-handling = { git = "https://github.com/SINTEF/
      fortran-error-handling.git", tag = "v0.2.0" }
```

In addition, `dev-dependencies` are specified in the `fpm.toml` file which are used in example applications and test programs:

```
1 [dev-dependencies]
2 csv-fortran = { git = "https://github.com/jacobwilliams/
      csv-fortran.git" }
3 pikaia = { git = "https://github.com/jacobwilliams/pikaia.
      git", tag = "2.0.0" }
```

The package `pikaia` is another GA framework implemented in Fortran [49,50]. It is included as a `dev-dependency` because it is used for comparisons with `evortran` in example applications, see Section 4.1.2 and Section 4.1.4. The package `csv-fortran` is used for exporting data to csv-files.

To use `evortran` as a dependency in another `fpm` project, one can simply add the following entry to the `fpm.toml` file of the project:

```
1 [dependencies]
2 evortran = { git = "https://gitlab.com/thomas.biekoetter/
      evortran" }
```

This makes it straightforward to integrate `evortran` into existing programs and libraries.

After building `evortran` one can install the compiled executables that are contained in the `app` folder locally by running:

```
1 fpm install
```

This command copies the built executables to a default location, typically `~/.local/bin`, and the compiled module files are copied to `~/.local/include`. If these paths are included in the PATH environment variable, one can run the executable from any directory without having to reference the full build path. If needed, you can change the installation prefix using the -prefix flag.

### 3.1.3 Running tests and applications

evortran includes a set of test programs to verify that the library is working correctly. Tu run a specific test program, use the fpm test command followed by the name of the test program:

```
1 fpm test <name_of_test_program>
```

This will execute the selected test and output the results to the terminal. To view the available test programs, navigate to the `test` directory and its sub-directories.

Similarly, evortran provides example application programs located in the `app` directory, which demonstrate how to use evortran in practical scenarios. To run a specific application, one can use the `fpm run` command followed by the name of the application program:

```
1 fpm run <name_of_app_program>
```

This will compile and execute the chosen application, displaying its output in the terminal. One can explore the available application programs by browsing the `app` directory and its sub-folders. These applications include several of the examples discussed in Section 4. These are intended to be a convenient starting point for understanding the library and for developing new programs that use evortran.

## 3.2 Quick start: basic usage and main features

To help new users quickly get started with evortran, this section presents a minimal yet complete example that demonstrates how to use the library to solve an optimization problem. The example program `quick_start` is located in the `app` directory and can be executed using the following command:

```
1 fpm run quick_start
```

This program uses a GA to find the global minimum of the well-known *Rosenbrock function* [61], which is commonly used as a benchmark for optimization algorithms. The Rosenbrock function is defined as

$$f(x,y) = (a-x)^2 + b(y-x^2)^2, \tag{7}$$

where $a = 1$ and $b = 100$ are parameters that define the shape of the function. The function has a global minimum at $(x,y) = (a,a^2) = (1,1)$, where $f(a,a^2) = 0$. The Rosenbrock function is a suitable test case for optimzation algorithms, as it poses challenges related to convergence to the global minimum since the minimum is located inside a long, narrow, parabolic-shaped flat valley. The source code for this example is the following:

```
1  program quick_start
2
3    use evortran__util_kinds, only : wp
4    use evortran__individuals_float, only : individual
5    use evortran__evolutions_float, only : evolve_population
6    use evortran__prng_rand, only : initialize_rands
7
8    implicit none
9
10   type(individual) :: best_ind
11   real(wp) :: x
```

```
1408   12      real(wp) :: y
1409   13
1410   14      real(wp), parameter :: a = 1.0e0_wp
1411   15      real(wp), parameter :: b = 1.0e2_wp
1412   16      real(wp), parameter :: xmin = -2.0e0_wp
1413   17      real(wp), parameter :: xmax = 2.0e0_wp
1414   18      real(wp), parameter :: ymin = -1.0e0_wp
1415   19      real(wp), parameter :: ymax = 3.0e0_wp
1416   20
1417   21      call initialize_rands(mode='twister')
1418   22
1419   23      best_ind = evolve_population(  &
1420   24        100, 2, rosenbrock,  &
1421   25        mating='blend',  &
1422   26        elite_size=1,  &
1423   27        fitness_target=1.0e-10_wp,  &
1424   28        mutate='gaussian',  &
1425   29        mutate_prob=0.5e0_wp,  &
1426   30        mutate_gene_prob=0.5e0_wp,  &
1427   31        mutate_gaussian_sigma=1.0e-3_wp)
1428   32
1429   33      call get_xy(best_ind%genes, x, y)
1430   34
1431   35      write(*,*) 'Rosenbrock function:'
1432   36      write(*,*) '  Minimum at x, y =', x, y
1433   37      write(*,*) '  f(x,y) =', best_ind%get_fitness()
1434   38
1435   39  contains
1436   40
1437   41    pure subroutine rosenbrock(ind, f)
1438   42
1439   43      class(individual), intent(in) :: ind
1440   44      real(wp), intent(out) :: f
1441   45
1442   46      real(wp) :: x
1443   47      real(wp) :: y
1444   48
1445   49      call get_xy(ind%genes, x, y)
1446   50
1447   51      f = (a - x)**2 + b * (y - x**2)**2
1448   52
1449   53    end subroutine rosenbrock
1450   54
1451   55    pure subroutine get_xy(genes, x, y)
1452   56
1453   57      real(wp), intent(in) :: genes(2)
1454   58      real(wp), intent(out) :: x
1455   59      real(wp), intent(out) :: y
1456   60
1457   61      x = xmin + genes(1) * (xmax - xmin)
1458   62      y = ymin + genes(2) * (ymax - ymin)
1459   63
1460   64    end subroutine get_xy
1461   65
1462   66  end program quick_start
```

The program illustrates the key components needed to define and solve an optimization prob-

lem using `evortran`:

– The program starts by importing the relevant modules and defining parameters for the Rosenbrock function and the domain of the variables $x$ and $y$.

– The pseudo-random number generator is initialized with the Mersenne Twister algorithm.

– The call to `evolve_population` is the main entry point for the optimization. Here, we define:

    – A population size of 100 and gene length 2 (for variables $x$ and $y$),

    – The objective function `rosenbrock` as the fitness function,

    – Blend crossover as mating method, and Gaussian mutation,

    – Elitism with one elite individual per generation,

    – A termination criterion based on reaching a fitness value below $10^{-10}$.

– The fitness function is defined as a pure subroutine taking an individual and returning the value of the Rosenbrock function.[8]

– The helper subroutine `get_xy` rescales the genes values from the normalized interval $[0, 1]$ to the domain $-2 \le x \le 2$ and $-1 \le y \le 3$ in which the global minimum should be determined by the algorithm.

– Finally, the location of the found minimum and the corresponding value of the Rosenbrock function is printed.

Executing this example yields the output:

```
1   Rosenbrock function:
2     Minimum at x, y =   0.99996858723522219
          0.99993706186036069
3     f(x,y) =    9.8805221531614218E-010
```

This demonstrates that `evortran` successfully locates the global minimum of the Rosenbrock function with high accuracy.

Users can adapt this minimal example to define and optimize their own objective functions by simply modifying the fitness function and adjusting the meta-parameters of the GA accordingly. Instead of defining the fitness function inside of the main program after the `contains` statement, one can also define the fitness function in a separate module and import the function at the beginning of the program.

## 3.3 Using evortran from Python

To increase accessibility and broaden the potential user base, a Python interface to the core optimization routines of `evortran` is provided through a separate Fortran `fpm` package called `pyevortran`. This interface allows users to take advantage of the performance of Fortran-based GAs while working within Python environments.

The `pyevortran` package is a minimal Fortran project that uses `evortran` as a dependency. It contains only two modules: the module `pyevortran__evolutions_float` provides a C-binding wrapper for the `evolve_population` routine, and the module `pyevortran__migrations_float` provides a C-binding wrapper for the `evolve_migration` routine. So far only the routines acting on float populations are implemented. The two routines are compiled into a shared library which is then used by the Python package `pyevortran` that is installed using `pip`. Within this Python package, the shared library is accessed using Python's `ctypes` module to expose the two functions

---

[8]Recall that, unlike many other GA frameworks which maximize the fitness function, `evortran` minimizes it, such that the Rosenbrock function is implemented as shown in Eq. (7).

evolve_population and evolve_migration. These functions can be called from Python to optimize arbitrary Python functions of the form f(x), where x is a NumPy array.

The Python interface currently supports only the two mentioned high-level optimization routines. Nevertheless, these routines allow the use of a variety of GA configurations by specifying optional arguments, just as in the native Fortran implementation. Another limitation to note is performance: when using the Python interface, the fitness function is typically written in Python and may run significantly slower than a compiled Fortran equivalent. If the evaluation of the fitness function dominates runtime of the GA, it may be worthwhile to translate the function into Fortran and use the native evortran package for optimal performance.

For installation, the pyevortran interface requires a Python v.3 installation, the gfortran Fortran compiler, and fpm version $\geq$ 0.12.0, as earlier versions do not support building shared libraries. The version of fpm currently distributed via PyPI is outdated (currently version 0.10.0) and will not work. To install a suitable version of fpm, one can download a recent binary from the GitHub releases page [60], make it executable, and move it to a directory in the system's PATH, e.g.:

```
1 chmod +x fpm
2 mv fpm ~/.local/bin/
```

In addition, patchelf must be installed to embed runtime library paths (RPATH) into the shared libraries, ensuring that Python can find them at runtime without needing to manually set LD_LIBRARY_PATH. On Debian-based systems, patchelf can be installed with:

```
1 sudo apt install patchelf
```

With the above mentioned prerequisites installed, one can proceed to install pyevortran. The first step is to clone the repository and navigate to its main folder:

```
1 git clone https://gitlab.com/thomas.biekoetter/pyevortran
2 cd pyevortran
```

Then one must source one of two provided environment setup scripts. As for evortran, see the discussion in Section 3.1.2, the script exports_debug.sh is intended for development and enables runtime checks and assertions, while exports.sh is optimized for production use by enabling compiler optimizations and disabling checks. These scripts also set up the number of OpenMP threads. To use the debug version, run:

```
1 source exports_debug.sh
```

Alternatively, to use the optimized configuration, run:

```
1 source exports.sh
```

Once the environment is configured, pyevortran can be built and installed with:[9]

```
1 make pyevortran
```

---

[9]Updating Python packaging tools may help prevent potential installation issues: `python -m pip install -upgrade pip setuptools wheel`.

The makefile invokes `fpm` to build the shared library, uses `pip` to install the Python wrapper into the current environment, and finally applies `patchelf` to embed the necessary RPATHs into the shared libraries. After successful installation, users can import `pyevortran` in Python and use the `evolve_population` and `evolve_migration` functions to apply GAs to their own objective functions.

To test the installation, and to demonstrate the usage of `pyevortran`, a simple test program `test_rastrigin.py` is provided in the `python/test` folder:

```python
import numpy as np
from pyevortran.evolutions import evolve_population

A = 10
def rastrigin(x):
    return A * len(x) + np.sum(x**2 - A * np.cos(2 * np.pi * x)
        )

print()
print("Minimizing Rastrigin function:")
print()
for dim in range(2, 11):
    print("  Number of dimensions =", dim)
    xmin = evolve_population(
        rastrigin, dim,
        pop_size=1000,
        lower_lim=-5.12, upper_lim=5.12,
        max_generations=1000,
        fitness_target=1e-9, verbose=False,
        selection="rank", selection_size=100,
        elite_size=100)
    print("    Minimum at xmin =", xmin)
    print("            f(xmin) =", rastrigin(xmin))
    print()
```

This script demonstrates the use of the `evolve_population` function from the Python interface to minimize the Rastrigin function, see Section 4.1.1 for details. The script performs minimization for dimensions ranging from two to ten and prints both the location of the minima and the corresponding function values. We note that only the first two arguments, the fitness function (`rastrigin`) and the dimensionality of the problem (`dim`), are required. All other arguments are optional and allow the user to specify GA parameters such as population size, selection strategy, elite size, and termination criteria.

# 4 Example applications

To demonstrate the flexibility and performance of `evortran`, this section presents several example applications ranging from standard mathematical benchmarks to physics-motivated use cases. In Section 4.1 we discuss the minimization of common multimodal test functions that are frequently used to evaluate optimization algorithms. Next, in Section 4.2 we showcase two realistic applications from theoretical physics. First, we present a fit of a beyond-the-Standard-Model (BSM) theory with an extended Higgs sector against data from the Large Hadron Collder (LHC) in Section 4.2.1. Afterwords, we show how `evortran` can be used to reconstruct a stochastic gravitational wave background predicted by a cosmological phase transition in the early Universe from mock data of the Laser Interferometer Space Antenne (LISA) in Section 4.2.2.

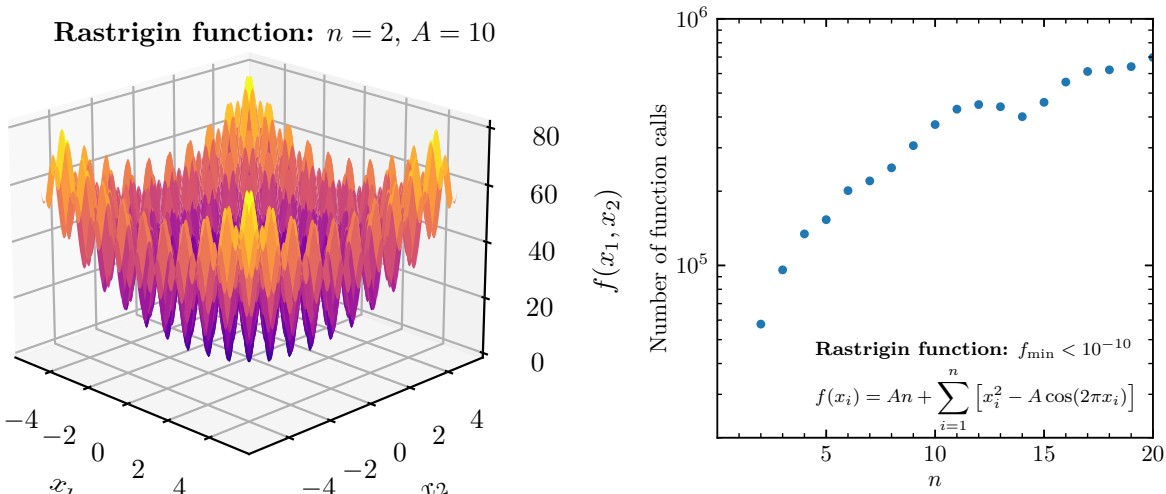

Figure 1: Left: Surface plot of the $n = 2$ dimensional Rastrigin function with $A = 10$ in the domain $x_{1,2} \in [-5.12, 5.12]$. Right: Number of calls to the Rastrigin function against the number of dimensions $n$ until the global minimum has been determined with a function value $f_{\min} < 10^{-10}$ using the example application `benchmark_rastrigin`.

## 4.1  Minimizing multimodal benchmark functions

Multimodal test functions are commonly used in the evaluation and benchmarking of optimization algorithms, particularly GAs, due to their complex landscapes characterized by multiple local minima. Successfully identifying the global minimum of such functions is a core strengths of GAs. In this subsection, we use `evortran` to minimize several standard multimodal functions to demonstrate its performance and to serve as templates that can be adapted by the user to new optimization problems.

### 4.1.1  Rastrigin function

The Rastrigin function [62–64] is a well-known and widely used multimodal benchmark function in global optimization. It is particularly challenging for optimization algorithms due to its highly repetitive landscape filled with many local minima. This makes it an excellent test function for GAs to locate the globally best solution under the presence of many sub-optimal candidate solutions. The Rastrigin function is defined as

$$f(x_i) = A \cdot n + \sum_{i=1}^{n} \left[ x_i^2 - A \cdot \cos(2\pi x_i) \right], \tag{8}$$

where $A = 10$ and $x_i = x_1, \ldots, x_n$, with $n$ being the number of dimensions. The global minimum is located at $x_i = 0$ for all $i$, where the function reaches the minimum value $f(x_i = 0) = 0$. The Rastrigin function is typically minimized in the domain $x_i \in [-5.12, 5.12]$. The left plot of Fig. 1 shows a surface plot of the Rastrigin function in two dimensions.

To demonstrate the capabilities of `evortran`, we performed a series of optimization runs using the `benchmark_rastrigin` example program included in the app folder of the repository. In this benchmark, the goal is to find the global minimum with a precision of $f_{\min} < 10^{-10}$ for dimensions ranging from $n = 2$ to $n = 20$. The GA is executed for each dimensionality separately, and the number of fitness function evaluations until convergence, as well as the final minimum value of the function found, are stored in a CSV file. The core

of the `benchmark_rastrigin` program consists of a loop over the number of dimensions (`currdim`) in which the `evolve_population` function is called:

```
1  best_ind = evolve_population(  &
2    10000, currdim, rastrigin,  &
3    mating='blend',  &
4    elite_size=100,  &
5    lower_lim=-5.12e0_wp,  &
6    upper_lim=5.12e0_wp,  &
7    selection='rank',  &
8    selection_size=100,  &
9    fitness_target=1.0e-10_wp,  &
10   mutate_prob=0.1e0_wp,  &
11   mutate_gene_prob=0.1e0_wp)
```

Here, a large population size of 10,000 individuals and a relatively high number of elite individuals (100) are used to ensure robust exploration across all dimensions. The GA uses blend crossover, rank-based selection, and gaussian mutation, see Section 2.3 for details. The GA terminates once a fitness value of $10^{-10}$ has been found, which is set via the argument `fitness_target`.[10]

Despite using the same GA settings for all dimensions, this basic setup is sufficient to reliably identify the global minimum of the Rastrigin function in all tested dimensions. The performance could be further optimized by tuning the GA parameters individually for each dimension (for example, using smaller population sizes in lower-dimensional cases) but the aim here is to demonstrate the generality and robustness of the algorithm rather than maximizing efficiency. Even with this general setup, the full benchmark test program completes in approximately 7 seconds on a standard laptop equipped with 8 CPU threads. In the right plot of Fig. 1 we show the number of fitness function evaluations required to reach the desired precision of $f < 10^{-10}$ as a function of the number of dimensions $n$. The plot reveals that approximately 60,000 evaluations are needed for the two-dimensional test case, while the number increases gradually with dimensionality, reaching about 700,000 evaluations at $n = 20$. This growth is expected due to the exponential increase in the search space volume with dimension.

To illustrate the effects of parallelization in a case where fitness evaluations are computationally inexpensive, we now consider the case of minimizing the 20-dimensional Rastrigin function. For an increasing number of threads, up to a maximum of 40 threads, the minimization was repeated ten times each, and the average wall time per minimization is shown in Fig. 2. The blue points correspond to results obtained with the `evolve_population` routine, and the orange points correspond to the `evolve_migration` routine. Minimization with `evolve_population` was performed as discussed above, except that the `fitness_target` argument was left unset, such that the GA runs through all generations without early convergence. For the minimization with the `evolve_migration` routine, a population size of 500 and a total number of 20 populations was used, giving a total of 10,000 individuals, the same as for the minimization using the `evolve_population`. As a result, the wall time for both routines is comparable. As one can see in Fig. 2, the `evolve_population` routine shows only modest speedup in this case, with the wall time decreasing from roughly eleven seconds using a single thread to just below six seconds using more than ten threads. The `evolve_migration` routine benefits more significantly from parallelization here, with the wall time dropping to about five seconds with only two threads, to roughly one second with ten threads, and to around 0.8 seconds for 20 threads or more. In both cases, no further improvement is observed

---

[10]Using a fitness target value to control the desired precision of the solution is only appropriate in this case because the global minimum of the Rastrigin function is known to be exactly zero. For functions where the global minimum is not known, convergence criteria must be defined differently.

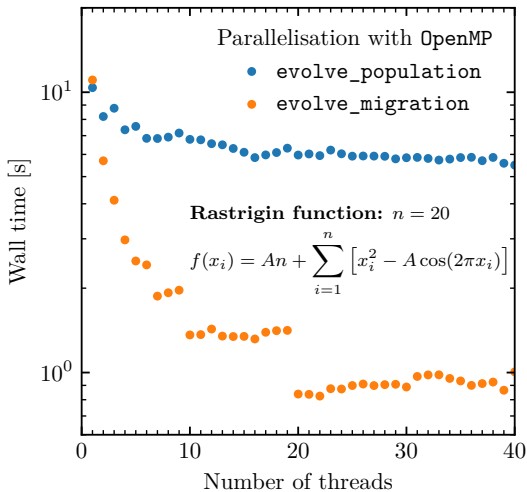

Figure 2: Wall time as a function of the number of threads for the example program `benchmark_rastrigin_nthreads`. For each number of threads, the 20-dimensional Rastrigin function was minized ten times, and the plot shows the average wall time per minimzation. The blue points show the wall time using the `evolve_population` routine, and the orang points show the wall time using the `evolve_migration` routine.

beyond approximately 20 threads, as the runtime is then dominated by the constant overhead of the GA itself. The difference in runtime improvement between both routines can be attributed to the larger parallelization overhead of `evolve_population`, which is more advantageous when fitness evaluations are costly, whereas `evolve_migration` incurs smaller overhead and scales more efficiently with thread count in this example. The runtime improvements with parallelization become more significant when the evaluation of the fitness function is computationally expensive and dominating the overall runtime, as will be demonstrated in a more realistic application in Section 4.2.2.

### 4.1.2 Michalewicz function

Another widely used multimodal test function for evaluating the performance of global optimization algorithms is the Michalewicz function [65]. The function is defined in $n$ dimensions as

$$f(x_i) = -\sum_{i=1}^{n} \sin(x_i) \left[ \sin\left( \frac{i x_i^2}{\pi} \right) \right]^{2m} \tag{9}$$

where the search domain is $x_i \in [0, \pi]$, and the parameter $m$ controls the sharpness of the valleys and ridges that structure the function. We show a surface plot of the two-dimensional Michalewicz function in the left plot of Fig. 3. A common and recommended value is $m = 10$, which is also adopted in our analysis. As $m$ increases, the local minima become steeper and narrower, making the optimization landscape significantly more difficult to navigate. In total, the Michalewicz function has $n!$ local minima, and only one of them corresponds to the global minimum.

A key distinctions between the Michalewicz function and the Rastrigin function discussed in Section 4.1.1 lies in the nature of their local minima. The local minima of the Rastrigin function are located in a regular pattern, whereas the Michalewicz function features broad, almost flat valleys that can mislead an optimizer into converging prematurely. These valleys exhibit very small gradients, which can render gradient-based methods ineffective. At the same

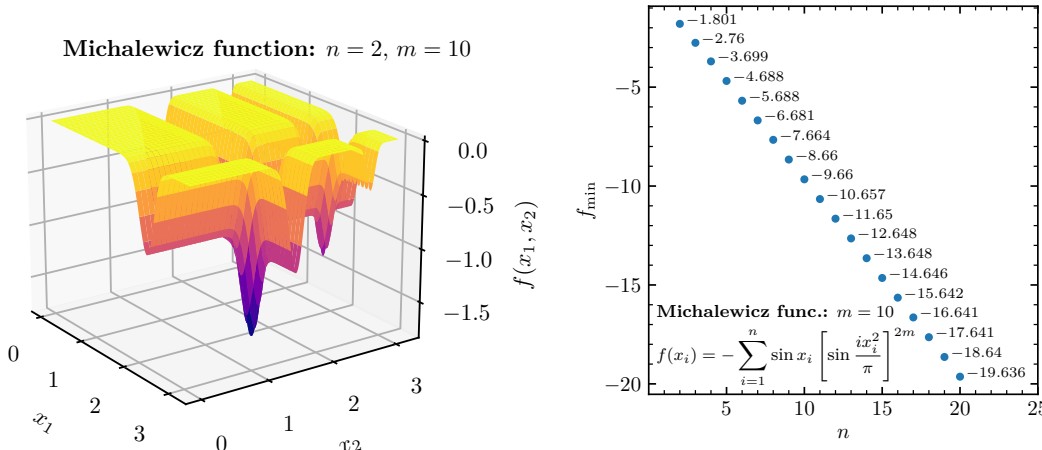

Figure 3: Left: Surface plot of the $n = 2$ dimensional Michalewicz function with $m = 10$ in the domain $x_{1,2} \in [0, \pi]$. Right: Minimal function values $f_{\min}$ of the Michalewicz function for $m = 10$ against the number of dimensions $n = 2, \ldots 25$ that were found using the example application `benchmark_michalewicz`.

time, the global minimum is confined to a narrow, sharply defined region of the parameter space, further increasing the difficulty of the search.

To demonstrate how `evortran` can be used to tackle this problem, we include a test program `benchmark_michalewicz` in the app folder of the repository. This program runs a GA on the Michalewicz function for dimensions ranging from $n = 2$ to $n = 100$. To search for the global minimum, we use the following call to `evolve_population` from evortran:

```
1 best_ind = evolve_population( &
2   1000, n, michalewicz, &
3   lower_lim=0.0e0_wp, &
4   upper_lim=pi, &
5   selection='rank', &
6   mating='blend', &
7   mutate='gaussian', &
8   selection_size=100, &
9   elite_size=10, &
10  mutate_prob=0.4e0_wp, &
11  mutate_gene_prob=0.1e0_wp)
```

This configuration uses a population size of 1,000, applies rank-based selection, blend crossover and gaussian mutation, and it includes both elitism with 10 individuals per generation. In the right plot of Fig. 3 we show the minimum value of the test function achieved at the end of the GA as a function of the dimension $n$ up to $n = 25$. For $n = 2, 5, 10$ the global minimum of the Michalewicz function is known, and we find good agreement with the minimum found using `evortran`.

For larger number of dimensions, the global minium of the Michalewicz function is in general not known. To assess the quality of the solutions found by `evortran`, we therefore apply also the `Pikaia` GA framework to the same problem, see also the discussion in Section 1. For the comparison, we use the default settings of `Pikaia`, except we increase the population size to 1,000 and the maximum number of generations to 10,000. The latter is ten times higher than the corresponding setting in the `evortran` run. We found that in this way both algorithms use roughly the same runtime. It is important to emphasize that our goal is not a performance comparison between `Pikaia` and `evortran`. The goal of this comparison is

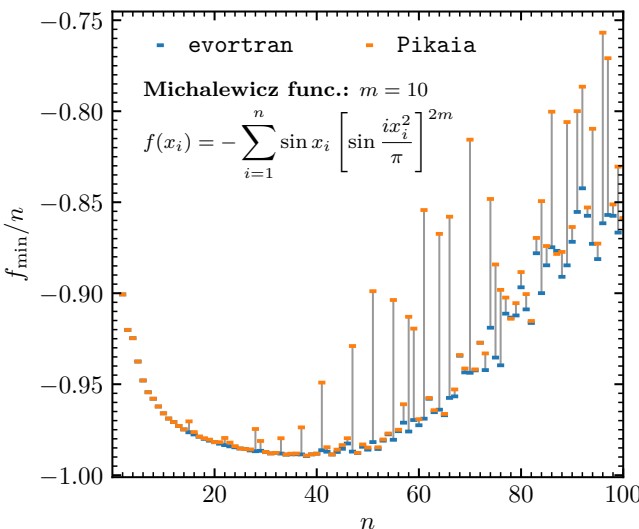

Figure 4: Minimal function values $f_{\min}$ of the Michalewicz function normalized to the dimensionality $n$ for $m = 10$ as a function of the number of dimensions $n = 2, \ldots 100$ that were found using the example application `benchmark_michalewicz`. Blue and orange points indicate the values obtained with `evortran` and `Pikaia`, respectively, see text for details.

simply to verify that `evortran` reliably identifies minima that are at least as deep as those found by `Pikaia`, and to therefore be able to validate the `evortran` result in the absence of known solutions to the optimization problem.

In Fig. 4 we show the minimum values of the Michalewicz function $f_{\min}$ devided by the number of dimensions $n$ as a function of $n$ that were found using `evortran` (blue) and `Pikaia` (orange). One can observe that for each value of $n$ the values of $f_{\min}/n$ determined with `evortran` are in agreement or slightly smaller than the values obtained with `Pikaia`. For $n \gtrsim 50$, we observe that for both `evortran` and `Pikaia` the values of $f_{\min}/n$ start to grow with increasing value of $n$. This potentially indicates that both codes struggle to determine the exact global minima in this regime given the settings used in the example program `benchmark_michalewicz`.

### 4.1.3 Himmelblau function

The Himmelblau function [66] serves as a classic benchmark in the study of multimodal optimization problems. Unlike the Rastrigin and Michalewicz functions discussed above, which feature a single global minimum (albeit in a complex landscape), the Himmelblau function presents a qualitatively different challenge because it possesses multiple global minima of equal depth. The four minima are separated by relatively shallow barriers. This makes it an ideal benchmark function for evaluating the ability of an optimization framework not just to find a global minimum, but to recover multiple degenerate global optima. The Himmelblau function is defined as

$$f(x, y) = (x^2 + y - 11)^2 + (x + y^2 - 7)^2. \tag{10}$$

Himmelblau function

Figure 5: Surface plot of the $n = 2$ dimensional Himmelblau function in the domain $x, y \in [-5, 5]$.

It has four known global minima in the search domain $x, y \in [-5, 5]$, located approximately at

$$(x, y) \approx (3.0, \ 2.0), \tag{11}$$

$$(x, y) \approx (-2.805, \ 3.131), \tag{12}$$

$$(x, y) \approx (-3.779, \ -3.283), \tag{13}$$

$$(x, y) \approx (3.584, \ -1.848), \tag{14}$$

all with a function value of $f(x, y) = 0$. In Fig. 5 we show a surface plot of the Himmelblau function in the search domain.

To demonstrate how `evortran` can successfully identify all of these minima, we make use of the `evolve_migration` routine, which is designed to evolve multiple populations in parallel, see the discussion in Section 2.5. In this example, we disable migration between the evolving populations (by chosing a single epoche, see the discussion below), such that each population evolves independently. Due to the stochastic nature of GAs, different populations initialized with random gene values can then converge to different minima. Using a sufficient number of evolving populations enhances the likelihood of recovering all four global minima of the Himmelblau function within a single execution. The corresponding example program `benchmar_himmelblau` can be found in the `app` folder of the repository. In this program, the call to the `evolve_migration` function is as follows:

```
1 best_ind = evolve_migration( &
2   pop_number=20, &
3   epoches=1, &
4   pop_size=50, &
5   gene_length=2, &
6   fit_func=himmelblau, &
7   mating='sbx', &
8   elite_size=1, &
9   lower_lim=-5.0e0_wp, &
10  upper_lim=5.0e0_wp, &
11  max_generations=100, &
12  fittest_inds_final_pops=fittest_inds_final_pops)
```

20 independent populations of size 50 are evolved in parallel for 100 generations. Each individual encodes two genes, corresponding to the two-dimensional input of the Himmelblau

function, with gene values bounded between -5 and 5. Simulated binary crossover is used for mating, and one elite individual is retained in each generation. Only a single epoch is used, which means no migration of individuals occurs between populations since migration in `evolve_migration` is only applied across multiple epochs. This setup allows each population to converge independently. The function returns the best-fit individual across all populations (`best_ind`). Since we are interested in finding all minima, we additionally store the best-fit individual of each population after the completion of the GA using the optional argument `fittest_inds_final_pops`.

Upon completion, the `benchmark_himmelblau` program prints the best solution found across all populations, followed by a detailed list of the best soloutions from within each of the 20 independent populations:

```
1  Fittest overall individual:
2   Genes:    3.5844E+00  -1.8481E+00
3   Fitness:   0.0000E+00
4
5  Fittest individuals in each population:
6      i           x           y          fmin
7      1   3.0000E+00   2.0000E+00   4.4727E-21
8      2   3.5844E+00  -1.8481E+00   0.0000E+00
9      3   3.5844E+00  -1.8481E+00   0.0000E+00
10     4   3.0000E+00   2.0000E+00   9.6754E-19
11     5   3.0000E+00   2.0000E+00   4.2069E-26
12     6  -2.8051E+00   3.1313E+00   7.8886E-31
13     7  -2.8051E+00   3.1313E+00   7.8886E-31
14     8  -2.8051E+00   3.1313E+00   1.3647E-28
15     9   3.5844E+00  -1.8481E+00   1.1177E-22
16    10   3.0000E+00   2.0000E+00   1.1292E-25
17    11  -3.7793E+00  -3.2832E+00   2.9196E-22
18    12   3.5844E+00  -1.8481E+00   0.0000E+00
19    13   3.0000E+00   2.0000E+00   0.0000E+00
20    14   3.5844E+00  -1.8481E+00   0.0000E+00
21    15   3.0000E+00   2.0000E+00   0.0000E+00
22    16  -2.8051E+00   3.1313E+00   1.3996E-24
23    17   3.5844E+00  -1.8481E+00   0.0000E+00
24    18  -3.7793E+00  -3.2832E+00   7.1510E-27
25    19   3.0000E+00   2.0000E+00   5.8376E-28
26    20  -3.7793E+00  -3.2832E+00   3.1554E-30
```

The table lists the population index, the gene values, i.e. the $x$ and $y$ coordinates, of the fittest individual, and the achieved minimum of the Himmelblau function. In the example shown, all four known global minima of the Himmelblau function are discovered by different populations. This demonstrates the effectiveness of evolving multiple independent populations in parallel for identifying multiple (approximately) degenerate solutions of multimodal objective functions.

### 4.1.4 Drop-Wave function

As a final benchmark, we consider the Drop-Wave function, a challenging multimodal test function often used to evaluate the robustness of global optimization methods. The definition of the Drop-Wave function is

$$f(x_i) = -\frac{1 + \cos(12||x||)}{0.5||x||^2 + 2} \quad \text{with } ||x||^2 = \sum_{i=1}^{n} x_i^2. \tag{15}$$

Its global minimum with $f(x_i) = -1$ is located at the center $x_i = 0$ of the search domain $x_i \in [-5.12, 5.12]$. In Fig. 6 we show a surface plot of the Drop-Wave function in two dimensions over the search domain. Due to the presence of steep central basins surrounded by increasingly shallow, oscillating ripples that form many local minima, the landscape of the function misleads optimization algorithms away from the global minimum. This structure makes it extremely difficult for optimization methods to locate the global minimum, even for a relatively small number of dimensions. The Drop-Wave function is particularly well-suited to test the ability of GAs to escape local minima. In this benchmark, we again compare the results obtained with `evortran` with the ones obtained using the `Pikaia` library.

We apply both frameworks to search for the global minimum of the Drop-Wave function in dimensionalities ranging from $n = 2$ to $n = 8$. The corresponding example program `benchmark_dropwave_pikaia` is located in the `app` folder of the `evortran` repository. For `evortran`, we use the `evolve_migration` routine with the following call:

```
1 best_ind = evolve_migration(  &
2   pop_number=90,  &
3   epochs=1000,  &
4   pop_size=200,  &
5   max_generations=300,  &
6   gene_length=i,  &
7   fit_func=dropwave,  &
8   migration_size=10,  &
9   elite_size=10,  &
10  selection_size=100,  &
11  mating='blend',  &
12  fitness_target=-1.0e0_wp + 1.0e-4_wp,  &
13  selection='roulette',  &
14  wheele_size=7,  &
15  mutate='uniform',  &
16  mutate_prob=0.2e0_wp)
```

Here the argument `i` for the `gene_size` corresponds to the number of dimensions which are iterated over. This setup results in a total of 18,000 individuals evolving in parallel in 90 different populations of size 200 each, over a maximum number of 300 generations and 1000 epochs, see the discussion in Section 2.5 for details. We make use of elitism (10 individuals in each population), roulette wheel selection (with `wheele_size=7`), blend crossover, and uniform mutation (with a mutation probability of 20%).[11] Migration exchanges 10 individuals between populations after each epoch. The algorithm halts when the fitness falls below -0.9999, which is within $10^{-4}$ of the known global minimum of the Drop-Wave function, or when the maximum number of epochs is reached.

For comparison, `Pikaia` is configured with similar computational resources by assigning it the same total number of individuals and a maximum of 5000 generations. The initialization is performed as follows:

```
1 call pikaia%init(  &
2   i, xl, xu,  &
3   dropwave_pikaia,  &
4   status_pikaia,  &
5   ngen=5000,  &
6   np=200*90)
```

---

[11]A total number of 90 populations is chosen because the program was executed on a cluster with 90 CPU cores available.

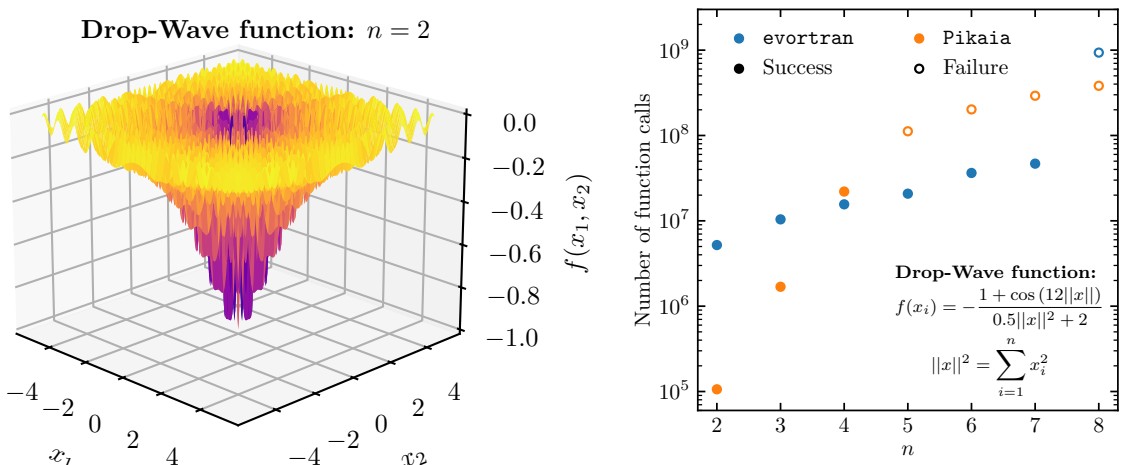

Figure 6: Left: Surface plot of the $n = 2$ dimensional Drop-Wave function in the domain $x_{1,2} \in [-5.12, 5.12]$. Right: Number of calls to the Drop-Wave function against the number of dimensions $n$ until the GA completes by either finding the global minimum with a precision of $10^{-4}$ (filled points) or by reaching the maximum number of epochs/generations (empty points) using the example application `benchmark_dropwave_pikaia`. Blue and orange points correspond to using `evortran` and `Pikaia`, respectively, with the GAs configured as described in the text.

The large number of maximum generations allows `Pikaia` to perform up to approximately 100 million fitness evaluations, which is a number comparable to what is typically required by `evortran` to converge to the global minimum in more than 5 dimensions. The arguments `xl` and `xu` are arrays that determine the search domain in each $x_i$-direction. Since `Pikaia` maximizes the fitness function, whereas `evortran` minimizes, we define a separate function `dropwave_pikaia` here, which differs by an overall minus sign from the fitness function `dropwave` used in the `evolve_migration` call of `evortran`. We again would like to emphasize that this setup is not intended to directly compare runtime performance between `evortran` and `Pikaia`, as the frameworks differ in design and parallelization, but only to validate the `evortran` results against a similar available GA implementation.

In the right plot of Fig. 6 we show the number of times the Drop-Wave function was evaluated for the different values of dimensions $n$ when using the example program `benchmark_dropwave_pikaia`, with the orange points indicating the calls from `evortran` and the blue points indicating the calls from `Pikaia`.[12] The filled points indicate if the GA converged successfully by converging to the global minimum, whereas the empty points indicate if the GA failed by converging to a local minimum. We observe that the GA applied with `evortran` is able to find the global minimum of the Drop-Wave function up to $n = 7$ dimensions, requiring a total number of about $5 \cdot 10^7$ function calls for $n = 2$ dimensions and of about $5 \cdot 10^8$ function calls for $n = 7$ dimensions. The GA applied with `Pikaia` is able to find the global minimum of the Drop-Wave function with a substantially smaller number of function evaluations for $n = 2$ and $n = 3$ dimensions, and with a similar number of functions calls than the GA applied with `evortran` for $n = 4$ dimensions. For $n > 4$ the GA applied with `Pikaia` is not able to locate the global minimum in the give maximum number of generations, and thus

---

[12]We slightly modified the `Pikaia` algorithm for this example by changing the convergence condition. We implemented that the `Pikaia` algorithm only halts if either the global minimum was found with a precision of $10^{-4}$, or if the maximum number of generations was reached. In this way the `Pikaia` algorithm allows for a larger number of function calls and behaves more similarly to the `evortran` algorithm.

shows a larger number of function calls for $n = 4, 5, 6$ than the GA applied with `evortran` because the latter halts once the global minimum is found within the required precision.

To demonstrate the flexibility of the `evortran` library, and to compare the performance of different crossover methods, we conducted a comprehensive set of runs using the `evolve_migration` function to minimze the Drop-Wave function using different crossover routines. For each dimensionality from $n = 2$ to $n = 10$, we tested each crossover operator (`blend`, `sbx`, `one-point`, `two-point`, `uniform`) in combination with all possible permutations of the three available selection routines (`tournament`, `rank`, `roulette`) and the three mutation strategies (`uniform`, `shuffle`, `gaussian`). Each unique combination was run 10 times, resulting in a total of $3 \cdot 3 \cdot 10 = 90$ runs per crossover method and dimension. The GA was configured via the following call:

```
1 best_ind = evolve_migration(  &
2   pop_number=90,  &
3   epoches=1000,  &
4   pop_size=200,  &
5   max_generations=300,  &
6   gene_length=number_dimensions,  &
7   fit_func=dropwave,  &
8   migration_size=2,  &
9   elite_size=2,  &
10  selection_size=100,  &
11  selection=selection_opts(i_selection),  &
12  mating=mating_opts(i_mating),  &
13  mutate=mutate_opts(i_mutate),  &
14  fitness_target=-0.999,  &
15  wheele_size=7,  &
16  verbose=verbose)
```

Here, `evolve_migration` was used to evolve 90 populations in parallel with light migration (`migration_size=2`) between them. The number of epochs was set to 1000, and each population consisted of 200 individuals, evolved over up to 300 generations per epoch. The `fitness_target` value was set to -0.999, which lies below the value of any local minimum of the Drop-Wave function, thereby serving as an effective criterion to identify convergence to the true global minimum. Default values were used for all optional arguments of the selection, crossover and mutation routines, except for a `wheele_size` of 7 for `roulette` selection.

The results are summarized in Fig. 7, which shows the number of function evaluations required for successful convergence versus dimensionality $n$. The total number of successful runs is indicated with numbers per method and dimension to quantify robustness. Also shown with horizontal bars are the mean values of the function calls per dimension for each crossover method, including only the successfully converged runs. The plot reveals several key features regarding the robustness of the different crossover strategies. The crossover methods specifically designed for continuous optimization problems, `blend` and `sbx`, outperform the more generic binary-inspired crossovers, `one-point`, `two-point`, and `uniform`, particularly in lower and intermediate dimensions $n \leq 6$. This suggests that using continuous crossover operators can significantly enhance the reliability of a GA in continuous solution spaces. Up to dimension $n = 7$, the `blend` crossover shows the highest robustness, achieving successful convergence in at least 75 out of 90 runs, highlighting its effectiveness across a wide range of settings. For higher dimensionalities ($n = 8, \ldots, 10$), the `sbx` crossover becomes more robust, maintaining a convergence success rate of over 20%. Interestingly, the `uniform` crossover, while generally less robust at lower dimensionality, shows a certain robustness in performance at high dimensionality, matching the robustness of `blend` and `sbx` for $n = 8$ and beyond. It maintains a minimum success rate of roughly 10% across all dimensions.

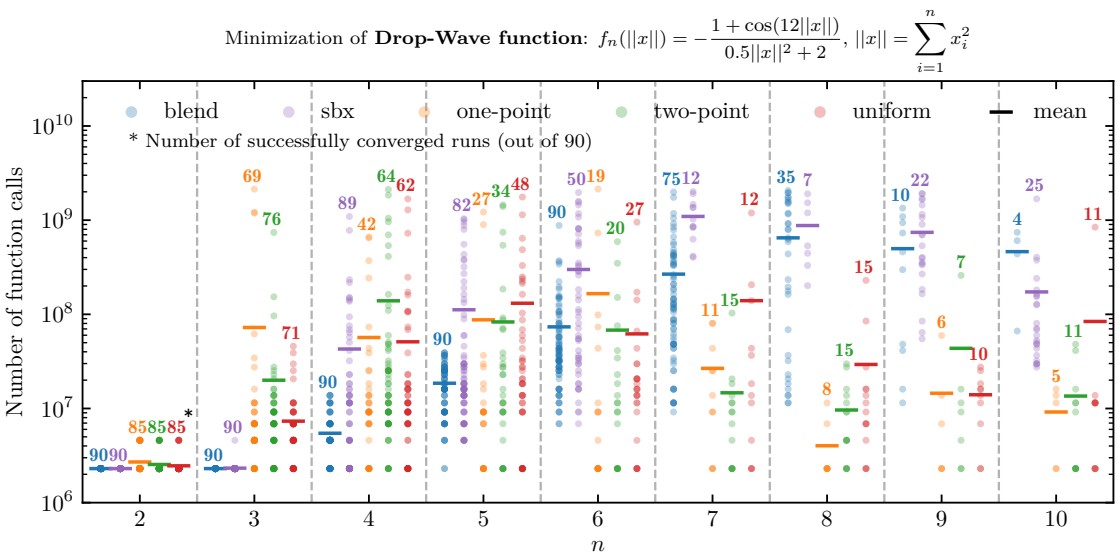

Figure 7: Number of calls to the Drop-Wave function against the number of dimensions $n$ until the GA successfully converged to the global minimum for the different crossover methods implemented in `evortran`. The horizontal bars indicate the mean values per dimension for each crossover method, and the colored numbers show the total number of successfully converged runs out of a total number of 90 runs (see the discussion in the text for details).

These results underscore the importance of carefully selecting the components of a GA to match the characteristics of the specific optimization problem. The flexible design of `evortran` allows users to easily explore and combine different evolutionary operators. This modularity is a key advantage of the library, making it a powerful tool in wide areas of in scientific computing.

## 4.2 Physics applications

To demonstrate the capabilities of `evortran` in real-world research settings, we present two physics-motivated applications that go beyond the mathematical benchmark problems discussed above. The first application focuses on a global fit of an extended Higgs sector in a beyond-the-Standard-Model (BSM) scenario to existing data from the Large Hadron Collider (LHC). This includes fitting the observed signal strengths of the Higgs boson at 125 GeV as well as incorporating cross section limits from searches for additional Higgs bosons that have (so far) not given rise to a discovery. The second application addresses a problem in cosmology: reconstructing the frequency spectrum of a stochastic gravitational wave background produced by a cosmological first-order phase transition, such as the ones expected from an electroweak phase transition in the early universe. Here, the goal is to reconstruct a consistent signal from LISA mock data, which includes both an injected gravitational wave signal and realistic instrumental noise.

### 4.2.1 Confronting extended Higgs sector with LHC data

A common challenge in studies of BSM physics is the efficient exploration of high-dimensional parameter spaces subject to a complex set of constraints. In particular, models with extended scalar/Higgs sectors often feature many free parameters (eleven in the first scenario studied below, and 14 in a second more general scan). These parameters are subject to both theoret-

ical consistency requirements (such as vacuum stability or perturbativity) and a wide range of experimental constraints. Among the latter are 95% confidence level exclusion bounds on production cross sections from direct searches for additional Higgs bosons at the LHC, as well as precision measurements of the observed 125 GeV Higgs boson. Many of these constraints are either non-differentiable (e.g., hard cutoffs on excluded cross sections) or involve discrete features in the parameter space, which makes gradient-based optimization methods less effective.

GAs offer a natural solution to this type of problem. Their population-based and non-gradient nature allows them to effectively navigate large, non-linear, and non-smooth landscapes, making them well-suited for parameter scans that need to identify viable regions consistent with a diverse set of constraints. An additional advantage of GAs is their ability to uncover multiple, qualitatively distinct regions of parameter space that are all consistent with theoretical and experimental constraints within their respective uncertainties. This is particularly important when different parameter regions compatible with the constraints yield different phenomenology. Since these parameter regions may correspond to local minima of the $\chi^2$-function or likelihood that are only marginally suboptimal compared to the global minimum within the margin of uncertainty, identifying (ideally) all of these regions may be crucial. GAs are well-suited for this task because their stochastic and population-based search can naturally discover such local optima, rather than converging exclusively to a single best-fit solution.

As an example, we focus here on the singlet-extended two Higgs doublet model (S2HDM) [16], a well-motivated BSM scenario that augments the Standard Model by a second Higgs doublet and a complex scalar singlet. We utilized a GA to explore the phenomenology of the S2HDM in previous studies in Refs. [16–18] using the Python GA framework DEAP [42]. In the S2HDM, the presence of the second Higgs doublet opens the possibility of a strong first-order electroweak phase transition, which is an essential requirement for electroweak baryogenesis to explain the baryon asymmetry of the universe, and which cannot be realized in the Standard Model. Additionally, the complex singlet respects a softly-broken global U(1) symmetry, which is spontaneously broken when the real component of the singlet acquires a vacuum expectation value. This results in a pseudo-Nambu-Goldstone boson from the imaginary part of the singlet field, which is stable and provides a viable Higgs-portal dark matter candidate. Notably, a pseudo-Nambu-Goldstone dark matter particle can achieve the observed relic abundance through the standard thermal freeze-out mechanism, while evading the stringent limits from dark matter direct detection experiments due to momentum-suppressed scattering cross sections at leading order [67,68].

A comprehensive discussion of the S2HDM can be found in Ref. [16]. Here, we briefly summarize the key features relevant for the following analysis. The model contains the same fermionic and gauge field content as the Standard Model, but due to the extended scalar sector the physical spectrum includes a total of three CP-even neutral scalar Higgs bosons $h_i$ ($i = 1, 2, 3$) which mix with each other, a CP-odd pseudoscalar Higgs boson $A$, a pair of singly charged Higgs bosons $H^\pm$, and a stable spin-0 dark matter candidate $\chi$.[13] We consider two Yukawa structures: first, a so-called Type I structure where only one Higgs doublet is coupled to fermions, and second, a flavour-aligned setup which can be parameterized by three so-called flavour alignment parameters $\xi_{u,d,\ell}$ [69]. The other free parameters used in the analysis below include the masses $m_{h_{1,2,3}}$, $m_A$, $m_{H^\pm}$, $m_\chi$, the three CP-even scalar mixing angles $\alpha_{1,2,3}$, the ratio of the Higgs doublet vacuum expectation values $\tan\beta$, the singlet vacuum expectation value $v_S$, and an additional mass scale parameter $M$, which controls the allowed mass range of the BSM Higgs states.

To identify phenomenologically viable points in the parameter space of the S2HDM, we

---

[13]The Standard Model of particle physics predicts a single CP-even Higgs boson.

construct a global $\chi^2$-function that incorporates a range of theoretical and experimental constraints. This function is then minimized using a GA implemented in `evortran`. Since evaluating the fitness function requires interfacing with external codes, namely `HDECAY` [70,71] for computing Higgs boson decay properties and `HiggsTools` [72–75] for checking consistency against LHC data (see below), the evaluation is computationally expensive. For this reason, we employ a relatively economical GA with a population size of only 40 individuals, evolved using `evortran`'s `evolve_population` routine over a maximum of 1000 generations, applying tournament selection, blend crossover and uniform mutation. All source code used to obtain the results presented in this section is publicly available in a dedicated Git repository [76].

The theory constraints included in the analysis are the following:

- **Vacuum stability:** We impose that the scalar potential is bounded from below, ensuring that no field direction leads to the potential tending to $-\infty$ for large field values. At tree-level, this leads to analytic conditions requiring certain combinations of quartic couplings appearing in the potential to be positive. The corresponding expressions can be found in Refs. [71,77].

- **Perturbative unitarity:** We verify that the leading-order $2 \rightarrow 2$ scalar scattering amplitudes remain below the threshold $8\pi$ in the high-energy limit. This results in a set of inequalities on combinations of the scalar quartic couplings, whose precise form can be found in Ref. [16].

Both theory constraints are implemented as hard cuts and introduce non-differentiable features into the $\chi^2$ function. The experimental constraints that we consider here are the following:

- **125 GeV Higgs cross sections:** We include cross section measurements of the observed Higgs boson in various production and decay channels, including also the measurements presented in the form of the Simplified Template Cross Section (STXS) framework.

- **Limits from BSM spin-0 resonance searches:** We apply 95% confidence level exclusion limits from a large set of direct LHC searches for additional neutral and charged Higgs bosons. These constraints are implemented as hard cuts.

- **Electroweak precision observables:** We require compatibility with the measured electroweak parameter $\Delta\rho$ within its experimental uncertainty [78], based on its one-loop prediction in the S2HDM [16]. This constraint is included as a hard cut, rejecting parameter points predicting a value of $\Delta\rho$ deviating by more than two standard deviations from the experimental central value, corresponding to a confidence level of about 95%.

The check against the LHC data is carried out in our analysis with the help of the public C++ package `HiggsTools`, which performs a global $\chi^2$ fit to the cross section measurements of the 125 GeV Higgs bosons and checks if all existing cross section limits from BSM scalar searches are satisfied. The `HiggsTools` library can be exposed to Fortran via the `iso_c_binding` module. We created a minimal working example for using `HiggsTools` within an `fpm` project that is available in a dedicated GitLab repository [79].

The global $\chi^2$ function minimized by `evortran` includes the contributions from `HiggsTools` for the 125 GeV Higgs measurements, denoted $\chi^2(h_{125})$ in the following, while all other constraints are incorporated via large fixed penalty values if they are not satisfied. This setup effectively translates the task of finding allowed parameter points into minimizing $\chi^2(h_{125})$ under the condition that all other theoretical and experimental constraints are satisfied, for which derivative-free optimization strategies like GAs are especially useful. To define a meaningful condition at which the GA shall halt, we subtract the Standard Model prediction $\chi^2_{\text{SM}}(h_{125})$ from the total $\chi^2$, such that $\chi^2 = 0$ corresponds to the S2HDM fitting the LHC data as well as the Standard Model (while satisfying all other constraints), while $\chi^2 < 0$ indicates a better fit than the Standard Model. The GA halts if a parameter point pre-

| Parameter | Range |
|---|---|
| $m_{h_1}$ | 125.1 GeV (fixed) |
| $m_{h_2}, m_{h_3}, m_A, m_{H^\pm}, m_\chi$ | [30, 1000] GeV |
| $\alpha_{1,2,3}$ | $[-\frac{\pi}{2}, \frac{\pi}{2}]$ |
| $\tan\beta$ | [1, 20] |
| $M(= m_{12}^2/(\sin\beta\cos\beta))$ | [30, 1000] GeV |
| $v_S$ | [10, 3000] GeV |
| Scenario 1: | Yukawa Type I |
| Scenario 2: | Flavour alignment: $\xi_{u,d,\ell} \in [10^{-3}, 10^3]$ (log prior) |

Table 4: The top rows show the S2HDM parameter ranges used in the scans. The last two rows indicate the Yukawa structure chosen for the scenario 1 and the scenario 2.

dicting $\chi^2 < 0$ was found. During the minimization process, we not only record the best-fit point but also store all parameter points generated during the evolution process that predict $\chi^2 = \chi^2(h_{125}) - \chi_{\text{SM}}^2(h_{125}) \leq 6.18$, which approximately corresponds to a 95% confidence region assuming two degrees of freedom and assuming that the Standard Model prediction is a good approximation of the best fit to the LHC data (see Ref. [75] for details). To obtain a diverse sample of viable parameter points across the scanned regions of parameter space, we run the GA multiple times with different random seeds.

**Scenario 1: Scanning the S2HDM Type I** – 11 free parameters: $\{m_{h_2}, m_{h_3}, m_A, m_{H^\pm}, m_\chi, \tan\beta, \alpha_{1,2,3}, M, v_S\}$

In the first scenario, we use `evortran` to perform a global scan of the S2HDM with Yukawa Type I, using the full parameter space summarized in Table 4. This constitutes a very generic scan where the goal is to test whether the GA is capable of identifying regions in parameter space that yield a 125 GeV Higgs boson (here $h_1$) with properties resembling those predicted by the Standard Model within current experimental precision, while also satisfying all other theoretical and experimental constraints discussed above. In the case where the additional Higgs bosons are substantially heavier, the properties of the 125 GeV Higgs boson are mainly governed by the mixing angles $\alpha_{1,2,3}$ and $\tan\beta$, which must be tuned with some precision in order to reproduce the observed Higgs boson signal strengths. The other parameters are relevant for satisfying the remaining constraints. For instance, the masses of the BSM states have to be chosen in a range that is not excluded by LHC searches. Their allowed values are also strongly correlated with the parameter $\tan\beta$, the mass parameters $M$ and $v_S$, and the mass splitting among the BSM states is constrained by the $\Delta\rho$ parameter.

Technically, this parameter scan is a minimization of a $\chi^2$ function defined over eleven free parameters. The complexity of the task is significant since the $\chi^2$ function encodes roughly 150 independent LHC measurements of the 125 GeV Higgs boson cross sections, in addition to a vast number of cross section limits from direct searches for BSM Higgs bosons. This results in a highly structured and non-trivial eleven-dimensional optimization landscape.

In Fig. 8 we show results from scenario 1, where each point corresponds to a valid parameter point identified by the GA. The left plot shows the distribution of $\chi^2(h_{125})$ values obtained with `HiggsTools` against the coupling coefficient $\kappa_V$ of the CP-even Higgs boson $h_1$ to vector bosons $V = W, Z$, defined as $\kappa_V = g_{h_1 VV}/g_{h_{\text{SM}}VV}$, where $g_{h_1 VV}$ is the coupling predicted in the S2HDM and $g_{h_{\text{SM}}VV}$ is the coupling predicted for a Standard Model Higgs boson of the same mass. This plot demonstrates that the GA successfully identified parameter points consistent with the 125 GeV Higgs boson data at the level of the Standard Model ($\chi^2 \approx 0$) and even slightly better ($\chi^2 < 0$), though the latter is not a statistically significant improvement con-

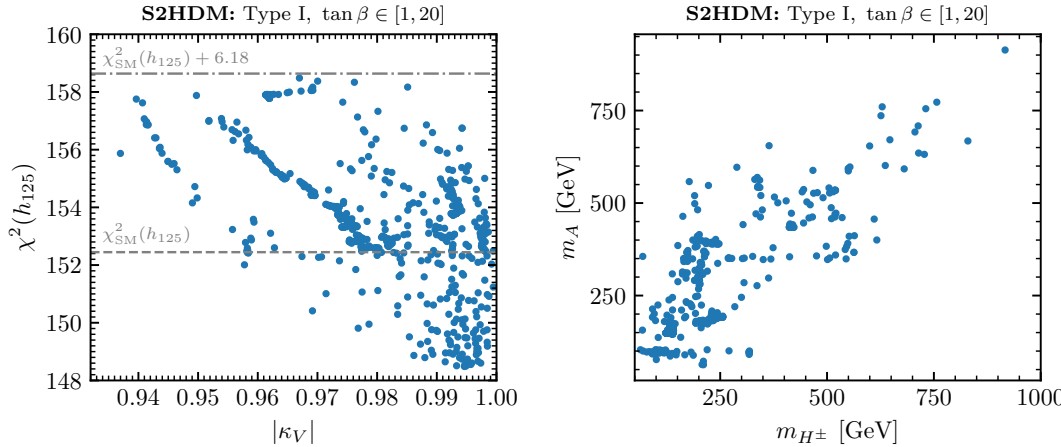

Figure 8: Each point in the two plots corresponds to a valid parameter point found with the GA in the scenario 1 in the S2HDM with type I Yukawa structure. Left: $\chi^2(h_{125})$ obtained with `HiggsTools` against the coupling coefficient $|\kappa_V|$ of the 125 GeV Higgs boson $h_1$. Right: mass of the CP-odd Higgs boson $m_A$ against the mass of the charged Higgs bosons $m_{H^\pm}$.

sidering the larger number of additional parameters of the S2HDM compared to the Standard Model.

The right plot of Fig. 8 shows the mass of the CP-odd Higgs boson $m_A$ against the mass of the charged Higgs boson $m_{H^\pm}$. The GA identified valid parameter points over a wide mass range up to the upper limit 1 TeV of the scan for the masses of the BSM states. The lightest masses found for $A$ and $H^\pm$ are around 200 GeV. Below this, experimental searches from the LHC increasingly constrain the parameter space, and a light $H^\pm$ also introduces substantial loop-level corrections to the di-photon decay rate of the 125 GeV Higgs boson $h_1$, making it harder to achieve agreement with the data. However, this does not imply that smaller values of $m_A$ or $m_{H^\pm}$ are excluded in the S2HDM. Uncovering valid parameter space in this regime would require a dedicated scan that biases the GA toward lower mass ranges (or higher values of $\tan\beta$), instead of the setup applied here that constitutes a broad and generic exploration of parameter space. Biasing the GA towards smaller masses could, for instance, be achieved using a logarithmic prior instead of linear prior for the masses of the BSM Higgs bosons when the GA is initialized, see also the discussion in Section 4.2.2.

In Fig. 9 we show the couplings and masses of the three CP-even Higgs bosons $h_1$, $h_2$, and $h_3$ in the S2HDM with type I Yukawa structure for the valid parameter points found in scenario 1. In each plot, for each valid parameter point three points are displayed with the following color-coding: blue for $h_1$, orange for $h_2$, and green for $h_3$. The left plot shows the vector boson coupling coefficient $\kappa_V(h_i)$ versus the Higgs boson mass $m_{h_i}$. Here it should be noted that due to unitarty there is a sum rule for these coupling coefficients: $\sum_i \kappa_V(h_i)^2 = 1$. Since LHC measurements require $\kappa_V(h_1) \gtrsim 0.93$, see left plot of Fig. 8, this forces the couplings of $h_2$ and $h_3$ to electroweak gauge bosons to be small. The GA identified parameter points with this coupling configuration by accurately sampling the mixing angles $\alpha_{1,2,3}$, as is visible in the plot where both $\kappa_V(h_2)$ and $\kappa_V(h_3)$ lie below about 0.3. Accordingly, for $h_1$ the GA efficiently converges to $\kappa_V(h_1) \approx 1$, in agreement with LHC data requiring the 125 GeV Higgs boson to resemble the Standard Model prediction within an experimental precision at the level of 10%.

Using the Type I Yukawa structure, the modifications of the Higgs boson couplings to fermions can be expressed by means of a single coupling coefficient $\kappa_f(h_i)$ for each Higgs boson, independently of the fermion kind. The right plot shows the fermion coupling coeffi-

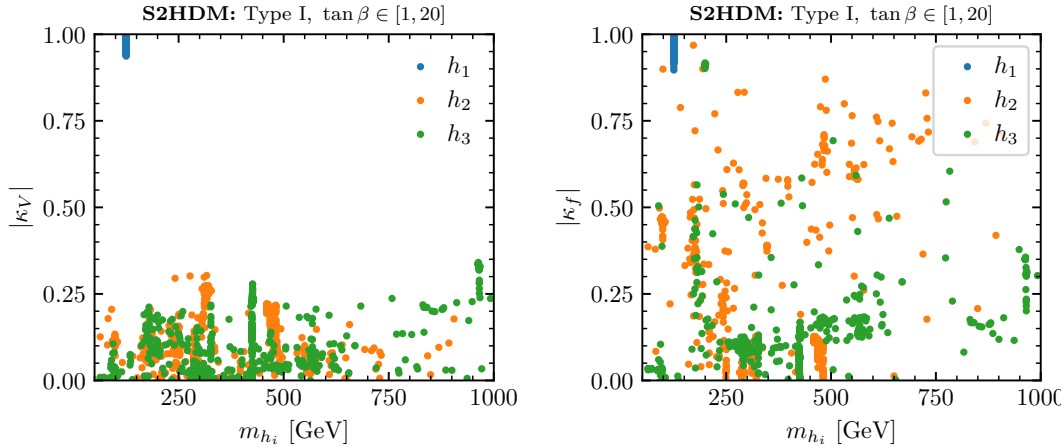

Figure 9: For each valid parameter point found with the GA in the scenario 1 in the S2HDM with type I Yukawa structure, the plots show three points with properties related to the Higgs bosons $h_1$, $h_2$ and $h_3$ in blue, orange and green, respectively. Left: coupling coefficient $\kappa_V$ of the respective state $h_i$ against the mass of the Higgs boson $m_{h_i}$. Left: coupling coefficient $\kappa_f$ of the respective state $h_i$ against the mass of the Higgs boson $m_{h_i}$.

cients defined as $\kappa_f(h_i) = g_{h_i f \bar{f}}/g_{h_{\mathrm{SM}} f \bar{f}}$, where $g_{h_i f \bar{f}}$ is the coupling predicted for the state $h_i$ in the S2HDM, and $g_{h_{\mathrm{SM}} f \bar{f}}$ is the coupling predicted in the Standard Model for a Higgs boson of the same mass. In the type I Yukawa structure, there is an upper bound of $|\kappa_f(h_i)| \leq 1$. Unlike the gauge couplings, $|\kappa_f(h_2)|$ and $|\kappa_f(h_3)|$ can be sizable, i.e. as large as $|\kappa_f(h_1)|$, without violating current LHC data. This behavior is visible in the right plot. Again, as before for $|\kappa_V(h_1)|$ shown in the left plot of Fig. 9, the GA converges to $\kappa_f(h_1) \approx 1$ for the 125 GeV Higgs boson, such that the state $h_1$ resembles a SM Higgs boson.

Since the scan covers all physically distinct combinations of the three scalar mixing angles $\alpha_{1,2,3}$, see Table 4, no distinction between $h_2$ and $h_3$ is imposed on their possible masses and couplings. As a result, the orange points corresponding to the state $h_2$ and the green points corresponding to the state $h_3$ show a similar distribution in the two plots of Fig. 9. Furthermore, the mass ranges for $h_2$ and $h_3$ span from about 63 GeV to 1 TeV. The lower bound arises because masses below 62.5 GeV would open the decay channels $h_1 \to h_2 h_2 / h_3 h_3$, which tends to spoil the Standard-Model-like nature of $h_1$ unless the corresponding couplings governing these decays are fine-tuned to suppress the decays. The GA does not converge to such tuned regions of parameter space in this broad scan. On the upper end, the scan reaches the maximum mass range of 1 TeV set in the input.

We finally note that some points shown in the plots in Fig. 8 and Fig. 9 are clustered along a line in the plot, which reflects the fact that during the GA evolution, all parameter points satisfying $\chi^2 \leq 6.18$ are stored, and not just the final best-fit point. As a result, parameter points originating from gradual changes to a successful candidate solutions during the optimization appears as an approximately continuous trajectory in parameter space.

**Scenario 2: Scanning the flavor-aligned S2HDM** – 14 free parameters: $\{m_{h_2}, m_{h_3}, m_A, m_{H^\pm},$ $m_\chi, \tan\beta, \alpha_{1,2,3}, M, v_S, \xi_{u,d,\ell}\}$
In this second scenario, we consider a more general Yukawa sector for the S2HDM, namely the flavor alignment mentioned above. In contrast to the Type I Yukawa structure where only one Higgs doublet couples to all fermions, the flavor-aligned setup allows both Higgs doublets to couple to fermions, controlled via three additional parameters $\xi_u$, $\xi_d$, and $\xi_\ell$. These alignment parameters act as proportionality factors between the Yukawa couplings of

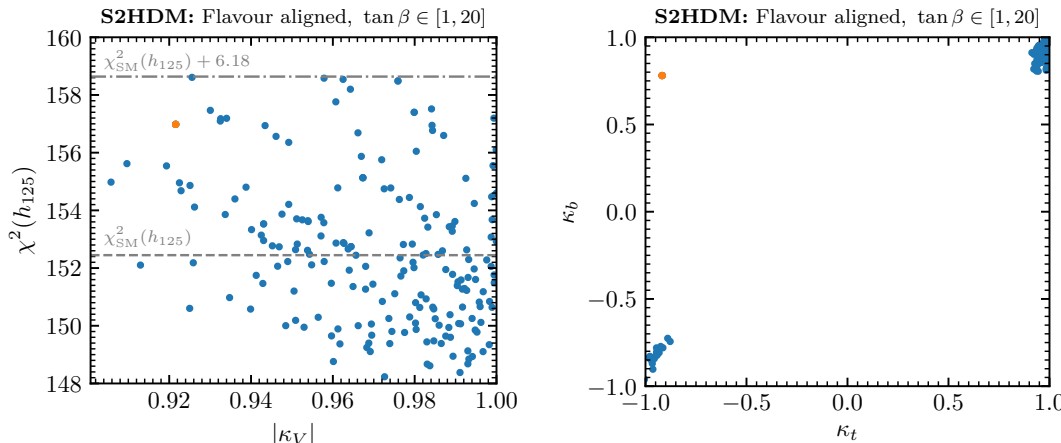

Figure 10: For each valid parameter point found with the GA in the scenario 2 in the flavor-aligned S2HDM, the left plot shows the the values of $\chi^2(h_{125})$ obtained with `HiggsTools` against the coupling coefficient $|\kappa_V|$ of the 125 GeV Higgs boson $h_1$. The right plot shows the $h_1$ coupling coefficient $\kappa_t$ against the coupling coefficient $\kappa_b$. In both plots, the orange points belong to the same parameter point which stands out as the only parameter point for which $\kappa_t$ and $\kappa_b$ have the opposite sign.

the two Higgs doublets and determine which doublet dominantly couples to a given fermion type: up-type quarks $u$, down-type quarks $d$, and charged leptons $\ell$. Specifically, values of $\xi_{u,d,\ell} \ll 1$ correspond to dominant couplings to the first Higgs doublet, while $\xi_{u,d,\ell} \gg 1$ correspond to dominant couplings to the second doublet.

An important consequence is that compared to the Type I Yukawa structure, where the couplings of the Higgs bosons to fermions are governed by a universal modifier $\kappa_f$ for all fermion types, the flavor alignment leads to an independent coupling modifier for each fermion kind: $\kappa_u$, $\kappa_d$, and $\kappa_\ell$, defined analogously to $\kappa_f$ as the ratio of the coupling of a state $h_i$ in the S2HDM to the corresponding coupling of a Higgs boson in the Standard Model with the same mass as the state $h_i$. This results in additional freedom to accommodate the LHC Higgs boson measurements. The coupling coefficients $\kappa_{u,d,\ell}$ depend on the parameters $\xi_{u,d,\ell}$, $\tan\beta$ and the mixing angles $\alpha_{1,2,3}$, such that the couplings of the 125 GeV Higgs bosons depends on various independent parameters, and where various entirely different regions of the complex parameter space can predict a Higgs boson $h_1$ in agreement with the LHC measurements.

In Fig. 10 we show the results obtained for scenario 2, where the GA was used to scan the flavor-aligned S2HDM, see Table 4. Each point in both plots corresponds to a valid parameter point that satisfies all theoretical and experimental constraints. In the left plot, we show the values of the global $\chi^2(h_{125})$ as computed with `HiggsTools` plotted against the absolute value of the coupling coefficient $|\kappa_V|$ of the 125 GeV Higgs boson $h_1$. The plot demonstrates that the GA successfully identifies parameter points where $|\kappa_V| \approx 1$ with $\chi^2(h_{125})$ close to or below the Standard Model value $\chi^2_{\mathrm{SM}}(h_{125})$, indicating good agreement with the LHC measurements of the Higgs boson. This confirms that even in the more general flavor-aligned setup with a total of 14 free parameters the GA is able to determine parameter space regions with the desired features. In this scenario, we find that smaller values of the coupling coefficient $|\kappa_V|$ are still compatible with current LHC data. Values down to $|\kappa_V| \approx 0.912$ are allowed while still yielding valid parameter points. This is in contrast to the more restrictive scenario 1 with Yukawa Type I, see the left plot of Fig. 8, where viable parameter points were only found for $|\kappa_V| \gtrsim 0.935$. Notably, even some parameter points with $|\kappa_V| \lesssim 0.92$ result in a total $\chi^2(h_{125})$ below the Standard Model value $\chi^2_{\mathrm{SM}}(h_{125})$. This indicates that, compared to the

Yukawa Type I, the added freedom of the flavor-aligned Yukawa structure allows the model to better accommodate the LHC measurements in case of sizable deviations in $\kappa_V$ from the SM prediction $\kappa_V = 1$.

In the right plot of Fig. 10, we show the values of the coupling modifiers $\kappa_u$ vs. $\kappa_d$, which specifically correspond to the ratios of the $h_1$ couplings to top and bottom quarks, respectively, compared to the Standard Model. Unlike in the Yukawa Type I, these two couplings are now controlled independently via the alignment parameters $\xi_u$ and $\xi_d$. The distribution shows that most viable parameter points cluster in the region where both $\kappa_u$ and $\kappa_d$ are close to either $+1$ or $-1$, reflecting Standard-Model-like behavior. However, one point clearly stands out as it corresponds to a parameter point where $\kappa_u$ and $\kappa_d$ have opposite signs. The two points in the left and the right plot of Fig. 10 corresponding to this parameter point are highlighted in orange. This sign configuration between $\kappa_u$ and $\kappa_d$ (also referred to as wrong-sign Yukawa coupling regime) is a distinctive feature of the flavor-aligned S2HDM compared to the S2HDM Type I and can, as observed here, still be consistent with current data, though it is associated with modified interference effects in loop-induced Higgs processes (such as gluon fusion production or the $h \to \gamma\gamma$ decay).

In summary, the two example scenarios demonstrate that GAs are effective tools for scanning the complex parameter space of BSM theories. Despite the high dimensionality of the search (eleven and fourteen free parameters) and the large amount of experimental constraints (including around 150 measurements of the 125 GeV Higgs boson and a plethora of exclusion limits from searches for additional scalars), the GA reliably identifies viable parameter regions. Notably, one of the key strengths of GAs is their ability to uncover distinct and potentially isolated solutions in parameter space that are consistent with all constraints and may even yield a better fit to the data. Such solutions might easily be missed by more local or gradient-based methods. This is exemplified in scenario 2 by the identification of a parameter point with opposite signs in the top and bottom Yukawa coupling modifiers, a feature that nonetheless yields an acceptable fit to the LHC data. The S2HDM serves here as one specific but sufficiently complex example to showcase the power of GAs, demonstrating their potential for parameter scans in BSM theories. More broadly, GAs hold great promise for exploring a wide range of particle physics models with a substantial number of unknown parameters and with similar or even greater complexity.

### 4.2.2 Reconstructing gravitational wave spectrum from LISA mock data

To explore the utility of GAs, and in particular the `evortran` library, in a cosmological context, we consider the problem of reconstructing a stochastic gravitational wave signal generated by a first-order phase transition in the early universe. The analysis begins with the construction of mock observational data for the upcoming LISA experiment, into which a synthetic gravitational wave signal is injected based on a physically motivated template predicted for such a cosmological phase transition. To simulate realistic observational conditions, Gaussian noise simulating the LISA instrument sensitivity is added to the signal. This introduces a stochastic component to the data, which complicates the parameter inference and makes traditional gradient-based minimization techniques less effective, whereas a GA is well suited for this type of problem. For the fitting process we use a GA implemented with `evortran` to minimize an overall $\chi^2$-function that quantifies the deviation between the theoretical signal (including LISA's sensitivity curve) and the noisy mock data. This procedure is repeated multiple times to generate a set of reconstructed signals. Each reconstructed signal corresponds to a set of cosmological parameters that govern the spectral shape and amplitude of the gravitational wave signal, such as the strength, duration, and energy release of the phase transition. The resulting distribution of these parameter sets reveals the regions of parameter space compatible with the data, and allows for a comparison with the values originally used for signal injection.

The gravitational wave spectrum produced during a cosmological phase transition carries information about the underlying physics of the early universe. The shape and amplitude of the signal can be parametrized in terms of the strength of the phase transition $\alpha$, defined as the ratio of the vacuum energy released to the radiation energy density, the inverse duration of the transition normalized to the Hubble rate, $\beta/H$, the transition temperature $T_*$ at which the signal is generated, the number of relativistic degrees of freedom in the plasma at that temperature, $g_*$, and the terminal velocity $v_w$ of the bubble walls expanding through the plasma. In this study, we use signal templates derived from numerical simulations that model the generation of gravitational waves produced during a first-order phase transitions. The total gravitational wave spectral power density $\Omega_{\mathrm{GW}}h^2$, with $h = 0.68$ being the dimensionless Hubble constant, is composed of three main contributions,

$$\Omega_{\mathrm{GW}}h^2 = \Omega_{\mathrm{sw}}h^2 + \Omega_{\mathrm{turb}}h^2 + \Omega_{\mathrm{coll}}h^2 \,. \tag{16}$$

Each of these components contributes with a characteristic spectral shape and peak frequency, dependent on the physical parameters mentioned above. The sound wave contribution $\Omega_{\mathrm{sw}}h^2$ results from gravitational wave production from the acoustic oscillations in the plasma after bubble collisions, which dominates the signal in scenarios in which the bubble walls reach a terminal velocity before colliding [80]. The turbulence contribution $\Omega_{\mathrm{turb}}h^2$ arises from magnetohydrodynamic turbulence, i.e. nonlinear plasma motion generated after the transition. The turbulence contribution are not yet well understood, but numerical simulations indicate that it peaks at slightly higher frequencies than the sound wave contribution and with peak amplitudes that are substantially smaller than the one of the sound wave contribution [81]. Finally, $\Omega_{\mathrm{coll}}h^2$ contains the contributions from bubble wall collisions, which account for the direct collisions of expanding bubbles during the transition [82]. Specifically, we model the three contributions using the following templates available in the literature: the sound wave contribution follows the power-law parametrization given in Ref. [83], the turbulence spectrum is implemented using the results of Ref. [84], and the bubble collision part is described by the broken power-law given in Ref. [85].

To simulate realistic LISA mock data and model the detector sensitivity, we follow the methodology outlined in Ref. [86]. The total power spectral density includes contributions from instrumental noise due to the optical metrology system and mass acceleration, assuming a standard LISA configuration with an arm length of $L = 2.5 \cdot 10^6$ km. We incorporate the full LISA response function as detailed in the same reference. For simplicity, we do not include uncertainties on the effective functions parametrizing the two noise components. In a more realistic scenario, these could be constrained by measuring in the low- and high-frequency regions where no gravitational wave signal is expected. Following the Welch method [87], the data are simulated over a frequency range from $f_{\min} = 3 \cdot 10^{-5}$ Hz to $f_{\max} = 0.5$ Hz, with a resolution of $\Delta f = 10^{-6}$ Hz, determined by the length of the time stream. To inject a signal, we generate at each frequency 94 individual signal power values with Gaussian noise and compute their average to obtain the final mock data. This mimics an expected 4-year observational run of LISA with approximately 75% observing efficiency, which results in 94 statistically independent data chunks.

To reconstruct the gravitational wave signal from the simulated LISA data, we use `evortran` to minimize a $\chi^2$-function that quantifies the difference between the model and the data, with the data including the injected signal. Specifically, we minimize

$$\chi^2(\alpha, \beta/H, T_*, g_*, v_w) = N_{\mathrm{chunks}} \sum_i \frac{1}{2} \left[ \frac{\bar{D}_i - \Omega_{\mathrm{GW}}h^2 - \Omega_s h^2}{\sigma_i} \right]^2 \,, \tag{17}$$

where $\Omega_{\mathrm{GW}}h^2$ is the template for the gravitational wave spectrum from a cosmological phase transition, see Eq. (16), depending on the five physical parameters $\alpha$, $\beta/H$, $T_*$, $G_*$, and $v_w$,

as discussed above. $\Omega_s h^2$ is the LISA face sensitivity curve, $\bar{D}_i$ are the averaged simulated signal powers at frequency bin $i$, and $\sigma_i$ are the variances over the $N_{\text{chunks}} = 94$ individual realizations that were averaged to produce the mock data (see Ref. [86] for details). The sum in Eq. (17) runs over the frequency bins from the minimum frequency $3 \cdot 10^{-5}$ Hz up to a frequency of $10^{-2}$ Hz, and the bins at higher frequency are discarded since the gravitational wave signals are far below the LISA sensitivity there.

We employ the `evolve_population` function of `evortran` to perform the optimization and locate parameter values that minimize $\chi^2$. The GA is configured with tournament selection, sbx crossover and uniform mutation. The populations size is 200, the selection size is 100, and the elite size is four. The population is evolved over a maximum number of 500 generations, or until a minimal $\chi^2$ threshold value of $1.11 \cdot 10^{-2}$ was achieved. This value was determined heuristically because we observed that below this value the instrumental noise prevents improving the signal reconstruction to a higher level of precision. For each example presented below, which differ by the injected signal or the parameter set to be reconstructed from the mock data, the minimization is repeated 200 times to explore potential variations and degeneracies in the fit. The resulting distribution of reconstructed parameter sets is then compared to the true parameters used to inject the signal, providing insight into the precision with which LISA may constrain phase transition parameters if a stochastic gravitational wave background is observed. All source code used to obtain the results presented in this section is publicly available in a dedicated Git repository [88].

Before turning to the discussion of specific example scenarios, we briefly comment on the simplifications made in this analysis. First, we do not include uncertainties in the LISA sensitivity curve, which in a full analysis would arise from limited knowledge of the noise power spectral densities and calibration uncertainties [86]. Second, we ignore stochastic astrophysical foregrounds, such as the unresolved foregrounds from stellar binary systems like white dwarfs, neutron stars or black holes [89,90], which are expected to contribute in the relevant frequency range and may complicate signal extraction. Third, we assume that the shape of the gravitational wave spectrum follows fixed template forms based on hydrodynamic simulations, rather than reconstructing the spectrum in a binned, model-independent way from the data, as considered in various LISA forecasting studies. Furthermore, we do not perform a statistically comprehensive likelihood analysis to determine allowed parameter ranges at a given confidence level. Instead, we effectively over-fit the data to provide signal reconstructions that agree well with the observations. Finally, we note that future improvements in theoretical modeling might reveal additional features in the signal templates, which could allow more precise parameter extraction if present in real data. These simplifications are justified here since our goal is to illustrate how `evortran` can be used for gravitational wave signal reconstruction as a general-purpose, customizable tool. A comprehensive treatment that includes these more realistic aspects and a more sound statistical interpretation is left for future work. For studies that address these issues in detail, see e.g. Refs. [85,86].

**Scenario 1: Reconstructing EWPT signal** – Free parameters: $\{\alpha, \beta/H, T_*, g_*, v_w\}$

As a first example, we consider a gravitational wave signal injected into the LISA mock data that is consistent with an electroweak phase transition in the early universe. The underlying parameters for this signal reflect a realistic scenario in BSM physics. We assume a transition temperature $T_* = 100$ GeV, corresponding to the order of the electroweak scale. For the transition strength we assume a value of $\alpha = 0.4$, which lies at the upper end of what can be achieved in simple scalar extensions of the Standard Model such as models with an additional singlet [91,92] and/or a second Higgs doublet [93–95], without invoking significant supercooling. The terminal velocity of the expanding bubbles is taken to be $v_w = 0.9$, close to the speed of light, since in strong transitions with $\alpha \gtrsim 0.1$ the bubble expansion is expected to proceed as relativistic detonations [96]. The number of effective relativistic degrees of free-

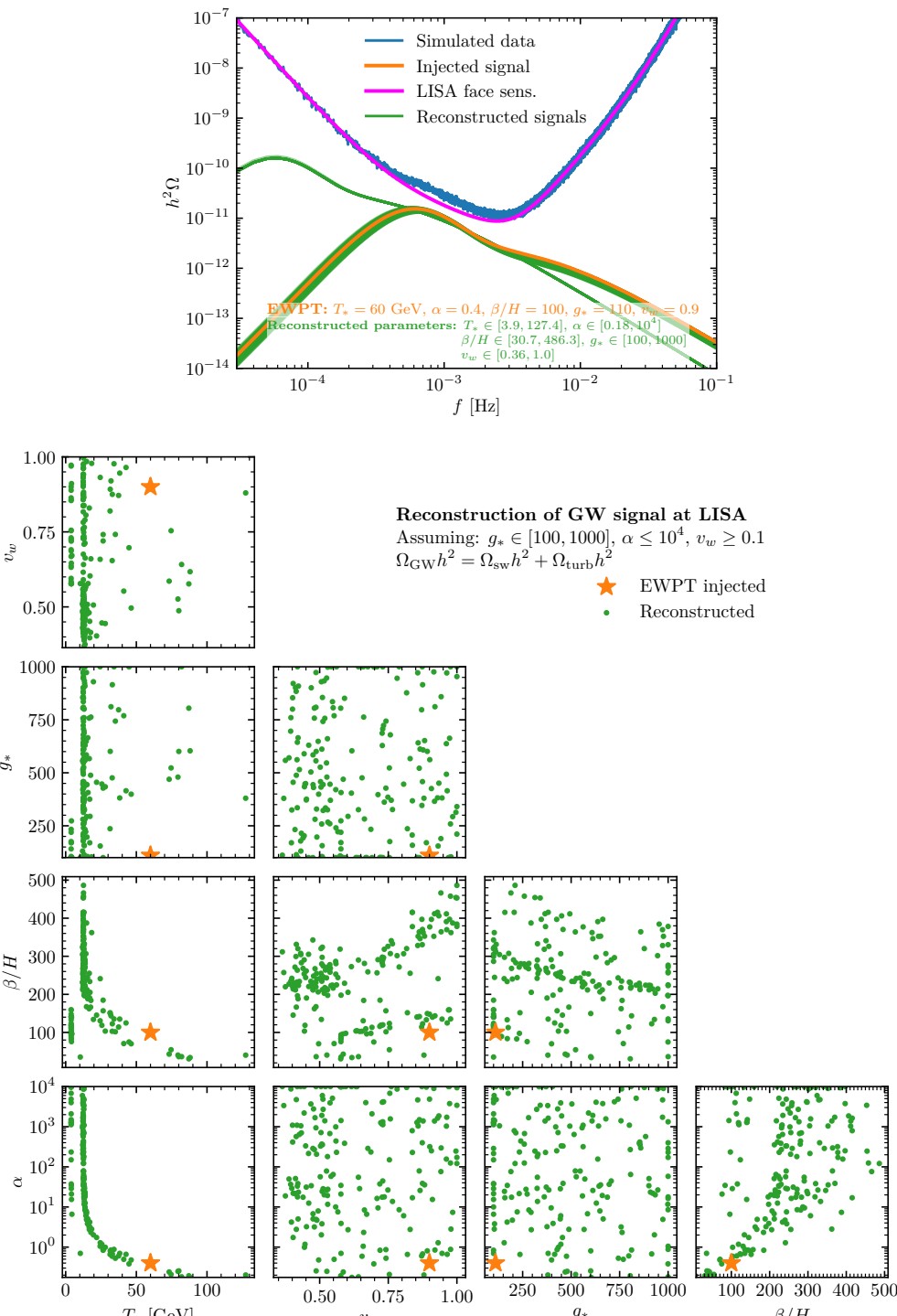

Figure 11: Top: Gravitational wave power spectrum as a function of frequency for the scenario 1. The magenta line shows the LISA power-law face sensitivity curve, the orange line is the injected gravitational wave signal composed of sound wave and turbulence contributions, and the blue line shows the mock data including Gaussian noise. The green lines represent 200 reconstructed gravitational wave signals obtained using the evortran by minimizing the $\chi^2$ function given in Eq. (17). Bottom: Corner plot showing the distributions of the phase transition parameters corresponding to the reconstructed signals. Each green point marks a parameter set that produced a signal consistent with the mock data, while the orange star indicates the parameters used to generate the injected signal.

dom is set to $g_* = 110$, accounting for the ones of the Standard Model plus a modest number of BSM states that facilitate a strong electroweak phase transition. Finally, we use $\beta/H = 100$ as a representative inverse duration of the transition, which is typical for such scenarios.

The upper plot of Fig. 11 shows the LISA face sensitivity curve (magenta), the injected gravitational wave signal (orange), and the mock data after adding Gaussian detector noise (blue). As described above, we use `evortran` to minimize the $\chi^2$ function shown in Eq. (17) in order to reconstruct the signal from this noisy data. This reconstruction is performed 200 times to explore degeneracies and noise-induced variance, where we include all five parameters $\{\alpha, \beta/H, T_*, g_*, v_w\}$ as free parameters to be fitted to the data, under the conditions that $g_* \leq 10^3$, $\alpha \leq 10^4$ and $v_w \geq 0.1$. The resulting reconstructed spectra are shown in green. For this specific scenario, the contribution from bubble collisions is omitted, as the expanding bubbles are not expected to runaway in a typical electroweak phase transition, and the resulting collision signal is expected to be subdominant compared to sound waves and turbulence in the LISA frequency band.

The plot reveals an important degeneracy in the signal reconstruction. `evortran` consistently identifies two qualitatively distinct classes of solutions that fit the data taking into account instrumental noise. The first class closely resembles the injected signal, with the peak from the sound wave contribution lying within the most sensitive frequency range of LISA. In this case, the turbulence contribution remains largely irrelevant, as its amplitude is suppressed, and its peak falls at higher frequencies where the experimental sensitivity is strongly reduced. The second class of reconstructed signals, however, fits the data using the turbulence peak instead. Here, the entire signal is shifted to lower frequencies, and although the corresponding sound wave component is much stronger, it is shifted to the left of the LISA sensitivity curve and thus effectively undetected. Despite arising from entirely different physical parameters and microphysics, these two alternative reconstructions produce a gravitational wave signal that would be indistinguishable from the injected signal within the noise level of the simulated data. This example highlights the practical challenges and degeneracies involved in interpreting gravitational wave observations from cosmological phase transitions.

In the same plot, we also show the distributions of the reconstructed parameter values corresponding to the signals that are consistent with the mock data. One can see that none of the parameters can be extracted given the available data. In particular, the parameters $\alpha$ and $g_*$ remain effectively unconstrained. Importantly, this imprecision is not solely due to the existence of the two different classes of viable signals, but also persists when considering only the subset of reconstructed signals that closely resemble the injected one. This residual uncertainty reflects inherent degeneracies in the dependence of the gravitational wave spectrum on the underlying parameters. In the future, if LISA will detect a signal consistent with an electroweak phase transition, these degeneracies might severely limit the possibility of distinguishing between different BSM theories that might have given rise to the electroweak phase transition.

The lower plot in Fig. 11 is a corner plot showing the parameter distributions of the 200 reconstructed signals. Each green point corresponds to a parameter combination that yields a signal compatible with the mock data (green lines in the top plot), while the orange star indicates the true values of the injected signal (orange line in the top plot). The corner plot shows the presence of strong degeneracies, as the green points are broadly scattered, filling large portions of the allowed parameter space across all pairwise projections. This behavior does not indicate a shortcoming of the applied GA or a lack of convergence. The reconstruction problem in the chosen parameter basis (called "thermodynamical" parameters) is known to exhibit strong parameter degeneracies [85], which prevent precise constraints regardless of the sampling technique employed. In particular, a prominent degeneracy is visible between the parameters $\alpha$, $\beta/H$ and $T_*$. This degeneracy will be further analyzed and discussed in the

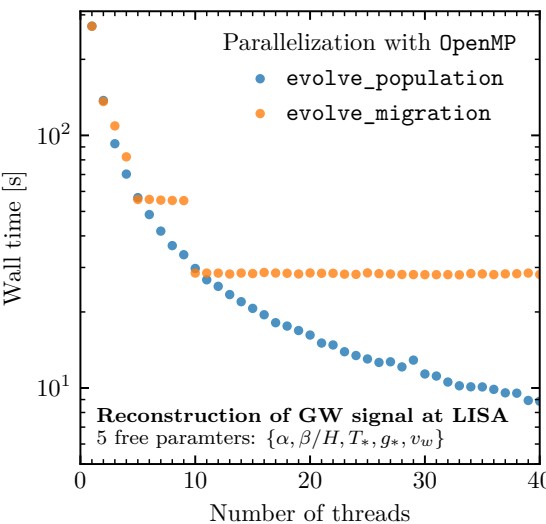

Figure 12: Wall time as a function of the number of threads for the scenario 1, using the `evolve_population` (blue points) and `evolve_migration` (orange points) routines.

next example.

To study the performance of the parallel implementation of `evortran` in a realistic setting (the scaling was studied for the minimization of the Rastrigin benchmark function in Section 4.1.1) with a computationally expensive fitness function, we analyzed the scaling of wall time with the number of threads in this LISA signal reconstruction example. The results are shown in Fig. 12. For both the `evolve_population` (blue points) and the `evolve_migration` (orange points) routines, the minimization of the $\chi^2$ function was performed here without setting a `fitness_target` to ensure that all generations were executed. In the case of `evolve_population`, a population size of 1000, a maximum of 250 generations, and a selection size of 1000 were used. For `evolve_migration`, ten populations with 100 individuals each were evolved over five epochs of up to 50 generations, resulting in the same total number of individuals and total number of generations. Moreover, the selection size was set equal to the population size. This setup yields similar wall times for both routines when using a single thread, although it leads to some overfitting in this scenario. The configuration was chosen deliberately to increase the total runtime so that the scaling with thread count remains visible before reaching the regime where algorithmic overhead dominates.

One can see in Fig. 12 that the `evolve_population` routine exhibits a significant reduction of the wall time with thread count. Starting from almost 300 seconds for a single thread, the wall time roughly halves when using two threads, decreases to about 5 seconds with five threads, and continues to improve gradually with even larger numbers of threads, dropping below one second at 30 threads. With the maximum of 40 threads, the wall time reaches about 0.9 seconds, corresponding to an overall speedup by a factor of more than 300. The `evolve_migration` shows similar improvement up to about five threads, where the wall time decreases to roughly 5 seconds. Then the wall time stagnates until ten threads, where the wall time improves the last time, reaching approximately 3 seconds, corresponding to a speedup of about two orders of magnitude relative to a run with only one thread. For even larger number of threads, no further improvement of the runtime is achieved. The better scaling of the `evolve_population` routine in this example compared to `evolve_migration` is the opposite of what was observed for the optimization of the Rastrigin function discussed in Section 4.1.1, see Fig. 2, where the `evolve_migration` routine performed significantly bet-

ter using parallel execution. The important difference here is the higher computational cost of the fitness function which makes the parallelization over individuals within a population, as implemented in the `evolve_population` routine, highly effective. In contrast, the parallelization implemented in the `evolve_migration` over the different populations is limited here by the small number of only five simultaneously evolving populations, such that using significantly more than about five threads provides no additional improvement in runtime.

**Scenario 2: Reconstructing EWPT signal assuming $\alpha \leq 1$ and $g_* = 110$ – Free parameters: $\{\alpha, \beta/H, T_*, v_w\}$**
In the second example, we build upon the previous analysis by introducing additional constraints that reflect a more model-dependent interpretation of the gravitational wave signal. Specifically, we restrict the strength of the phase transition to values $\alpha \leq 1$, and we fix the effective number of relativistic degrees of freedom to $g_* = 110$. These choices are motivated by the expectation that an electroweak phase transition occurring in minimal extensions of the Standard Model, such as those with a singlet scalar, a second Higgs doublet, or a Higgs triplet, will involve only a modest increase in the particle content and will not exhibit significant supercooling. As a result, this setup provides a more focused reconstruction of the signal under the assumption that the underlying physics corresponds to a specific electroweak-scale model. In contrast, the broader parameter space explored in the previous example remains more agnostic to the origin of the signal and is also compatible with other cosmological phase transitions that might have occurred in the early universe. We again take into account only the sound wave and turbulence contribution to the gravitational wave signal, which is consistent with the considered range of $\alpha$.

The results of this second example are shown in Fig. 13, with the top plot displaying the power spectra and the bottom plot presenting the corner plot of reconstructed parameter values. Due to the additional assumptions on $\alpha$ and $g_*$, only reconstructed signals that closely resemble the injected one are found. The second class of solutions observed in the previous example, characterized by a peak at lower frequencies dominated by the turbulence contribution as visible in the upper plot of Fig. 11, disappears. This is a consequence of requiring $\alpha \leq 1$ here. As a result of the additional assumptions, the parameter reconstruction becomes more useful. From the distribution of reconstructed signals, we infer that the transition temperature $T_*$ is constrained between about 24 GeV and 154 GeV, $\alpha$ must be larger than about 0.1, $\beta/H$ lies between 27 and 223, and the bubble wall velocity $v_w$ is reconstructed to be above about 0.4. Here one should note that the lower limit on $T_*$ and the upper bound on $\beta/H$ are both consequences of the upper limit assumed on $\alpha$, and thus not a direct consequence of the fitting procedure.

The corner plot at the bottom of Fig. 13 further reveals that the bubble wall velocity $v_w$ remains practically unconstrained. However, fixing $g_* = 110$ gives rise to a clearer correlation between the other three parameters $T_*$, $\alpha$ and $\beta/H$, with $\alpha$ and $\beta/H$ decreasing with increasing values of $T_*$, see the two bottom panes in the left column of the corner plot. The size of the bands in which the green points are concentrated in these plots results from the unknown bubble wall velocity $v_w$ which is also fitted to the data in this example. In the following example we will investigate the correlations between the parameters by further assuming that a prediction for $v_w$ is available for the electroweak phase transition, in which case $v_w$ does not have to be reconstructed from the LISA data but can be set to the predicted value.

**Scenario 3: Reconstructing EWPT signal assuming $\alpha \leq 1$, $g_* = 110$ and $v_w = 0.9$ – Free parameters: $\{\alpha, \beta/H, T_*\}$**
The third example builds upon the second scenario with an even more constrained fit. In addition to assuming $\alpha \leq 1$ and fixing the effective number of degrees of freedom to $g_* = 110$, we now also fix the bubble wall velocity to $v_w = 0.9$, motivated by the expectation of relativistic expansion for phase transitions with $\mathcal{O}(0.1\text{--}1)$ values of $\alpha$. This setup reflects a more model-

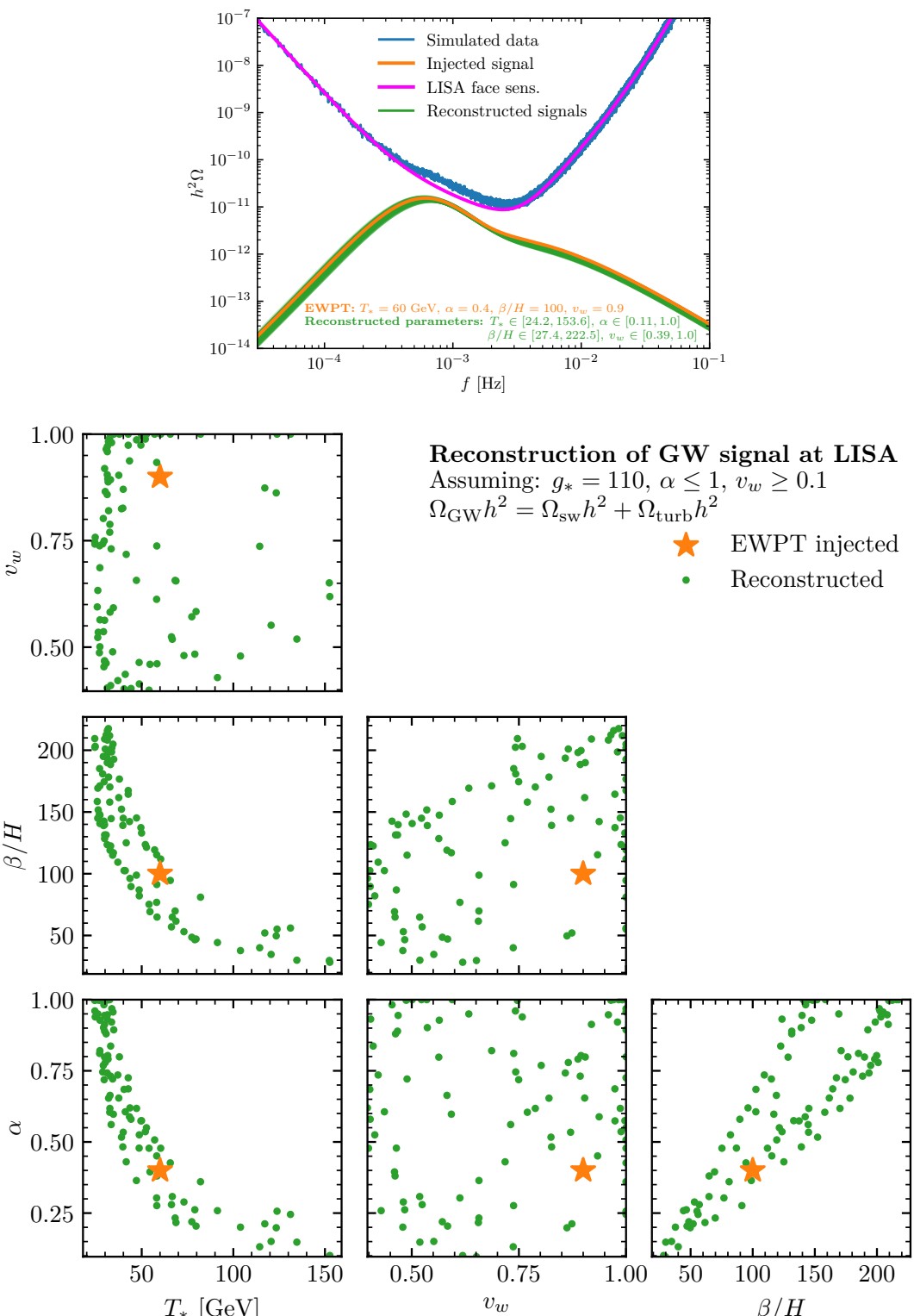

Figure 13: As in Fig. 11 for the scenario 2.

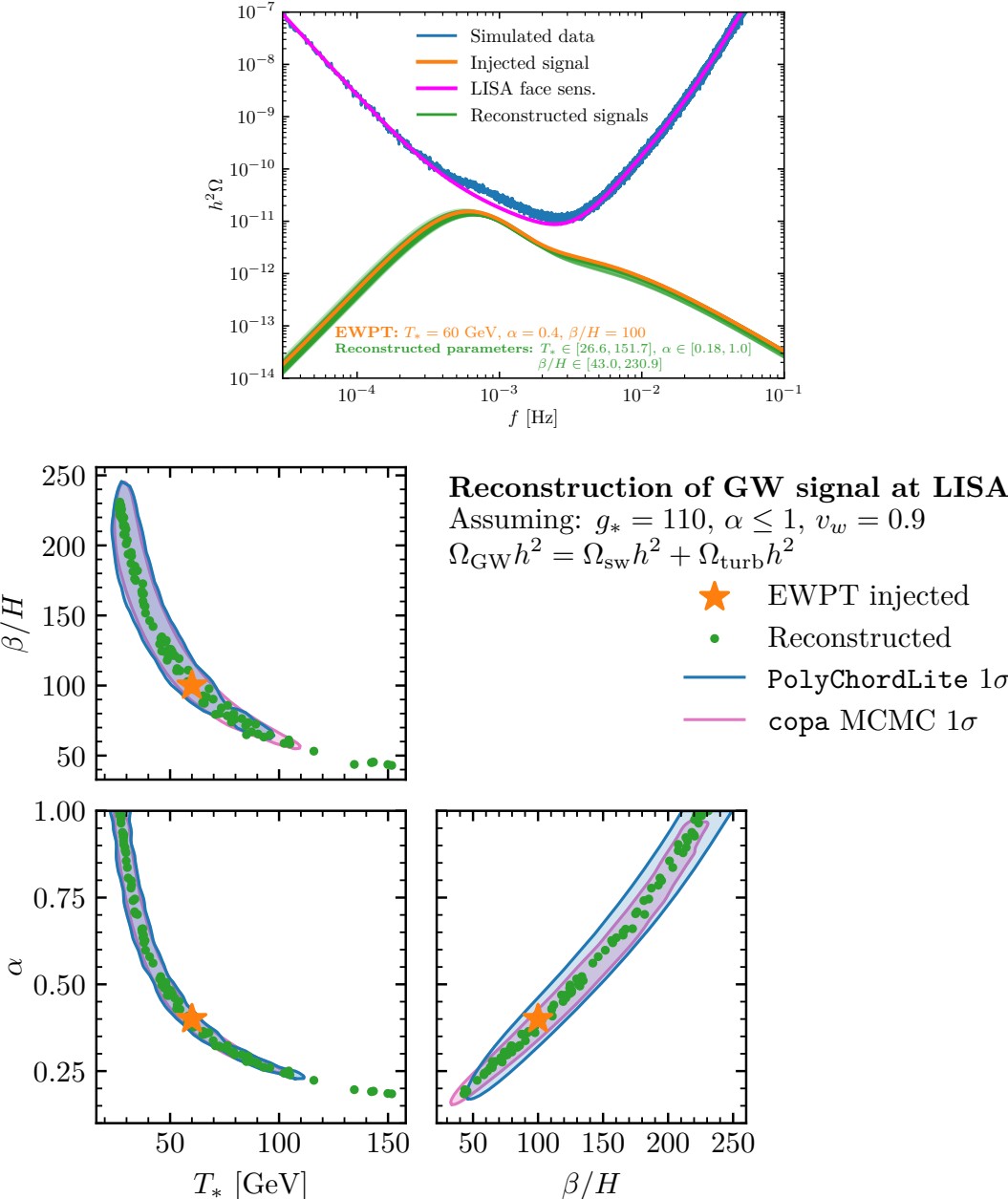

Figure 14: As in Fig. 11 for the scenarion 3. The corner plot additionally shows the $1\sigma$ confidence level credible regions obtained with two independent Bayesian inference methods: nested sampling using `PolyChordLite` (blue shaded regions) and an ensemble Markov Chain Monte Carlo sampler implemented using `copa` (purple shaded regions).

dependent interpretation in which we assume that in the future when LISA is in operation it may be possible to reliably compute the bubble wall velocity from first principles in a given BSM scenario. Consequently, the reconstruction is now performed only for the three parameters $\alpha$, $\beta/H$ and $T_*$. While these additional assumptions reduce generality, they significantly enhance the precision of the parameter reconstruction, as we will demonstrate. The injected signal parameters remain the same as in the previous two examples discussed above, and we again only consider the sound wave and turbulence contribution to the gravitational wave signal.

In Fig. 14 we show the results for this third example, again showing the reconstructed power spectra in the top plot and the reconstructed values of the fitted parameters in the corner plot at the bottom. The upper plot illustrates that the signal is now reconstructed with high precision across the frequency range relevant for LISA. However, despite the overall good match between the reconstructed signals and the injected one, the underlying parameter values corresponding to the injected signal are still not recovered. This is a result of the degeneracy between the three free parameters $\alpha$, $\beta/H$ and $T_*$. As shown in the corner plot, the viable parameter values consistent with the data lie along thin lines in the parameter space. This indicates that a detection of a stochastic signal with LISA could effectively be used to express two of the parameters as functions of the third.

While the observed degeneracy between the parameters still prevents precise extraction of the parameter values of the injected signal, the reconstruction becomes significantly more informative. In particular, it enables meaningful model discrimination power. If a given BSM theory can predict a set of values for $\alpha$, $\beta/H$ and $T_*$ that fall on top of the reconstructed lines in the corner plot, that model remains viable within experimental uncertainty and could provide an explanation for the detected signal. Conversely, if the predicted parameter correlations in a specific BSM theory fall outside the reconstructed viable region, not overlapping with the lines in the corner plot, the model could potentially be ruled out. It is important to note, however, that this type of model testing hinges on the assumption that the bubble wall velocity $v_w$ is known and fixed. If a precise prediction for $v_w$ is not available in the future when LISA is in operation, the discrimination power between different models is significantly worse, see the scenarios discussed above.

In addition to the reconstructed points obtained with evortran, the corner plot in Fig. 14 also shows the $1\sigma$ credible regions derived from Bayesian parameter inference, obtained using nested sampling with the Fortran library PolyChordLite [97,98] (blue shaded regions) and an ensemble Markov Chain Monte Carlo sampler implemented in the fpm project copa [99] (purple shaded regions). The chains produced by both the ensemble MCMC sampler and the nested sampler were processed using the Python package GetDist [100] to determine the $1\sigma$ credible regions, making use of kernel density estimation to construct smooth posterior distributions. The comparison of the signal reconstruction using evortran with with Bayesian samplers is carried out only for this example scenario, since the previous two scenarios exhibit multiple degeneracies in the parameter space, which make a meaningful construction of credible regions difficult. In contrast, in the present scenario, the degeneracy between the three free parameters $\alpha$, $\beta/H$, and $T_*$ is confined to a single, relatively well-defined direction in parameter space, allowing for a more direct comparison between the results obtained with evortran and the statistically inferred credible regions from PolyChordLite and copa, respectively.

In the comparison between the results from evortran against the ones from PolyChord and copa, it is important to keep in mind that GAs and Bayesian sampling methods serve complementary but distinct purposes. GAs are designed to efficiently identify the best-fit solutions that minimize the $\chi^2$ function, whereas Bayesian sampling explores the posterior probability distribution around these solutions, providing statistically meaningful confidence regions that quantify parameter uncertainties. However, sampling methods can struggle to locate the global best-fit solutions if the likelihood landscape is sufficiently complex or if some optima

appear isolated from the others in the analyzed parameter space. In such cases, GAs provide an ideal tool to verify whether the samplers have identified all viable regions of parameter space.

One can see that most of the reconstructed parameter points obtained with `evortran` lie within the $1\sigma$ credible regions, and both the `PolyChordLite` and `copa` results agree very well with each other. The strong degeneracy among the parameters $\alpha$, $\beta/H$, and $T_*$ is visible across all methods, confirming that it is an intrinsic feature of the reconstruction problem rather than a limitation of the GA optimization or the sampling methods. A few of the `evortran` points at low transition temperatures and transition strengths appear outside of the $1\sigma$ credible regions. This behavior arises because the credible regions from the Bayesian analyses were constructed using uniform sampling, which, combined with finite resolution, can artificially truncate the regions at the lowest parameter values along the flat direction in the $\chi^2$ function. The fact that `evortran`, where also uniform initialization of the initial population was employed, identifies viable solutions in this region highlights this limitation and demonstrates the advantage of combining GA-based reconstructions with Bayesian sampling. A GA efficiently explores the global parameter space more exploratory and can expose paramter regions that may be undersampled in a statistical analysis. However, also the results of GAs depend on the priors that are used to initialize the population. The impact of different prior choices on the inferred parameter regions is discussed in more detail in the following example.

**Scenario 4: Reconstructing stronger signal assuming $\alpha \leq 10^3$, $g_* \leq 150$ and $v_w = 1$** – Free parameters: $\{\alpha, \beta/H, T_*, g_*\}$
In this fourth and final scenario, we return to a more general setup and consider a substantially stronger gravitational wave signal originating from a cosmological phase transition with a large strength parameter of $\alpha = 45$. The motivation behind this choice is to investigate how well the signal reconstruction and parameter inference is improved when the injected signal has a significantly higher signal-to-noise ratio, thereby enhancing its detectability across a wider range in the LISA frequency band. In this example, the contribution from bubble collisions might not be negligible, and we therefore include this third source component in the signal template. The bubble collision peak appears at slightly lower frequencies than the sound wave and turbulence contributions, providing additional structure in the spectrum that could potentially help break some of the degeneracies observed in earlier examples.

A key focus of this example is to explore the influence of the prior distribution used for the reconstructed parameters. In particular the sampled range of the strength parameter $\alpha$ span several orders of magnitude. To investigate the impact of different initialization of gene values in a GA, we perform two separate reconstructions: one using a linear prior on $\alpha$ and the other using a logarithmic prior. The comparison between the resulting distributions of reconstructed parameters highlights that the performance and coverage of the solution space of GAs can depend sensitively on the initial seeding of gene values, i.e. on how the sampling space is explored from the start. As this example will demonstrate, the choice of prior can have significant consequences for the robustness and reliability of the inferred parameter ranges in multi-scale parameter spaces.

In Fig. 15 we show the results for the fourth scenario, where a strong gravitational wave signal is injected with parameters $T_* = 100$ GeV, $\alpha = 45$, $\beta/H = 100$ $g_* = 110$, and $v_w = 1$. The large signal-to-noise ratio allows for a very precise reconstruction of the spectral shape, as visible in the top plot, with the sound wave peak placed near the maximum sensitivity of LISA and the turbulence and collision peaks located to either side but still within the sensitive band. In the corner plot below, we show two sets of reconstructed parameter values. The green points correspond to a linear prior on $\alpha$, while purple points come from using a log prior. The linear prior leads to a more concentrated reconstruction, especially for $g_*$, as visible in the upper pane of the corner plot. This might misleadingly suggest that this parameter

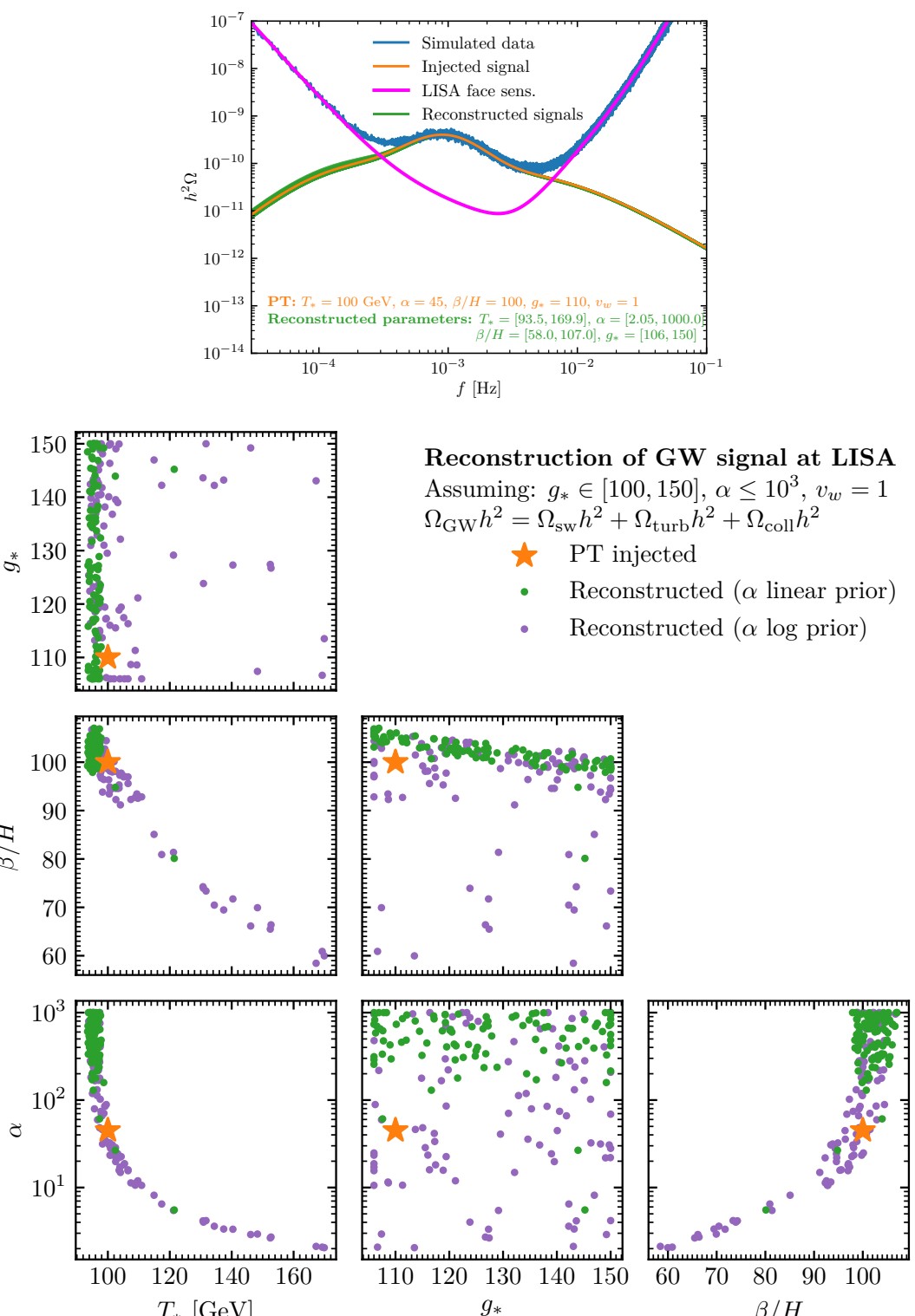

Figure 15: Same as in Fig. 11 for the scenario 4. The corner plot additionally shows reconstructed parameter values using a logarithmic prior on the parameter $\alpha$ at initialization of the GA.

is well constrained by the data. However, the results using the log prior reveal that a wide range of values, including the whole sampled range of $g_*$, are actually compatible with the injected signal. This discrepancy arises because the linear prior oversamples values of $\alpha$ at the upper end of the sampled interval, whereas the log prior allows the GA to explore several orders of magnitude more evenly as visible in the bottom row of the corner plot. Notably, only the log prior reconstruction recovers the injected parameter values (orange stars), while the linear prior biases the fit toward larger $\alpha$ and $\beta/H$ values. This example illustrates the importance of prior choice in GA-based inference, especially when parameters span several orders of magnitude.

## 5  Conclusions

We have introduced `evortran`, a lightweight, flexible, and efficient genetic algorithm (GA) library written in modern Fortran. The `evortran` package is available at:

[https://gitlab.com/thomas.biekoetter/evortran](https://gitlab.com/thomas.biekoetter/evortran).

With its modular design and simple user interface, `evortran` enables users to easily apply evolutionary strategies to complex optimization problems, including those with non-differentiable, discontinuous, or noisy fitness functions. The library supports customization of GA components, native real and integer encodings, and parallel execution via `OpenMP`. `evortran` is installed with the Fortran package manager `fpm`, ensuring a straightforward dependency management, compilation and installation process, as well as a seamless integration into both simple scripts and larger code bases. To further enhance accessibility, `evortran` provides Python bindings available at:

[https://gitlab.com/thomas.biekoetter/pyevortran](https://gitlab.com/thomas.biekoetter/pyevortran).

This interface allows users to run the core optimization routines of `evortran` directly from Python.

To demonstrate its robustness and versatility, we first validated `evortran` on a set of well-known multi-modal benchmark functions commonly used in global optimization, showing reliable convergence and the ability to locate global optima even in rugged fitness landscapes. As a complex, real-world application from particle physics, we used `evortran` to perform high-dimensional parameter scans of the Singlet-extended Two-Higgs-Doublet Model (S2HDM), involving eleven to 14 free parameters, a combination of theoretical constraints, and an extensive set of LHC data. As a second physics application from cosmology, we then applied the library to the reconstruction of primordial gravitational wave signals and their underlying parameters from mock data of the upcoming LISA space observatory. In both cases, `evortran` performed successfully, identifying viable solutions efficiently in challenging search spaces.

While our focus here has been on specific scientific use cases, the design of `evortran` makes it broadly applicable to a wide range of optimization problems in physics, engineering, and other fields requiring global search strategies. We hope that `evortran` becomes a useful tool for GAs in the Fortran ecosystem and scientific computing.

## Acknowledgements

I gratefully acknowledge the open-source Fortran community for their continued efforts in developing and modernizing the Fortran ecosystem, with special thanks to the contributors of the Fortran Package Manager (FPM), whose work was beneficial for the development of `evortran`.

**Funding information** The project that gave rise to these results received the support of a fellowship from the "la Caixa" Foundation (ID 100010434). The fellowship code is LCF/BQ/PI24/12040018. We acknowledge the support of the Spanish Agencia Estatal de Investigación through the grant "IFT Centro de Excelencia Severo Ochoa CEX2020-001007-S".

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
