# Peer review of "evortran: a modern Fortran package for genetic algorithms with applications from LHC data fitting to LISA signal reconstruction"

_SciPost Physics Codebases_

## Round 1 · Referee Report · Anonymous (Referee 2) · 2025-10-3

Strengths

1) The author explains the trade-offs between debug and performance modes (exports_debug.sh vs. exports_run.sh) and the option to compile in different precision: double or quadruple. Such a discussion is valuable for reproducibility and performance benchmarking.

2) Benchmarking with test functions The Rastrigin and Michalewicz benchmarks demonstrate that the implementation is efficient and robust on complex landscapes.

3) Validation with physics-motivated problems The library is tested not only on toy benchmarks but also on realistic physics cases: • A high-dimensional BSM Higgs-sector fit. • Reconstruction of gravitational wave spectrum from mock LISA data. This demonstrates direct relevance for cosmology and gravitational-wave physics.

4) Parallelization design with OpenMP The authors emphasize how OpenMP parallelization is applied both at the population level (evolve_population) and across multiple populations (evolve_migration) - explaining how the library can scale across CPU cores, useful for simulation-heavy problems like LISA data analysis.

Weaknesses

1) Simplistic treatment of LISA data analysis The author acknowledges neglecting confounding sources of errors introduced by the galactic binary foregrounds, stellar mass black holes, etc as well as assumptions made about the LISA noise power spectrum and the gravitational wave spectrum. That said, these are not necessary points for improvement in this manuscript, as the approach is presented more as a proof of concept rather than a ready-to-use inference tool.

2) Lack of GPU support or distributed-memory parallelization The library is restricted to OpenMP (shared memory). For LISA, which will require HPC-scale inference, MPI or GPU support might be beneficial. This limitation is might benefit a broader discussion.

3) Limited quantitative performance benchmarks While OpenMP parallelization is described, there is no systematic scaling study (e.g., runtime vs. number of cores, memory usage). For LISA-scale problems, where millions of evaluations are required, a deeper discussion would be useful.

Report

The manuscript presents evortran, a Fortran library for genetic algorithms designed for high-performance scientific optimization tasks. The package is built to be modular, flexible, and parallelizable. It includes OpenMP-based parallel execution and provides Python bindings for integration into broader workflows. To this end, the author discusses uses for LHC and LISA data analysis. Overall, it aims to offer a fast GA framework that provides Fortran performance with Python accessibility.

Requested changes

The paper would benefit from a deeper discussion into:

1) Support for high performace computing resources like GPU/MPI. 2) Quantitative benchmark of runtime, number of cpus and and memory usage in comparison to other inference techniques (likelihood based and simulation-based inference)

Recommendation

Publish (meets expectations and criteria for this Journal)

  • validity: high
  • significance: good
  • originality: high
  • clarity: high
  • formatting: good
  • grammar: excellent

Author:  Thomas Biekötter  on 2025-11-13  [id 6028]

(in reply to Report 2 on 2025-10-03)

We thank the referee for their careful review of our manuscript. Below we comment on the issues raised by the referee, and we discuss the corresponding changes to the manuscript.

The referee writes:

1) Simplistic treatment of LISA data analysis The author acknowledges neglecting confounding sources of errors introduced by the galactic binary foregrounds, stellar mass black holes, etc as well as assumptions made about the LISA noise power spectrum and the gravitational wave spectrum. That said, these are not necessary points for improvement in this manuscript, as the approach is presented more as a proof of concept rather than a ready-to-use inference tool.

Our response:

We agree that the example study of LISA mock data is simplistic as it disregards several sources of complifications. We think that this is extensively discussed in the paper in the third paragraph on page 56 (lines 2291-2309). We wish to stress that the discussion in section 4.2.2 should be regarded as a demonstration of the potential of evortran to navigate complicated and computationally demanding function landscapes. A fully fledged statistical analysis of LISA signal recunstructions and parameter inference, including additional background and foreground sources of gravitational waves, would go beyond the scope of this paper.

The referee writes:

2) Lack of GPU support or distributed-memory parallelization The library is restricted to OpenMP (shared memory). For LISA, which will require HPC-scale inference, MPI or GPU support might be beneficial. This limitation is might benefit a broader discussion.

Our response:

We thank the referee for this valuable remark. In response, we have added a discussion in a new paragraph at the end of the OpenMP parallelization section (lines 1146-1161) addressing the limitations of shared-memory parallelism and the potential benefits of distributed-memory (MPI) and GPU-based extensions. The new paragraph explains that the current OpenMP implementation in evortran provides efficient and portable shared-memory parallelization across multi-core CPUs, which is suitable for typical workstation and server environments. There we discuss now that for larger-scale applications, future versions of evortran could incorporate MPI or GPU acceleration. We further comment that the effectiveness of GPU acceleration depends on the structure of the fitness function and, in some cases, the entire population-level fitness evaluation could be offloaded to GPUs if it can be formulated as a matrix operation.

The referee writes:

3) Limited quantitative performance benchmarks. While OpenMP parallelization is described, there is no systematic scaling study (e.g., runtime vs. number of cores, memory usage). For LISA-scale problems, where millions of evaluations are required, a deeper discussion would be useful.

Our response:

We thank the referee for raising this point. In response to the remark by the referee, we added two new plots and corresponding discussions in which the performance improvements (in terms of the wall time) of the GAs as a function of the number of CPU threads is analyzed. These additional discussions serve as a demonstration of the impact of parallelization across different optimization problems. The first parallelization benchmark was added to section 4.1.1 about the minimization of the Rastrigin function, including the new figure 2 and an additional paragraph at the end of the section (lines 1645-1669). The second benchmark was added to the section 4.2.2 about the reconstruction of a stochastic graviational wave signal at LISA, including the new figure 12 and two new paragraphs at the end of the discussion of the scenario 1 on pages 59-60 (lines 2372-2406).

Regarding the study of memory usage mentioned by the referee, we note that the intrinsic memory footprint of the GA implementation is small and scales primarily with the number of individuals and gene parameters, while substantial memory usage may generally arise only from the user-defined fitness function. We therefore focused our benchmark on runtime and scaling with the number of CPU threads as discussed above, as this better reflects the performance of the library itself. On the other hand, for typical use cases where genes are represented by arrays of floating-point or integer numbers, the memory overhead of GAs remain negligible unless extremely large population sizes are employed. For these reasons, we do not consider it necessary to include a detailed discussion of memory usage in the manuscript.

The referee writes:

The paper would benefit from a deeper discussion into: 1) Support for high performace computing resources like GPU/MPI. 2) Quantitative benchmark of runtime, number of cpus and and memory usage in comparison to other inference techniques (likelihood based and simulation-based inference)

Our response:

We thank the referee for this constructive suggestion. Regarding point (1), we have added a discussion on potential support for high-performance computing resources such as GPU and MPI parallelization at the end of the OpenMP section, as detailed in our response to the related comment above. The added discussion explains the current design choice of shared-memory OpenMP parallelization in evortran, as well as possible future extensions to GPU-based implementations.

Regarding point (2), we have now included a comparison with likelihood-based inference methods in section 4.2.2, where we discuss the reconstruction of gravitational wave signals from LISA mock data. In the discussion of the scenario 3, we compare the results obtained with \texttt{evortran} to the 1σ credible regions derived from nested sampling using the Fortran library \texttt{PolyChordLite} and from ensemble Markov Chain Monte Carlo sampling with the FPM project \texttt{copa}. These regions were added to the corner plot in figure 14. We find overall good agreement between the different methods. The added credible regions and a comparison to the reconstructed signals using the GA are discussed in three new paragraphs at the end of the discussion of the scenario 3 (lines 2480-2519).

We would like to emphasize that this comparison is intended to illustrate the complementary nature of the methods rather than to perform a quantitative performance benchmark. Likelihood-based samplers such as MCMC and nested sampling aim to explore the full posterior probability distribution and provide statistically interpretable confidence regions, whereas GAs, as implemented here with evortran, are optimization tools designed to locate global best-fit solutions efficiently, even in complex or multimodal parameter spaces. The main purpose of this paper is to present \texttt{evortran} as a flexible, general-purpose Fortran library for implementing GAs, rather than to conduct a broad methodological comparison or establish performance rankings relative to other inference techniques.

Additional changes to the manuscript

1) We added footnote 5 on page 26.

We thank again the referee for their helpful remarks and hope that the revised paper can now be published in SciPost.

---

## Round 1 · Referee Report · Anonymous (Referee 1) · 2025-10-3

Strengths

-The paper is well-written, and the description of the algorithm is clear.
-The code is highly modular.

Weaknesses

There is no effective benchmarking of computational cost, and no comparison with other methods, which raises doubts about its practical applicability to realistic problems, e.g., in the LISA context.

Report

In this manuscript, the author presents a library that performs optimization via a genetic algorithm. The design of the code is clearly described throughout the manuscript. Furthermore, the code is easily installable and user-friendly, and highly flexible . The authors present two possible applications: a global fit of the Higgs sector in a BSM framework, and the reconstruction of the cosmological stochastic gravitational wave background in the LISA context. Concerning the second scenario, the author claims a successful solution, but this is not entirely clear to me.

Overall, I am happy to recommend the publication for SciPost Physics Codebases, with the following corrections and suggestions:

-I suspect that in line 1522 the correct command is: source exports_debug.sh.

-For the Python wrapper, I suggest running the following command before installation: 'python -m pip install --upgrade pip setuptools wheel' These are crucial tools for building and installing Python packages. If they are outdated, certain packages may fail to install. -For the Python wrapper, I recommend not labeling a folder as python within your project directory (i.e., python/pyevortran), as this can lead to issues with Python packaging and module imports. -Lines 2216–2222 appear to describe the construction of the Welch spectrum. I suggest avoiding this convoluted explanation and instead mentioning it briefly with an appropriate reference. -The two examples should be fully reproducible. I did not find a script in the repository; I suggest adding one. - As mentioned earlier, I am concerned about the application of evortran in the LISA context. Even if the code reaches convergence, the resulting parameter estimates appear to be poorly constrained. Traditional sampling techniques or alternative machine learning approaches (e.g., normalizing flows) seem to be more efficient and precise, at least in this scenario, especially considering the simplified assumptions stated by the author. Given this, it would be helpful if the author could provide a more detailed assessment of the algorithm’s quality and reliability in this context. -Furthermore, there is no benchmark regarding the speed of the code, computational resources, or overall computational cost. Including such information would greatly improve the manuscript and help potential users evaluate the practicality of the library.

Recommendation

Ask for minor revision

  • validity: good
  • significance: ok
  • originality: good
  • clarity: good
  • formatting: good
  • grammar: excellent

Author:  Thomas Biekötter  on 2025-11-13  [id 6027]

(in reply to Report 1 on 2025-10-03)

We thank the referee for their careful review of our manuscript. Below we comment on the issues raised by the referee, and we discuss the corresponding changes to the manuscript.

The referee writes:

I suspect that in line 1522 the correct command is: source exports_debug.sh.

Our response:

We thank the referee for spotting this typo. We fixed it in the revised manuscript.

The referee writes:

For the Python wrapper, I suggest running the following command before installation: 'python -m pip install --upgrade pip setuptools wheel' These are crucial tools for building and installing Python packages. If they are outdated, certain packages may fail to install.

Our response:

We have added the footnote 9 in which we provide the command suggested by the referee to update the Python enviornment.

The referee writes:

For the Python wrapper, I recommend not labeling a folder as python within your project directory (i.e., python/pyevortran), as this can lead to issues with Python packaging and module imports.

Our response:

We deliberately use a top-level python/ subfolder to separate the Python interface (pyevortran) from the Fortran library (evortran) because the Python package requires a pyproject.toml file. Placing pyproject.toml at the repository root would conflict with the existing fpm.toml used for building the Fortran library, as both tools create a build/ folder and would interfere with each other.

Within the python/ folder, the actual Python package is contained in the pyevortran/ directory. Therefore, the folder name python/ does not affect the Python package namespace. Importing pyevortran works as usual via 'import pyevortran', and all standard Python packaging and installation tools are supported.

We believe this structure provides a clear separation between the Fortran and Python components while avoiding any technical or packaging issues, and we therefore intend to keep this organization.

The referee writes:

Lines 2216–2222 appear to describe the construction of the Welch spectrum. I suggest avoiding this convoluted explanation and instead mentioning it briefly with an appropriate reference.

Our response:

We thank the referee for the suggestion regarding the description of the Welch spectrum. We have added the phrase "Following the Welch method, ..." at the beginning of the relevant paragraph and included an appropriate reference. We have otherwise retained the original description, since the current level of detail clarifies the procedure for readers who may not be familiar with the construction of mock LISA power spectra.

The referee writes:

The two examples should be fully reproducible. I did not find a script in the repository; I suggest adding one.

Our response:

The two examples discussed in Section 4.2.1 and 4.2.2, respectively, which we think the referee is referring to, rely on additional pieces of software that are external to evortran. Therefore, the corresponding scripts have not been included in the evortran repository.

Following the suggestion by the referee, we have created independent repositories for these example in which evortran is used as external dependency together with the other tools that are required to produce the results shown in the paper.

The scripts that produce the results discussed in Section 4.2.1 are now publicly available under the following link: https://gitlab.com/thomas.biekoetter/evortran_s2hdm_fit To run these scripts, in addition to evortran, one has to install the public program cs2hdmtools (including its dependencies HiggsTools and micrOMEGAs).

In the manuscript, we added a sentence on page 49 (lines 2012-2013) in which we provide the link to this repository.

The scripts that produce the results discussed in Section 4.2.2 are now publicly available under the following link: https://gitlab.com/thomas.biekoetter/evortran_lisa_fit To run these scripts, an additional FPM project called gwlisa is defined which so far was an in-house tool to generate and analyze LISA mock data. It contains, among other things, routines to generate the LISA mock data and an implementation of GW templates based on the LISA recommendations.

In the manuscript, we added a sentence on page 56 (lines 2289-2290) in which we provide the link to this repository.

The referee writes:

As mentioned earlier, I am concerned about the application of evortran in the LISA context. Even if the code reaches convergence, the resulting parameter estimates appear to be poorly constrained. Traditional sampling techniques or alternative machine learning approaches (e.g., normalizing flows) seem to be more efficient and precise, at least in this scenario, especially considering the simplified assumptions stated by the author. Given this, it would be helpful if the author could provide a more detailed assessment of the algorithm’s quality and reliability in this context.

Our response:

We agree that the parameter reconstructions in the LISA examples show that the obtained fits only weakly (or not at all) constrain the parameter values. However, this behavior does not indicate a shortcoming of the applied GAs or a lack of convergence. The reconstruction problem in the chosen parameter basis (called "thermodynamical" parameters) is known to exhibit strong parameter degeneracies, which prevent precise constraints regardless of the sampling technique employed. This parameter basis featuring degeneracies was intentionally chosen to demonstrate that GAs can successfully operate in situations where the underlying likelihood or chi-squared landscape contains extended flat directions. In this sense, the ability of the GA to identify optimal solutions along such directions demonstrates its usefulness in exploring such scenarios where other optimization and sampling methods often struggle. In the paper, we added a sentence in the last paragraph on page 58 (lines 2365-2369) in which we stress that the poor reconstruction of the parameters is not a consequence of a lack of convergence, but results from parameter degeneracies intrinsic to the problem at hand.

To address the comment of the referee regarding the potential efficiency of alternative sampling techniques, we have added a comparison in scenario 3 in section 4.2.2 using a nested sampler (PolyChordLite) and an ensemble MCMC sampler (copa), see the updated corner plot in figure 14. As discussed in our response to the other referee, these comparisons confirm that the same parameter degeneracies observed with evortran are also present when using these alternative methods. This reinforces that the poor reconstruction of the parameters is intrinsic to the chosen parameter basis, rather than a limitation of the GA approach. We have updated the discussion in the manuscript accordingly in section 4.2.2 (lines 2480–2519) to include a brief discussion of these results, and we kindly ask the referee to refer to our response to Referee 1 for further details.

The referee writes:

Furthermore, there is no benchmark regarding the speed of the code, computational resources, or overall computational cost. Including such information would greatly improve the manuscript and help potential users evaluate the practicality of the library.

Our response:

We thank the referee for this suggestion. The computational overhead of GAs is largely determined by the cost of evaluating the fitness function, and is therefore highly problem-specific. For this reason, we did not provide general benchmarking of absolute performance. Instead, we added an analysis of wall-time scaling with the number of CPU threads in two representative examples: the Rastrigin function minimization (figure 2 and discussion added in lines 1645–1669) and scenario 1 of the LISA signal reconstruction example (figure 12 and discussion added in lines 2372–2406). These examples provide practical guidance on performance in typical use cases. We also kindly refer the referee to our response to the other referee report for further comments on computational performance.

Additional changes to the manuscript

1) We added footnote 5 on page 26.

We thank again the referee for their helpful remarks and hope that the revised paper can now be published in SciPost.

---

## Editorial Decision

editorial_decision: